# LOCALIZED RANDOMIZED SMOOTHING
# FOR COLLECTIVE ROBUSTNESS CERTIFICATION

## ABSTRACT

Models for image segmentation, node classification and many other tasks map a single input to multiple labels. By perturbing this single shared input (e.g. the image) an adversary can manipulate several predictions (e.g. misclassify several pixels). A recent collective robustness certificate provides strong guarantees on the number of predictions that are simultaneously robust. This method is however limited to *strictly local* models, where each prediction is associated with a small receptive field. We propose a more general collective certificate for the larger class of *softly local* models, where each output is dependent on the entire input but assigns different levels of importance to different input regions (e.g. based on their proximity in the image). The certificate is based on our novel localized randomized smoothing approach, where the random perturbation strength for different input regions is proportional to their importance for the outputs. The resulting locally smoothed model yields strong collective guarantees while maintaining high prediction quality on both image segmentation and node classification tasks.

## 1 INTRODUCTION

There is a wide range of tasks that require models making multiple predictions based on a single input. For example, semantic segmentation requires assigning a label to each pixel in an image. When deploying such *multi-output* classifiers in practice, their robustness should be a key concern. After all – just like simple classifiers (Szegedy et al., 2014) – they can fall victim to adversarial attacks (Xie et al., 2017; Zügner & Günnemann, 2019; Belinkov & Bisk, 2018). Even without an adversary, random noise or measuring errors could cause one or multiple predictions to unexpectedly change.

In the following, we derive a method that provides provable guarantees on *how many* predictions can be changed by an adversary. Since all outputs operate on the same input, they also have to be attacked simultaneously by choosing a single perturbed input. While attacks on a single prediction may be easy, attacks on different predictions may be mutually exclusive. We have to explicitly account for this fact to obtain a proper *collective robustness certificate* that provides tight bounds.

There already exists a dedicated collective robustness certificate for multi-output classifiers (Schuchardt et al., 2021), but it is only benefical for models we call *strictly local*, where each output depends only on a small, well-defined subset of the input. One example are graph neural networks that classify each node in a graph based only on its neighborhood. Multi-output classifiers used in practice, however, are often only *softly local*. While – unlike strictly local models – all of their predictions are in principle dependent on the entire input, each output may assign different importance to different components. For example, deep convolutional networks used for image segmentation can have very small effective receptive fields (Luo et al., 2016; Liu et al., 2018b), i.e. primarily use a small region of the input in labeling each pixel. Many models used in node classification are based on the homophily assumption that connected nodes are mostly of the same class. Thus, they primarily use features from neighboring nodes to classify each node. Even if an architecture is not inherently softly local, a model may learn a softly local mapping through training. For example, a transformer (Vaswani et al., 2017) can in principle attend to any part of an input sequence. However, in practice the learned attention maps may be "sparse", with the prediction for each token being determined primarily by a few (not necessarily nearby) tokens (Shi et al., 2021).

While an adversarial attack on a single prediction of a softly local model is conceptually no different from that on a single-output classifier, attacking multiple predictions simultaneously can be much

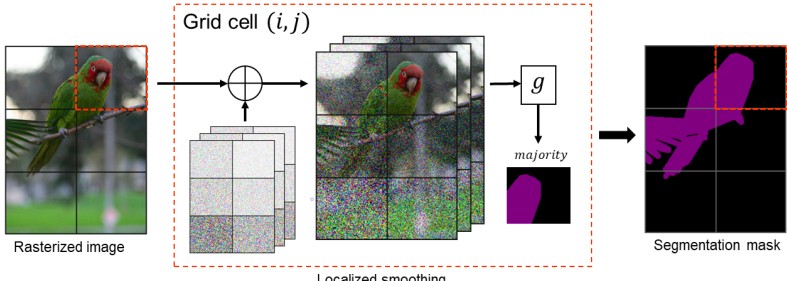

Figure 1: Localized randomized smoothing applied to semantic segmentation. We assume that the most relevant information for labeling a pixel is contained in other nearby pixels. We partition the input image into multiple grid cells. For each grid cell, we sample noisy images from a localized smoothing distribution that applies more noise to far-away, less relevant grid cells. Segmenting all noisy images using base model $g$, cropping the result and computing the majority vote (i.e. the most common label for each pixel) yields a local segmentation mask. These per-cell segmentation masks can then be combined into a complete segmentation mask.

more challenging. By definition, adversarial attacks have to be unnoticeable, meaning the adversary only has a limited budget for perturbing the input. When each output is focused on a different part of the input, the adversary has to decide on where to allocate their adversarial budget and may be unable to attack all outputs at once. Our collective robustness certificate explicitly accounts for this budget allocation problem faced by the adversary and can thus provide stronger robustness guarantees.

Our certificate is based on randomized smoothing (Liu et al., 2018a; Lécuyer et al., 2019; Cohen et al., 2019). Randomized smoothing is a versatile black-box certification method that has originally been proposed for single-output classifiers. Instead of directly analysing a model, it constructs a smoothed classifier that returns the most likely prediction of the model under random perturbations of its input. One can then use statistical methods to certify the robustness of this smoothed classifier. We discuss more details in Section 2. Randomized smoothing is typically used with i.i.d. noise: Each part of the input (e.g. each pixel) independently undergoes random perturbations sampled from the same noise distribution. One can however also use non-i.i.d. noise (Eiras et al., 2021). This results in a smoothed classifier that is certifiably more robust to parts of the input that are smoothed with higher noise levels (e.g. larger standard deviation).

We apply randomized smoothing to softly-local multi-output classifiers in a scheme we call *localized randomized smoothing*: Instead of using the same smoothing distribution for all outputs, we randomly smooth each output (or set of outputs) using a *different* non-i.i.d. distribution that matches its inherent soft locality. Using a low noise level for the most relevant parts of the input allows us to retain a high prediction quality (e.g. accuracy). Less relevant parts of the input can be smoothed with a higher noise level. The resulting certificates (one per output) explicitly quantify how robust each prediction is to perturbations of which section of the input – they are certificates of soft locality.

After certifying each prediction independently using localized randomized smoothing, we construct a (mixed-integer) linear program that combines these per-prediction *base certificates* into a collective certificate that provably bounds the number of simultaneously attackable predictions. This linear program explicitly accounts for soft locality and the budget allocation problem it causes for the adversary. This allows us to prove much stronger guarantees of collective robustness than simply certifying each prediction independently. Our core contributions are:

- *Localized randomized smoothing*, a novel smoothing scheme for multi-output classifiers.
- A *variance smoothing* method for efficiently certifying smoothed models on discrete data.
- A *collective certificate* that leverages our identified common interface for base certificates.

## 2 BACKGROUND AND RELATED WORK

**Randomized smoothing.** Randomized smoothing is a flexible certification technique that can be used for various data types, perturbation models and tasks. For simplicity, we focus on a

classification certificate for $l_2$ perturbations (Cohen et al., 2019). Assume we have a continuous $D$-dimensional input space $\mathbb{R}^D$, a label set $\mathbb{Y}$ and a classifier $g : \mathbb{R}^D \to \mathbb{Y}$. We can use isotropic Gaussian noise with standard deviation $\sigma \in \mathbb{R}_+$ to construct the *smoothed classifier* $f = \arg\max_{y \in \mathbb{Y}} \Pr_{\boldsymbol{z} \sim \mathcal{N}(\boldsymbol{x}, \sigma)} [g(\boldsymbol{z}) = y]$ that returns the most likely prediction of *base classifier* $g$ under the input distribution [1]. Given an input $x \in \mathbb{R}^D$ and the smoothed prediction $y = f(\boldsymbol{x})$, we want to determine whether the prediction is robust to all $l_2$ perturbations of magnitude $\epsilon$, i.e. whether $\forall \boldsymbol{x}' : ||\boldsymbol{x}' - \boldsymbol{x}||_2 \le \epsilon : f(\boldsymbol{x}') = y$. Let $q = \Pr_{\boldsymbol{z} \sim \mathcal{N}(\boldsymbol{x}, \sigma)} [g(\boldsymbol{x}) = y]$ be the probability of $g$ predicting label $y$. The prediction of our smoothed classifier is robust if $\epsilon < \sigma \Phi^{-1}(q)$ (Cohen et al., 2019). This result showcases a trade-off we alluded to in the previous section: The certificate can become stronger if the noise-level (here $\sigma$) is increased. But doing so could also lower the accuracy of the smoothed classifier or reduce $q$ and thus weaken the certificate.

**White-box certificates for multi-output classifiers.** There are multiple recent methods for certifying the robustness of specific multi-output models (see, for example, (Tran et al., 2021; Zügner & Günnemann, 2019; Bojchevski & Günnemann, 2019; Zügner & Günnemann, 2020; Ko et al., 2019; Ryou et al., 2021; Shi et al., 2020; Bonaert et al., 2021)) by analyzing their specific architecture and weights. They are however not designed to certify collective robustness. They can only determine independently for each prediction whether or not it can be adversarially attacked.

**Collective robustness certificates.** Most directly related to our work is the certificate of Schuchardt et al. (2021). Like ours, it combines many per-prediction certificates into a collective certificate. But, unlike our novel localized smoothing approach, their certification procedure is only beneficial for strictly local models, i.e. models whose outputs operate on small subsets of the input. Furthermore, their certificate assumes binary data, while our certificate defines a common interface for various data types and perturbation models. A more detailed comparison can be found in Section D. Recently, Fischer et al. (2021) proposed a certificate for semantic segmentation. They consider a different notion of collective robustness: They are interested in determining whether *all* predictions are robust. In Section C.4 we discuss their method in detail and show that, when used for certifying our notion of collective robustness (i.e. the number of robust predictions), their method is no better than certifying each output independently using the certificate of Cohen et al. (2019). Furthermore, our certificate can be used to provide equally strong guarantees for their notion of collective robustness by checking whether the number of certified predictions equals the overall number of predictions. Another method that can be used for certifying collective robustness is center smoothing (Kumar & Goldstein, 2021). Center smoothing bounds how much a vector-valued prediction changes w.r.t to a distance function under adversarial perturbations. With the $l_0$ pseudo-norm as the distance function, center smoothing bounds how many predictions of a classifier can be simultaneously changed.

**Randomized smoothing with non-i.i.d. noise.** While not designed for certifying collective robustness, two recent certificates for non-i.i.d. Gaussian (Fischer et al., 2020) and uniform smoothing (Eiras et al., 2021) can be used as a component of our collective certification approach: They can serve as per-prediction base certificates, which can then be combined into our stronger collective certificate (more details in Section 4). Note that we do not use the procedure for optimizing the smoothing distribution proposed by Eiras et al. (2021), as this would enable adversarial attacks on the smoothing distribution itself and invalidate the certificate (see discussion by Wang et al. (2021)).

## 3 COLLECTIVE THREAT MODEL

Before certifying robustness, we have to define a threat model, which specifies the type of model that is attacked, the objective of the adversary and which perturbations they are allowed to use. We assume that we have a multi-output classifier $f : \mathbb{X}^{D_{\text{in}}} \to \mathbb{Y}^{D_{\text{out}}}$, that maps from a $D_{\text{in}}$-dimensional vector space to $D_{\text{out}}$ labels from label set $\mathbb{Y}$. We further assume that this classifier $f$ is the result of randomly smoothing a base classifier $g$, as discussed in Section 2. To simplify our notation, we write $f_n$ to refer to the function $\boldsymbol{x} \mapsto f(\boldsymbol{x})_n$ that outputs the $n$-th label. Given this multi-output classifier $f$, an input $\boldsymbol{x} \in \mathbb{X}^{D_{\text{in}}}$ and the resulting vector of predictions $\boldsymbol{y} = f(\boldsymbol{x})$, the objective of the adversary is to cause as many predictions from a set of targeted indices $\mathbb{T} \subseteq \{1, \dots, D_{\text{out}}\}$ to change. That is, their objective is $\min_{\boldsymbol{x}' \in \mathbb{B}_{\boldsymbol{x}}} \sum_{n \in \mathbb{T}} \mathrm{I}[f_n(\boldsymbol{x}') = y_n]$, where $\mathbb{B}_{\boldsymbol{x}} \subseteq \mathbb{X}^{D_{\text{in}}}$ is the perturbation model. Importantly, note that the minimization operator is outside the sum, meaning the predictions have to

---

[1]In practice, all probabilities have to be estimated using Monte Carlo sampling (see discussion in Section C).

be attacked using a single input. As is common in robustness certification, we assume a norm-bound perturbation model. That is, given an input $\boldsymbol{x} \in \mathbb{X}^{D_{\text{in}}}$, the adversary is only allowed to use perturbed inputs from the set $\mathbb{B}_{\boldsymbol{x}} = \left\{ \boldsymbol{x}' \in \mathbb{X}^{D_{\text{in}}} \mid ||\boldsymbol{x}' - \boldsymbol{x}||_p \leq \epsilon \right\}$ with $p, \epsilon \geq 0$.

## 4  A RECIPE FOR COLLECTIVE CERTIFICATES

Before discussing technical details, we provide a high-level overview of our method. In localized randomized smoothing, we assign each output $g_n$ of a base classifier $g$ its own smoothing distribution $\Psi^{(n)}$ that matches our assumptions or knowledge about the base classifier's soft locality, i.e. for each $n \in \{1, \ldots, D_{\text{out}}\}$ choose a $\Psi^{(n)}$ that induces more noise in input components that are less relevant for $g_n$. For example, in Fig. 1, we assume that far-away regions of the image are less relevant and thus perturb pixels in the bottom left with more noise when classifying pixels in the top-right corner. The chosen smoothing distributions can then be used to construct the smoothed classifier $f$.

Given an input $\boldsymbol{x} \in \mathbb{X}^{D_{\text{in}}}$ and the corresponding smoothed prediction $\boldsymbol{y} = f(\boldsymbol{x})$, randomized smoothing makes it possible to compute per-prediction *base certificates*. That is, for each $y_n$, one can compute a set $\mathbb{H}^{(n)} \subseteq \mathbb{X}^{D_{\text{in}}}$ of perturbed inputs that the prediction is robust to, i.e. $\forall \boldsymbol{x}' \in \mathbb{H}_n :$ $f_n(\boldsymbol{x}') = y_n$. Our motivation for using non-i.i.d. distributions is that the $\mathbb{H}^{(n)}$ will guarantee more robustness for input dimensions smoothed with more noise, i.e. quantify model locality.

The objective of our adversary is $\min_{\boldsymbol{x}' \in \mathbb{B}_{\boldsymbol{x}}} \sum_{n \in \mathbb{T}} \mathbb{I}\left[f_n(\boldsymbol{x}') = y_n\right]$ with collective perturbation model $\mathbb{B}_{\boldsymbol{x}} \subseteq \mathbb{X}^{D_{\text{in}}}$. That is, they want to change as many predictions from the targeted set $\mathbb{T}$ as possible. A trivial lower bound can be obtained by counting how many predictions are – according to the base certificates – provably robust to the collective threat model. This can be expressed as $\sum_{n \in \mathbb{T}} \min_{\boldsymbol{x}' \in \mathbb{B}_{\boldsymbol{x}}} \mathbb{I}\left[\boldsymbol{x}' \in \mathbb{H}^{(n)}\right]$. In the following, we refer to this as the *naïve collective certificate*.

Thanks to our proposed localized smoothing scheme, we can use the following, tighter bound:

$$\min_{\boldsymbol{x}' \in \mathbb{B}_{\boldsymbol{x}}} \sum_{n \in \mathbb{T}} \mathbb{I}\left[f_n(\boldsymbol{x}') = y_n\right] \geq \min_{\boldsymbol{x}' \in \mathbb{B}_{\boldsymbol{x}}} \sum_{n \in \mathbb{T}} \mathbb{I}\left[\boldsymbol{x}' \in \mathbb{H}^{(n)}\right], \tag{1}$$

which preserves the fact that the adversary has to choose a single perturbed input. Because we use different non-i.i.d. smoothing distributions for different outputs, we provably know that each $f_n$ has varying levels of robustness for different parts of the input and that these robustness levels differ among outputs. Thus, in the r.h.s. problem the adversary has to allocate their limited budget across various input dimensions and may be unable to attack all predictions at once, just like when attacking the classifier in the l.h.s. objective (recall Section 1). This makes our collective certificate stronger than the naïve collective certificate, which allows each prediction to be attacked independently.

As stated in Section 1, the idea of combining base certificates into stronger collective certificates has already been explored by Schuchardt et al. (2021). But instead of using localized smoothing to capture the (soft) locality of a model, their approach leverages the fact that perturbations outside an output's receptive field can be ignored. For softly local models, which have receptive fields covering the entire input, their certificate is no better than the naïve certificate. Another novel insight underlying our approach is that various non-i.i.d. randomized smoothing certificates share a common interface, which makes our method applicable to diverse data types and perturbation models. In the next section, we formalize this common interface. We then discuss how it allows us to compute the collective certificate from Eq. 1 using (mixed-integer) linear programming.

## 5  COMMON INTERFACE FOR BASE CERTIFICATES

A base certificate for a prediction $y_n = f_n(\boldsymbol{x})$ is a set $\mathbb{H}_n \subseteq \mathbb{X}^{D_{\text{in}}}$ of perturbed inputs that $y_n$ is provably robust to, i.e $\forall \boldsymbol{x}' \in \mathbb{H}_n : f_n(\boldsymbol{x}') = y_n$. Note that base certificates do not have to be exact, but have to be sound, i.e. they do not have to specify *all* inputs to which the $f_n$ are robust but they must not contain any adversarial examples. As a common interface for base certificates, we propose that the sets $\mathbb{H}_n$ are parameterized by a weight vector $\boldsymbol{w}^{(n)} \in \mathbb{R}^{D_{\text{in}}}$ and a scalar $\eta^{(n)}$ that define a linear constraint on the element-wise distance between perturbed inputs and the clean input:

$$\mathbb{H}^{(n)} = \left\{ \boldsymbol{x}' \in \mathbb{X}^{D_{\text{in}}} \,\middle|\, \sum_{d=1}^{D_{\text{in}}} w_d^{(n)} \cdot |x_d' - x_d|^\kappa < \eta^{(n)} \right\}. \tag{2}$$

The weight vector encodes how robust $y_n$ is to perturbations of different components of the input. The scalar $\kappa$ is important for collective robustness certification, because it encodes which collective perturbation model the base certificate is compatible with. For example, $\kappa = 2$ means that the base certificate can be used for certifying collective robustness to $l_2$ perturbations.

In the following, we present two base certificates implementing our interface: One for $l_2$ perturbations of continuous data and one for perturbations of binary data. In Section B, we further present a certificate for binary data that can distinguish between adding and deleting bits and a certificate for $l_1$ perturbations of continuous data. All base certificates guarantee more robustness for parts of the input smoothed with a higher noise level. The certificates for continuous data are based on known results (Fischer et al., 2020; Eiras et al., 2021) and merely reformulated to match our proposed interface, so that they can be used as part of our collective certification procedure. The certificates for discrete data however are original and based on the novel concept of *variance smoothing*.

**Gaussian smoothing for $l_2$ perturbations of continuous data** The first base certificate is a generalization of Gaussian smoothing to anisotropic noise, a corollary of Theorem A.1 from (Fischer et al., 2020). In the following, $\mathrm{diag}(\boldsymbol{z})$ refers to a diagonal matrix with diagonal entries $\boldsymbol{z}$ and $\Phi^{-1} : [0,1] \to \mathbb{R}$ refers to the the standard normal inverse cumulative distribution function.

**Proposition 1.** *Given an output $g_n : \mathbb{R}^{D_{\mathrm{in}}} \to \mathbb{Y}$, let $f_n(\boldsymbol{x}) = \mathrm{argmax}_{y \in \mathbb{Y}} \Pr_{\boldsymbol{z} \sim \mathcal{N}(\boldsymbol{x}, \boldsymbol{\Sigma})} [g_n(\boldsymbol{z}) = y]$ be the corresponding smoothed output with $\boldsymbol{\Sigma} = \mathrm{diag}(\boldsymbol{\sigma})^2$ and $\boldsymbol{\sigma} \in \mathbb{R}_+^{D_{\mathrm{in}}}$. Given an input $\boldsymbol{x} \in \mathbb{R}^{D_{\mathrm{in}}}$ and smoothed prediction $y_n = f_n(\boldsymbol{x})$, let $q = \Pr_{\boldsymbol{z} \sim \mathcal{N}(\boldsymbol{x}, \boldsymbol{\Sigma})} [g_n(\boldsymbol{z}) = y_n]$. Then, $\forall \boldsymbol{x}' \in \mathbb{H}^{(n)} :$ $f_n(\boldsymbol{x}') = y_n$ with $\mathbb{H}^{(n)}$ defined as in Eq. 2, $w_d = \frac{1}{\sigma_d^2}$, $\eta = \left(\Phi^{(-1)}(q)\right)^2$ and $\kappa = 2$.*

**Bernoulli variance smoothing for perturbations of binary data** For binary data, we use a smoothing distribution $\mathcal{F}(\boldsymbol{x}, \boldsymbol{\theta})$ with $\theta \in [0,1]^{D_{\mathrm{in}}}$ that independently flips the $d$'th bit with probability $\theta_d$, i.e. for $\boldsymbol{x}, \boldsymbol{z} \in \{0,1\}^{D_{\mathrm{in}}}$ and $\boldsymbol{z} \sim \mathcal{F}(\boldsymbol{x}, \boldsymbol{\theta})$ we have $\Pr[z_d \neq x_d] = \theta_d$. A corresponding certificate could be derived by generalizing (Lee et al., 2019), which considers a single shared $\theta \in [0,1]$ with $\forall d : \theta_d = \theta$. However, the cost for computing this certificate would be exponential in the number of unique values in $\boldsymbol{\theta}$. We therefore propose a more efficient alternative. Instead of constructing a smoothed classifier that returns the most likely labels of the base classifier (as discussed in Section 2), we construct a smoothed classifier that returns the labels with the highest expected softmax scores (similar to CDF-smoothing (Kumar et al., 2020)). For this smoothed model, we can compute a robustness certificate in constant time. The certificate requires determining both the expected value and variance of softmax scores. We therefore call this method *variance smoothing*. While we use it for binary data, it is a general-purpose technique that can be applied to arbitrary domains and smoothing distributions (see discussion in Section B.2). In the following, we assume the label set $\mathbb{Y}$ to consist of numerical labels $\{1, \ldots, |\mathbb{Y}|\}$, which simplifies our notation.

**Theorem 1.** *Given an output $g_n : \{0,1\}^{D_{\mathrm{in}}} \to \Delta_{|\mathbb{Y}|}$ mapping to scores from the $|\mathbb{Y}|$-dimensional probability simplex, let $f_n(\boldsymbol{x}) = \mathrm{argmax}_{y \in \mathbb{Y}} \mathbb{E}_{\boldsymbol{z} \sim \mathcal{F}(\boldsymbol{x}, \boldsymbol{\theta})} [g_n(\boldsymbol{z})_y]$ be the corresponding smoothed classifier with $\boldsymbol{\theta} \in [0,1]^{D_{\mathrm{in}}}$. Given an input $\boldsymbol{x} \in \{0,1\}^{D_{\mathrm{in}}}$ and smoothed prediction $y_n = f_n(\boldsymbol{x})$, let $\mu = \mathbb{E}_{\boldsymbol{z} \sim \mathcal{F}(\boldsymbol{x}, \boldsymbol{\theta})} [g_n(\boldsymbol{z})_y]$ and $\sigma^2 = \mathrm{Var}_{\boldsymbol{z} \sim \mathcal{F}(\boldsymbol{x}, \boldsymbol{\theta})} [g_n(\boldsymbol{z})_y]$. Then, $\forall \boldsymbol{x}' \in \mathbb{H}^{(n)} : f_n(\boldsymbol{x}') = y_n$ with $\mathbb{H}^{(n)}$ defined as in Eq. 2, $w_d = \ln\left(\frac{(1-\theta_d)^2}{\theta_d} + \frac{(\theta_d)^2}{1-\theta_d}\right)$, $\eta = \ln\left(1 + \frac{1}{\sigma^2}\left(\mu - \frac{1}{2}\right)^2\right)$ and $\kappa = 0$.*

## 6 COMPUTING THE COLLECTIVE ROBUSTNESS CERTIFICATE

With our common interface for base certificates in place, we can discuss how to compute the collective robustness certificate $\min_{\boldsymbol{x}' \in \mathbb{B}_{\boldsymbol{x}}} \sum_{n \in \mathbb{T}} \mathbb{I}\left[\boldsymbol{x}' \in \mathbb{H}^{(n)}\right]$ from Eq. 1. The result bounds the number of predictions $y_n$ with $n \in \{1, \ldots, D_{\mathrm{out}}\}$ that can be simultaneously attacked by the adversary. In the following, we assume that the base certificates were obtained by using a smoothing distribution that is compatible with our $l_p$ collective perturbation model (i.e. $\kappa = p$), for example by using Gaussian noise for $p = 2$ or Bernoulli noise for $p = 0$. Inserting the definition of our base certificate interface from Eq. 2 and rewriting our perturbation model $\mathbb{B}_{\boldsymbol{x}} = \left\{\boldsymbol{x}' \in \mathbb{X}^{D_{\mathrm{in}}} \mid ||\boldsymbol{x}' - \boldsymbol{x}||_p \leq \epsilon\right\}$ as $\left\{\boldsymbol{x}' \in \mathbb{X}^{D_{\mathrm{in}}} \mid \sum_{d=1}^{D_{\mathrm{in}}} |x_d' - x_d|^p \leq \epsilon^p\right\}$, our objective from Eq. 1 can be expressed as

$$\min_{\boldsymbol{x}' \in \mathbb{X}^{D_{\mathrm{in}}}} \sum_{n \in \mathbb{T}} \mathbb{I}\left[\sum_{d=1}^{D_{\mathrm{in}}} w_d^{(n)} \cdot |x_d' - x_d|^p < \eta^{(n)}\right] \quad \text{s.t.} \quad \sum_{d=1}^{D_{\mathrm{in}}} |x_d' - x_d|^p \leq \epsilon^p. \tag{3}$$

We can see that the perturbed input $\boldsymbol{x}'$ only affects the element-wise distances $|x_d' - x_d|^p$. Rather than optimizing $\boldsymbol{x}'$, we can instead directly optimize these distances, i.e. determine how much adversarial budget is allocated to each input dimension. For this, we define a vector of variables $\boldsymbol{b} \in \mathbb{R}_+^{D_{\text{in}}}$ (or $\boldsymbol{b} \in \{0,1\}^{D_{\text{in}}}$ for binary data). Replacing sums with inner products, we can restate Eq. 3 as

$$\min_{\boldsymbol{b} \in \mathbb{R}_+^{D_{\text{in}}}} \sum_{n \in \mathbb{T}} \mathrm{I}\left[\boldsymbol{b}^T \boldsymbol{w}^{(n)} < \eta^{(n)}\right] \quad \text{s.t.} \quad \text{sum}\{\boldsymbol{b}\} \leq \epsilon^p. \tag{4}$$

In a final step, we replace the indicator functions in Eq. 4 with a vector of boolean variables $\boldsymbol{t} \in \{0,1\}^{D_{\text{out}}}$. Define the constants $\underline{\eta}^{(n)} = \epsilon^p \cdot \min\left(0, \min_d w_d^{(n)}\right)$. Then,

$$\min_{\boldsymbol{b} \in \mathbb{R}_+^{D_{\text{in}}}, \boldsymbol{t} \in \{0,1\}^{D_{\text{out}}}} \sum_{n \in \mathbb{T}} t_n \quad \text{s.t.} \quad \forall n : \boldsymbol{b}^T \boldsymbol{w}^{(n)} \geq t_n \underline{\eta}^{(n)} + (1 - t_n)\eta^{(n)}, \quad \text{sum}\{\boldsymbol{b}\} \leq \epsilon^p. \tag{5}$$

is equivalent to Eq. 4. The first constraint guarantees that $t_n$ can only be set to $0$ if the l.h.s. is greater or equal $\eta^{(n)}$, i.e. only when the base certificate can no longer guarantee robustness. The term involving $\underline{\eta}^{(n)}$ ensures that for $t_n = 1$ the problem is always feasible[2]. Eq. 5 can be solved using any mixed-integer linear programming solver.

While the resulting MILP bears some semblance to that of Schuchardt et al. (2021), it is conceptually different. When evaluating their base certificates, they mask out parts of the budget vector $\boldsymbol{b}$ based on a model's strict locality, while we weigh the budget vector based on the soft locality guaranteed by the base certificates. In addition, thanks to the interface specified in Section 5, our problem only involves a single linear constraint per prediction, making it much smaller and more efficient to solve. Interestingly, when using randomized smoothing base certificates for binary data, our certificate subsumes theirs, i.e. can provide the same robustness guarantees (see Section D.2).

**Improving efficiency.** Still, the efficiency of our certificate in Eq. 5. certificate can be further improved. In Section A, we show that partitioning the outputs into $N_{\text{out}}$ subsets sharing the same smoothing distribution and the the inputs into $N_{\text{in}}$ subsets sharing the same noise level (for example like in Fig. 1), as well as quantizing the base certificate parameters $\eta^{(n)}$ into $N_{\text{bin}}$ bins reduces the number of variables and constraints from $D_{\text{in}} + D_{\text{out}}$ and $D_{\text{out}} + 1$ to $N_{\text{in}} + N_{\text{out}} \cdot N_{\text{bins}}$ and $N_{\text{out}} \cdot N_{\text{bins}} + 1$, respectively. We can thus control the problem size independent of the data's dimensionality. We further derive a linear relaxation of the mixed-integer problem, which can be more efficiently solved while preserving the soundness of the certificate.

## 7 LIMITATIONS

The main limitation of our approach is that it assumes softly local models. While it can be applied to arbitrary multi-output classifiers, it may not necessarily result in better certificates than randomized smoothing with i.i.d. distributions. Furthermore, choosing the smoothing distributions requires some a-priori knowledge or assumptions about which parts of the input are how relevant to making a prediction. Our experiments show that natural assumptions like homophily can be sufficient for choosing effective smoothing distributions. But doing so in other tasks may be more challenging.

A limitation of (most) randomized smoothing certificates is that they use sampling to approximate the smoothed classifier. Because we use different smoothing distributions for different outputs, we can only use a fraction of the samples for each output. As discussed in Section A.1, we can alleviate this problem by sharing smoothing distributions among multiple outputs. Our experiments show that despite this issue, our method outperforms certificates that use a single smoothing distribution. Still, future work should try to improve the sample efficiency of randomized smoothing (for example by developing more methods for de-randomized smoothing (Levine & Feizi, 2020)). Any such advance could then be incorporated into our localized smoothing framework.

## 8 EXPERIMENTAL EVALUATION

Our experimental evaluation has three objectives 1.) Verifying our main claim that localized randomized smoothing offers a better trade-off between accuracy and certifiable robustness than smoothing

---

[2]Because $\underline{\eta}^{(n)}$ is the smallest value $\boldsymbol{b}^T \boldsymbol{w}^{(n)}$ can take on, i.e. $\min_{\boldsymbol{b} \in \mathbb{R}_+^{D_{\text{in}}}} \boldsymbol{b}^T \boldsymbol{w}_d^{(n)}$ s.t. $\text{sum}\{\boldsymbol{b}\} \leq \epsilon^p$.

with i.i.d. distributions. 2.) Determining to what extend the linear program underlying the proposed collective certificate strengthens our robustness guarantees. 3.) Assessing the efficacy of our novel variance smoothing certificate for binary data. Any of the used datasets and classifiers only serve as a means of comparing certificates. We thus use well-known and well-established architectures instead of overly focusing on maximizing prediction accuracy by using the latest SOTA models.

We use two metrics to quantify certificate strength: Certified accuracy (i.e. the percentage of correct and certifiably robust predictions) and certified ratio (i.e. the percentage of certifiably robust predictions, regardless of correctness)[3]. As single-number metrics, we report the AUC of the certified accuracy/ratio functions w.r.t. adversarial budget $\epsilon$ (not to be confused with certifying some AUC metric). For localized smoothing, we evaluate both the naïve collective certificate, i.e. certifying predictions independently (see Section 4), and the proposed LP-based certificate (using the linearly relaxed version from Appendix A.4). We compare our method to two baselines using i.i.d. randomized smoothing: The naïve collective certificate and center smoothing (Kumar & Goldstein, 2021). For softly local models, the certificate of Schuchardt et al. (2021) is equivalent to the naïve baseline. When used to certify the number of robust predictions, the segmentation certificate of Fischer et al. (2021) is at most as strong as the naïve baseline (see Section C.4). Thus, our method is compared to all existing collective certificates listed in Section 2. In all experiments, we use Monte Carlo randomized smoothing. More details on the experimental setup can be found in Section E.

## 8.1 SEMANTIC SEGMENTATION

**Dataset and model.** We evaluate our certificate for continuous data and $l_2$ perturbations on the Pascal-VOC 2012 segmentation validation set. Training is performed on 10582 pairs of training samples extracted from SBD[4] (Hariharan et al., 2011), To increase batch sizes and thus allow a more thorough investigation of different smoothing parameters, all images are downscaled to $50\%$ of their original size. Our base model is a U-Net segmentation model with a ResNet-18 backbone. To obtain accurate and robust smoothed classifiers, base models should be trained on the smoothing distribution. We thus train $51$ different instances of our base model, augmenting the training data with a different $\sigma_{\text{train}} \in \{0, 0.01, \ldots, 0.5\}$. At test time, when evaluating a baseline i.i.d. certificate with smoothing distribution $\mathcal{N}(0, \sigma)$, we load the model trained with $\sigma_{\text{train}} = \sigma$. To perform localized randomized smoothing, we choose parameters $\sigma_{\min}, \sigma_{\max} \in \mathbb{R}_+$ and partition all images into regular grids of size $4 \times 6$ (similar to

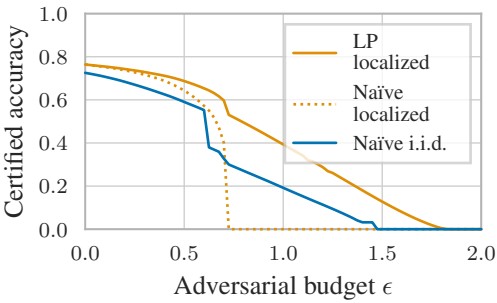

Figure 2: Certified ratios of U-Net models under varying $\epsilon$. We compare the naïve i.i.d. baseline ($\sigma = 0.4$) to localized smoothing ($\sigma_{\min} = 0.25, \sigma_{\max} = 1.5$). Combining the base certificates via linear programming (solid orange line) instead of evaluating them independently (dotted orange line) outperforms the baseline.

example Fig. 1). To classify pixels in grid cell $(i, j)$, we sample noise for grid cell $(k, l)$ using $\mathcal{N}(0, \sigma')$, with $\sigma' \in [\sigma_{\min}, \sigma_{\max}]$ chosen proportional to the distance of $(i, j)$ and $(k, l)$ (more details in Section E.2.1). As the base model, we load the one trained with $\sigma_{\text{train}} = \sigma_{\min}$. Using the same distribution at train and test time for the i.i.d. baselines but not for localized smoothing is meant to skew the results in the baseline's favor. But, in Section E.2.3, we also repeat our experiments using the same base model for i.i.d. and localized smoothing.

**Evaluation.** The main goal of our experiments on segmentation is to verify that localized smoothing can offer a better trade-off between accuracy and certifiable robustness. That is, for all or most $\sigma$, there are $\sigma_{\min}, \sigma_{\max}$ such that the locally smoothed model has higher accuracy and certifiable collective robustness than i.i.d. smoothing baselines using $\mathcal{N}(0, \sigma)$. Because $\sigma, \sigma_{\min}, \sigma_{\max} \in \mathbb{R}_+$, we can not evaluate all possible combinations. We therefore use the following scheme: We focus on the case $\sigma \in [0, 0.5]$, which covers all distributions used in (Kumar & Goldstein, 2021) and

---

[3]In the case of image segmentation, we compute these metrics per image and then average over the dataset.
[4]Also known as "Pascal trainaug"

(Fischer et al., 2021). First, we evaluate our two baselines for five $\sigma \in \{0.1, 0.2, 0.3, 0.4, 0.5\}$. This results in baseline models with diverse levels of accuracy and robustness (e.g. the accuracy of the naïve baseline shrinks from $87.7\%$ to $64.9\%$ and the AUC of its certified accuracy grows from $0.17$ to $0.644$). We then test whether, for each of the $\sigma$, we can find $\sigma_{\min}, \sigma_{\max}$ such that the locally smoothed models attains higher accuracy and is certifiably more robust. Finally, to verify that $\{0.1, 0.2, 0.3, 0.4, 0.5\}$ were not just a particularly poor choice of baseline parameters, we fix the chosen $\sigma_{\min}, \sigma_{\max}$. We then perform a fine-grained search over $\sigma \in [0, 0.5]$ with resolution $0.01$ to find a baseline model that has at least the same accuracy and certifiable robustness (as measured by certificate AUC) as any of the fixed locally smoothed models. If this is not possible, this provides strong evidence that the proposed smoothing scheme and certificate indeed offer a better trade-off.

Fig. 2 shows one example. For $\sigma = 0.4$, the naïve i.i.d. baseline has an accuracy of $72.5\%$. With $\sigma_{\min} = 0.25, \sigma_{\max} = 1.5$, the proposed localized smoothing certificate yields both a higher accuracy of $76.4\%$ and a higher certified accuracy for all $\epsilon$. It can certify robustness for $\epsilon$ up to $1.825$, compared to $1.45$ of the baseline and the AUC of its certified accuracy curve is $43.1\%$ larger. Fig. 2 also highlights the usefulness of the linear program we derived in Section 5: Evaluating the localized smoothing base certificates independently, i.e. computing the naïve collective certificate (dotted orange line), is not sufficient for outperforming the baseline. But combining them via the proposed linear program drastically increases the certified accuracy

The results for all other combinations of smoothing distribution parameters, both baselines and both metrics of certificate strength can be found in Section E.2.3. Tables 1 and 2 summarize the first part of our evaluation procedure, in which we optimize the localized smoothing parameters. Safe for one exception (with $\sigma = 0.2$, center smoothing has a lower accuracy, but slightly larger certified ratio), the locally smoothed models have the same or higher accuracy, but provide stronger robustness guarantees. The difference is particularly large for $\sigma \in \{0.3, 0.4, 0.5\}$, where the accuracy of models smoothed with i.i.d. noise drops off, while our localized smoothing distribution preserves the most relevant parts of the image to allow for high accuracy. Table 5 summarizes the second part of our evaluation scheme, in which we perform a fine-grained search over $[0, 0.5]$. We find that there is no $\sigma$ such that either of the i.i.d. baselines can outperform any of the chosen locally smoothed models w.r.t. AUC of their certified accuracy or certified ratio curves. This is ample evidence for our claim that localized smoothing offers a better trade-off than i.i.d. smoothing. Also, the collective LPs caused little computational overhead (avg. $0.68\,\mathrm{s}$ per LP, more details in Section E.2.3).

## 8.2 Node classification

**Dataset and model.** We evaluate our certificate for binary data on the Cora-ML node classification dataset. We use two different base-models: Approximate Personalized Propagation of Neural Predictions (APPNP) (Klicpera et al., 2019) and a 6-layer Graph Convolutional network (GCN) (Kipf & Welling, 2017). Both models have a receptive field that covers most or all of the graph, meaning they are softly local. For details on model and training parameters, see Section E.3.1.

As center smoothing has only been derived for Gaussian smoothing, we only compare to the naïve baseline. For both, the baseline and our localized smoothing certificate, we use sparsity-aware randomized smoothing (Bojchevski et al., 2020) , i.e. flip 1-bits and 0-bits with different probabilities ($\theta^-$ and $\theta^+$, respectively), which allows us to certify different levels of robustness to deletions and additions of bits. With localized randomized smoothing, we use the variance smoothing base certificate derived in Section B.2.2. We choose the distribution parameters for localized smoothing based on an assumption of homophily, i.e. nearby nodes are most relevant for classifying a node. We partition the graph into 5 clusters and define parameters $\theta^{\pm}_{\min}$ and $\theta^{\pm}_{\max}$. When classifying a node in cluster $i$, we randomly smooth attributes in cluster $j$ with $\theta^+_{ij}, \theta^-_{ij}$ that are based on linearly interpolating in $[\theta^-_{\min}, \theta^-_{\max}]$ and $[\theta^-_{\min}, \theta^-_{\max}]$ based on the affinity of the clusters (details in Section E.3.1).

**Evaluation.** We first evaluate the new variance-based certificate and compare it to the certificate derived by Bojchevski et al. (2020). For this, we use only one cluster, meaning we use the same smoothing distribution for both. Fig. 11 in Section E.3 shows that the variance certificate is weaker than the baseline for additions, but better for deletions. It appears sufficiently effective to be used as a base certificate and integrated into a stronger, collective certificate.

The parameter space of our smoothing distributions is large. For the localized approach we have four continuous parameters, as we have to specify both the minimal and maximal noise values.

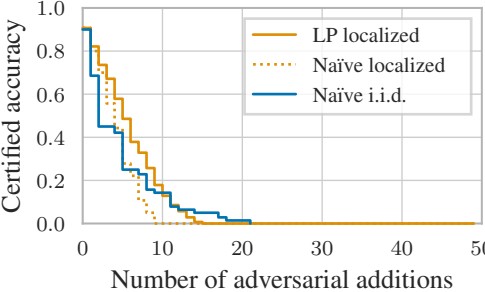 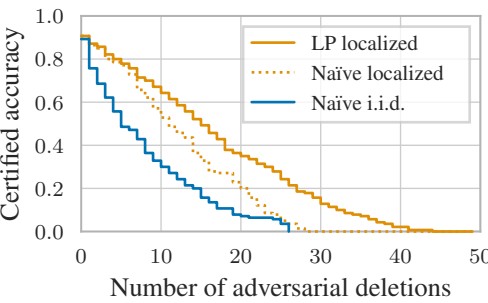

Figure 3: Certified accuracy for an APPNP model under varying number of attribute additions (left) and deletions (right). We compare localized smoothing ($\theta^+_{\min} = 0.05$, $\theta^-_{\min} = 0.65$, $\theta^+_{\max} = 0.08$, $\theta^-_{\max} = 0.95$) to two separately optimized naïve i.i.d. smoothing baselines: addition ($\theta^+ = 0.04$, $\theta^- = 0.61$) and deletion ($\theta^+ = 0.04$, $\theta^- = 0.68$). Combining the localized smoothing base certificates via linear programming (solid orange line) instead of evaluating them independently (dotted line) allows us to outperform the baselines for most adversarial budgets.

Therefore, it is difficult to show that our approach achieves a better accuracy-robustness trade-off over the whole noise space. However, we can investigate the accuracy-robustness trade-off within some areas of this space. For the localized approach we choose a few fixed combinations of the noise parameters $\theta \pm_{\min}$ and $\theta^\pm_{\max}$. To show our claim, we then optimise the baselines with parameters in an interval around our $\theta^+_{\min}$ and $\theta^-_{\min}$. This is a smaller space, as the baselines only have two parameters. We select the baseline whose certified accuracy curve has the largest AUC. We perform the search for the best baseline for the addition and deletion scenario independently, i.e., the best baseline model for addition and deletion does not have to be the same.

In Fig. 3, we see the certified accuracy of an APPNP model for a varying number of attribute additions and deletions (left and right respectively). To find the best distribution parameters for the baselines, we evaluated combinations of $\theta^+ \in \{0.04, 0.055, 0.07\}$ and $\theta^- \in [0.1, \ldots, 0.827]$, using 11 equally spaced values for the interval. For adversarial additions, the best baseline yields a certified accuracy curve with an AUC of $4.51$ compared to our $5.65$. The best baseline for deletions has an AUC of $7.76$ compared to our $16.26$. Our method outperforms these optimized baselines for most adversarial budgets, while maintaining the same clean accuracy (i.e. certified accuracy at $\epsilon = 0$). Experiments with different noise parameters and classifiers can be found in Section E.3. In general, we find that we significantly outperform the baseline when certifying robustness to deletions, but often have weaker certificates for additions (which may be inherent to the variance smoothing base certificates). Due to the large continuous parameter space, we cannot claim that localized smoothing outperforms the naïve baseline everywhere. However, our results show that, for the tested parameter regions, localized smoothing can provide a significantly better accuracy-robustness trade-off.

We found that using the collective LP instead of naïvely combining the base certificates can result in much stronger certificates: The AUC of the certified accuracy curve (averaged over all experiments) increased by $38.8\%$ and $33.6\%$ for addition and deletion, respectively. The collective LPs caused little computational overhead (avg. $10.9$ s per LP, more details in Section E.3.3).

## 9 CONCLUSION

In this work, we have proposed the first collective robustness certificate for softly local multi-output classifiers. It is based on localized randomized smoothing, i.e. randomly smoothing different outputs using different non-i.i.d. smoothing distributions matching the model's locality. We have shown how per-output certificates based on localized smoothing can be computed and that they share a common interface. This interface allows them to be combined into a strong collective robustness certificate. Experiments on image segmentation and node classification tasks demonstrate that localized smoothing can offer a better robustness-accuracy trade-off than existing randomized smoothing techniques. Our results show that locality is linked to robustness, which suggests the research direction of building more effective local models to robustly solve multi-output tasks.

## 10 REPRODUCIBILITY STATEMENT

We prove all theoretic results that were not already derived in the main text in Appendices A to C. To ensure reproducibility of the experimental results we provide detailed descriptions of the evaluation process with the respective parameters in Section E.2 and Section E.3. Code will be made available to reviewers via an anonymous link posted on OpenReview, as suggested by the guidelines.

## 11 ETHICS STATEMENT

In this paper, we propose a method to increase the robustness of machine learning models against adversarial perturbations and to certify their robustness. We see this as an important step towards general usage of models in practice, as many existing methods are brittle to crafted attacks. Through the proposed method, we hope to contribute to the safe usage of machine learning. However, robust models also have to be seen with caution. As they are harder to fool, harmful purposes like mass surveillance are harder to avoid. We believe that it is still necessary to further research robustness of machine learning models as the positive effects can outweigh the negatives, but it is necessary to discuss the ethical implications of the usage in any specific application area.

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

## A  IMPROVING EFFICIENCY

In this section, we discuss different modifications to our collective certificate that improve its sample efficiency and allow us fine-grained control over the size of the collective linear program. We further discuss a linear relaxation of our collective linear program. All of the modifications preserve the soundness of our collective certificate, i.e. we still obtain a provable bound on the number of predictions that can be simultaneously attacked by an adversary. To avoid constant case distinctions, we first present all results for real-valued data, i.e. $\mathbb{X} = \mathbb{R}$, before mentioning any additional precautions that may be needed when working with binary data.

### A.1  SHARING SMOOTHING DISTRIBUTIONS AMONG OUTPUTS

In principle, our proposed certificate allows a different smoothing distribution $\Psi^{(n)}$ to be used per output $g_n$ of our base model. In practice, where we have to estimate properties of the smoothed classifier using Monte Carlo methods, this is problematic: Samples cannot be re-used, each of the many outputs requires its own round of sampling. We can increase the efficiency of our localized smoothing approach by partitioning our $D_{\text{out}}$ outputs into $N_{\text{out}}$ subsets that share the same smoothing distribution. When making smoothed predictions or computing base certificates, we can then reuse the same samples for all outputs within each subsets.

More formally, we partition our $D_{\text{out}}$ output dimensions into sets $\mathbb{K}^{(1)}, \dots, \mathbb{K}^{(N_{\text{out}})}$ with

$$\dot{\bigcup}_{i=1}^{N_{\text{out}}} \mathbb{K}^{(i)} = \{1, \dots, D_{\text{out}}\}. \tag{6}$$

We then associate each set $\mathbb{K}^{(i)}$ with a smoothing distribution $\Psi^{(i)}$. For each base model output $g_n$ with $n \in \mathbb{K}^{(i)}$, we then use smoothing distribution $\Psi^{(i)}$ to construct the smoothed output $f_n$, e.g. $f_n(\boldsymbol{x}) = \operatorname{argmax}_{y \in \mathbb{Y}} \Pr_{\boldsymbol{z} \sim \Psi^{(i)}} [f(\boldsymbol{x} + \boldsymbol{z}) = y]$ (note that we use a different smoothing paradigm for binary data, see Section 5).

## A.2 Quantizing certificate parameters

Recall that our base certificates from Section 5 are defined by a linear inequality: A prediction $y_n = f_n(\boldsymbol{x})$ is robust to a perturbed input $\boldsymbol{x}' \in \mathbb{X}^{D_{\text{in}}}$ if $\sum_{d=1}^{D} w_d^{(n)} \cdot |x'_d - x_d|^p < \eta^{(n)}$, for some $p \geq 0$. The weight vectors $\boldsymbol{w}^{(n)} \in \mathbb{R}^{D_{\text{in}}}$ only depend on the smoothing distributions. A side of effect of sharing the same smoothing $\Psi^{(i)}$ among all outputs from a set $\mathbb{K}^{(i)}$, as discussed in the previous section, is that the outputs also share the same weight vector $\boldsymbol{w}^{(i)} \in \mathbb{R}^{D_{\text{in}}}$ with $\forall n \in \mathbb{K}^{(i)} : \boldsymbol{w}^{(i)} = \boldsymbol{w}^{(n)}$. Thus, for all smoothed outputs $f_n$ with $n \in \mathbb{K}^{(i)}$, the smoothed prediction $y_n$ is robust if $\sum_{d=1}^{D} w_d^{(i)} \cdot |x'_d - x_d|^p < \eta^{(n)}$.

Evidently, the base certificates for outputs from a set $\mathbb{K}^{(i)}$ only differ in their parameter $\eta^{(n)}$. Recall that in our collective linear program we use a vector of variables $\boldsymbol{t} \in \{0,1\}^{D_{\text{out}}}$ to indicate which predictions are robust according to their base certificates (see Section 6). If there are two outputs $f_n$ and $f_m$ with $\eta^{(n)} = \eta^{(m)}$, then $f_n$ and $f_m$ have the same base certificate and their robustness can be modelled by the same indicator variable. Conversely, for each set of outputs $\mathbb{K}^{(i)}$, we only need one indicator variable per unique $\eta^{(n)}$. By quantizing the $\eta^{(n)}$ within each subset $\mathbb{K}^{(i)}$ (for example by defining equally sized bins between $\min_{n \in \mathbb{K}^{(i)}} \eta^{(n)}$ and $\max_{n \in \mathbb{K}^{(i)}} \eta^{(n)}$ ), we can ensure that there is always a fixed number $N_{\text{bins}}$ of indicator variables per subset. This way, we can reduce the number of indicator variables from $D_{\text{out}}$ to $N_{\text{out}} \cdot N_{\text{bins}}$.

To implement this idea, we define matrix of thresholds $\boldsymbol{E} \in \mathbb{R}^{N_{\text{out}} \times N_{\text{bins}}}$ with $\forall i : \min \{\boldsymbol{E}_{i,:}\} \leq \min_{n \in \mathbb{K}^{(i)}} \left( \{\eta^{(n)} \mid n \in \mathbb{K}^{(i)}\} \right)$. We then define a function $\xi : \{1, \ldots, N_{\text{out}}\} \times \mathbb{R} \to \mathbb{R}$ with

$$\xi(i, \eta) = \max \left( \{E_{i,j} \mid j \in \{1, \ldots, N_{\text{bins}} \wedge E_{i,j} < \eta\} \right) \tag{7}$$

that quantizes base certificate parameter $\eta$ from output subset $\mathbb{K}^{(i)}$ by mapping it to the next smallest threshold in $\boldsymbol{E}_{i,:}$. For feasibility, like in Section 6 we need to compute the constant $\underline{\eta}^{(i)} = \min_{\boldsymbol{b} \in \mathbb{R}_+^{D_{\text{in}}}} \boldsymbol{b}^T \boldsymbol{w}_d^{(i)}$ s.t. $\text{sum}\{\boldsymbol{b}\} \leq \epsilon^p$ to ensure feasibility of the problem. Note that, because all outputs from a subset $\mathbb{K}^{(i)}$ share the same weight vector $\boldsymbol{w}^{(i)}$, we only have to compute this constant once per subset. We can bound the collective robustness of the targeted dimensions $\mathbb{T}$ of our vector of predictions $\boldsymbol{y} = f(\boldsymbol{x})$ as follows:

$$\min \sum_{i \in \{1, \ldots, N_{\text{out}}\}} \sum_{j \in \{1, \ldots, N_{\text{bins}}\}} T_{i,j} \left| \left\{ n \in \mathbb{T} \cap \mathbb{K}^{(i)} \middle| \xi\left(i, \eta^{(n)}\right) = E_{i,j} \right\} \right| \tag{8}$$

$$\text{s.t.} \quad \forall i, j : \boldsymbol{b}^T \boldsymbol{w}^{(i)} \geq T_{i,j} \underline{\eta}^{(i)} + (1 - T_{i,j}) E_{i,j}, \quad \text{sum}\{\boldsymbol{b}\} \leq \epsilon^p \tag{9}$$

$$\boldsymbol{b} \in \mathbb{R}_+^{D_{\text{in}}}, \quad \boldsymbol{T} \in \{0,1\}^{N_{\text{out}} \times N_{\text{bins}}}. \tag{10}$$

Constraint Eq. 9 ensures that $T_{i,j}$ is only set to 0 if $\boldsymbol{b}^T \boldsymbol{w}^{(i)} \geq E_{i,j}$, i.e. all predictions from subset $\mathbb{K}^{(i)}$ whose base certificate parameter $\eta^{(n)}$ is quantized to $E_{i,j}$ are no longer robust. When this is the case, the objective function decreases by the number of these predictions. For $N_{\text{out}} = D_{\text{out}}$, $N_{\text{bins}} = 1$ and $E_{n,1} = \eta^{(n)}$, we recover our general certificate from Section 6. Note that, if the quantization maps any parameter $\eta^{(n)}$ to a smaller number, the set $\mathbb{H}^{(n)}$ becomes more restrictive, i.e. $y_n$ is considered robust to a smaller set of perturbed inputs. Thus, Eq. 8 is a lower bound on our general certificate from Section 6.

## A.3 Sharing noise levels among inputs

Similar to how partitioning the output dimensions allows us to control the number of output variables $\boldsymbol{t}$, partitioning the input dimensions and using the same noise level within each partition allows us to control the number of variables $\boldsymbol{b}$ that model the allocation of adversarial budget.

Assume that we have partitioned our output dimensions into $N_{\text{out}}$ subsets $\mathbb{K}^{(1)}, \ldots, \mathbb{K}^{(N_{\text{out}}}$, with outputs in each subset sharing the same smoothing distribution $\Psi^{(i)}$, as explained in Section A.1.

Let us now define $N_{\text{in}}$ input subsets $\mathbb{J}^{(1)}, \ldots, \mathbb{J}^{(N_{\text{in}})}$ with

$$\dot{\bigcup}_{i=1}^{N_{\text{out}}} \mathbb{J}^{(i)} = \{1, \ldots, D_{\text{out}}\}. \tag{11}$$

Recall that a prediction $y_n = f_n(\boldsymbol{x})$ with $n \in \mathbb{K}^{(i)}$ is robust to a perturbed input $\boldsymbol{x}' \in \mathbb{X}^{D_{\text{in}}}$ if $\sum_{d=1}^{D} w_d^{(i)} \cdot |x_d' - x_d|^p < \eta^{(n)}$ and that the weight vectors $\boldsymbol{w}^{(i)}$ only depend on the smoothing distributions. Assume that we choose each smoothing distribution $\Psi^{(i)}$ such that $\forall l \in \{1, \ldots, N_{\text{in}}\}, \forall d, d' \in \mathbb{J}^{(l)} : w_d^{(i)} = w_{d'}^{(i)}$, i.e. all input dimensions within each set $\mathbb{J}^{(l)}$ have the same weight. This can be achieved by choosing $\Psi^{(i)}$ so that all dimensions in each input subset $\mathbb{J}^l$ are smoothed with the noise level (note that we can still use different $\Psi^{(i)}$, i.e. different noise levels for smoothing different sets of outputs). For example, one could use a Gaussian distribution with covariance matrix $\boldsymbol{\Sigma} = \text{diag}(\boldsymbol{\sigma})^2$ with $\forall l \in \{1, \ldots, N_{\text{in}}\}, \forall d, d' \in \mathbb{J}^{(l)} : \sigma_d = \sigma_{d'}$.

In this case, the evaluation of our base certificates can be simplified. Prediction $y_n = f_n(\boldsymbol{x})$ is robust to a perturbed input $\boldsymbol{x}' \in \mathbb{X}^{D_{\text{in}}}$ if

$$\sum_{d=1}^{D} w_d^{(i)} \cdot |x_d' - x_d|^p < \eta^{(n)} \tag{12}$$

$$= \sum_{l=1}^{N_{\text{in}}} \left( u^{(i)} \cdot \sum_{d \in \mathbb{J}^{(l)}} |x_d' - x_d|^p \right) < \eta^{(n)}, \tag{13}$$

with $\boldsymbol{u} \in \mathbb{R}^{N_{\text{in}}}$ and $\forall i \in \{1, \ldots, N_{\text{out}}\}, \forall l \in \{1, \ldots, N_{\text{in}}\}, \forall d \in \mathbb{J} : u_l^i = w_d^i$. That is, we can replace each weight vector $\boldsymbol{w}^{(i)}$ that has one weight $w_d^{(i)}$ per input dimension $d$ with a smaller weight vector $\boldsymbol{u}^{(i)}$ with one weight $u_l^{(i)}$ per input subset $\mathbb{J}^{(l)}$.

For our linear program, this means that we no longer need a budget vector $\boldsymbol{b} \in \mathbb{R}_+^{D_{\text{in}}}$ to model the element-wise distance $|x_d' - x_d|^p$ in each dimension $d$. Instead, we can use a smaller budget vector $\boldsymbol{b} \in \mathbb{R}_+^{N_{\text{in}}}$ to model the overall distance within each input subset $\mathbb{J}^{(l)}$, i.e. $\sum_{d \in \mathbb{J}^{(l)}} |x_d' - x_d|^p$. Combined with the quantization of certificate parameters from the previous section, our optimization problem becomes

$$\min \sum_{i \in \{1, \ldots, N_{\text{out}}\}} \sum_{j \in \{1, \ldots, N_{\text{bins}}\}} T_{i,j} \left| \left\{ n \in \mathbb{T} \cap \mathbb{K}^{(i)} \middle| \xi\left(i, \eta^{(n)}\right) = E_{i,j} \right\} \right| \tag{14}$$

$$\text{s.t.} \quad \forall i, j : \boldsymbol{b}^T \boldsymbol{u}^{(i)} \geq T_{i,j} \underline{\eta}^{(i)} + (1 - T_{i,j}) E_{i,j}, \quad \text{sum}\{\boldsymbol{b}\} \leq \epsilon^p, \tag{15}$$

$$\boldsymbol{b} \in \mathbb{R}_+^{N_{\text{in}}}, \quad \boldsymbol{T} \in \{0, 1\}^{N_{\text{out}} \times N_{\text{bins}}}. \tag{16}$$

with $\boldsymbol{u} \in \mathbb{R}^{N_{\text{in}}}$ and $\forall i \in \{1, \ldots, N_{\text{out}}\}, \forall l \in \{1, \ldots, N_{\text{in}}\}, \forall d \in \mathbb{J} : \omega_l^i = w_d^i$. For $N_{\text{out}} = D_{\text{out}}$, $N_{\text{in}} = D_{\text{in}}$, $N_{\text{bins}} = 1$ and $E_{n,1} = \eta^{(n)}$, we recover our general certificate from Section 6.

When certifying robustness for binary data, we impose different constraints on $\boldsymbol{b}$. To model that the adversary can not flip more bits than are present within each subset, we use a budget vector $\boldsymbol{b} \in \mathbb{N}_0^{N_{\text{in}}}$ with $\forall l \in \{1, \ldots, N_{\text{in}}\} : b_l \leq |\mathbb{J}^{(l)}|$, instead of a continuous budget vector $\boldsymbol{b} \in \mathbb{R}_+^{N_{\text{in}}}$.

### A.4 LINEAR RELAXATION

Combining the previous steps allows us to reduce the number of problem variables and linear constraints from $D_{\text{in}} + D_{\text{out}}$ and $D_{\text{out}} + 1$ to $N_{\text{in}} + N_{\text{out}} \cdot N_{\text{bins}}$ and $N_{\text{out}} \cdot N_{\text{bins}} + 1$, respectively. Still, finding an optimal solution to the mixed-integer linear program may be too expensive. One can obtain a lower bound on the optimal value and thus a valid, albeit more pessimistic, robustness certificate by relaxing all to be continuous.

When using the general certificate from Section 6, the binary vector $\boldsymbol{t} \in \{0, 1\}^{D_{\text{out}}}$ can be relaxed to $\boldsymbol{t} \in [0, 1]^{D_{\text{out}}}$. When using the certificate with quantized base certificate parameters from Section A.2 or Section A.3, the binary matrix $\boldsymbol{T} \in [0, 1]^{N_{\text{out}} \times N_{\text{bins}}}$ can be relaxed to $\boldsymbol{T} \in [0, 1]^{N_{\text{out}} \times N_{\text{bins}}}$. Conceptually, this means that predictions can be partially certified, i.e. $t_n \in (0, 1)$

or $T_{i,j} \in (0,1)$. In particular, a prediction can be partially certified even if we know that is impossible to attack under the collective perturbation model $\mathbb{B}_{\boldsymbol{x}} = \{ \boldsymbol{x}' \in \mathbb{X}^{D_{\text{in}}} \mid ||\boldsymbol{x}' - \boldsymbol{x}||_p \leq \epsilon \}$. Just like Schuchardt et al. (2021), who encountered the same problem with their collective certificate, we circumvent this issue by first computing a set $\mathbb{L} \subseteq \mathbb{T}$ of all targeted predictions in $\mathbb{T}$ that are guaranteed to always be robust:

$$\mathbb{L} = \left\{ n \in \mathbb{T} \left| \left( \max_{x \in \mathbb{B}_{\boldsymbol{x}}} \sum_{d=1}^{D} w_d^{(n)} \cdot |x_d' - x_d|^p \right) < \eta^{(n)} \right. \right\} \tag{17}$$

$$= \left\{ n \in \mathbb{T} \left| \max \left( \max \left\{ \boldsymbol{w}^{(n)} \right\} \cdot \epsilon^p, 0 \right) < \eta^{(n)} \right. \right\}. \tag{18}$$

The equality follows from the fact that the most effective way of attacking a prediction is to allocate all adversarial budget to the least robust dimension, i.e. the dimension with the largest weight – unless all weights are negative. Because we know that all predictions with indices in $\mathbb{L}$ are robust, we do not have to include them in the collective optimization problem and can instead compute

$$|\mathbb{L}| + \min_{\boldsymbol{x}' \in \mathbb{B}_{\boldsymbol{x}}} \sum_{n \in \mathbb{T} \setminus \mathbb{L}} \mathbb{I} \left[ \boldsymbol{x}' \in \mathbb{H}^{(n)} \right]. \tag{19}$$

The r.h.s. optimization can be solved using the general collective certificate from Section 6 or any of the more efficient, modified certificates from previous sections.

When using the general collective certificate from Section 6 with binary data, the budget variables $\boldsymbol{b} \in \{0,1\}^{D_{\text{in}}}$ can be relaxed to $\boldsymbol{b} \in [0,1]^{D_{\text{in}}}$. When using the modified collective certificate from Section A.3, the budget variables with $\boldsymbol{b} \in \mathbb{N}_0^{N_{\text{in}}}$ can be relaxed to $\boldsymbol{b} \in \mathbb{R}_+^{N_{\text{in}}}$. The additional constraint $\forall l \in \{1, \ldots, N_{\text{in}}\} : b_l \leq |\mathbb{J}^{(l)}|$ can be kept in order to model that the adversary cannot flip (or partially flip) more bits than are present within each input subset $\mathbb{J}^{(l)}$.

## B    BASE CERTIFICATES

In the following, we show why the base certificates presented in Section 5 hold and present alternatives for other collective perturbation models.

### B.1    GAUSSIAN SMOOTHING FOR $l_2$ PERTURBATIONS OF CONTINUOUS DATA

**Proposition 1.** *Given an output $g_n : \mathbb{R}^{D_{\text{in}}} \to \mathbb{Y}$, let $f_n(\boldsymbol{x}) = \operatorname{argmax}_{y \in \mathbb{Y}} \operatorname{Pr}_{\boldsymbol{z} \sim \mathcal{N}(\boldsymbol{x}, \boldsymbol{\Sigma})} [g_n(\boldsymbol{z}) = y]$ be the corresponding smoothed output with $\boldsymbol{\Sigma} = \operatorname{diag}(\boldsymbol{\sigma})^2$ and $\boldsymbol{\sigma} \in \mathbb{R}_+^{D_{\text{in}}}$. Given an input $\boldsymbol{x} \in \mathbb{R}^{D_{\text{in}}}$ and smoothed prediction $y_n = f_n(\boldsymbol{x})$, let $q = \operatorname{Pr}_{\boldsymbol{z} \sim \mathcal{N}(\boldsymbol{x}, \boldsymbol{\Sigma})} [g_n(\boldsymbol{z}) = y_n]$. Then, $\forall \boldsymbol{x}' \in \mathbb{H}^{(n)} :$ $f_n(\boldsymbol{x}') = y_n$ with $\mathbb{H}^{(n)}$ defined as in Eq. 2, $w_d = \frac{1}{\sigma_d{}^2}$, $\eta = \left( \Phi^{(-1)}(q) \right)^2$ and $\kappa = 2$.*

*Proof.* Based on the definition of the base certificate interface, we need to show that, $\forall \boldsymbol{x}' \in \mathbb{H} :$ $f_n(\boldsymbol{x}') = y_n$ with

$$\mathbb{H} = \left\{ \boldsymbol{x}' \in \mathbb{R}^{D_{\text{in}}} \left| \sum_{d=1}^{D_{\text{in}}} \frac{1}{\sigma_d^2} \cdot |x_d - x_d'|^2 < \left( \Phi^{-1}(q) \right)^2 \right. \right\}. \tag{20}$$

Eiras et al. (2021) have shown that under the same conditions as above, but with a general covariance matrix $\boldsymbol{\Sigma} \in \mathbb{R}_+^{D_{\text{in}} \times D_{\text{in}}}$, a prediction $y_n$ is certifiably robust to a perturbed input $\boldsymbol{x}'$ if

$$\sqrt{(\boldsymbol{x} - \boldsymbol{x}') \boldsymbol{\Sigma}^{-1} (\boldsymbol{x} - \boldsymbol{x}')} < \frac{1}{2} \left( \Phi^{-1}(q) - \Phi^{-1}(q') \right), \tag{21}$$

where $q' = \max_{y_n' \neq y_n} \operatorname{Pr}_{\boldsymbol{z} \sim \mathcal{N}(\boldsymbol{x}, \boldsymbol{\Sigma})} [g_n(\boldsymbol{z}) = y_n']$ is the probability of the second most likely prediction under the smoothing distribution. Because the probabilities of all possible predictions have to sum up to 1, we have $q' \leq 1 - q$. Since $\Phi^{-1}$ is monotonically increasing, we can obtain a lower bound on the r.h.s. of Eq. 21 and thus a more pessimistic certificate by substituting $1 - q$

for $q'$ (deriving such a "binary certificate" from a "multiclass certificate" is common in randomized smoothing and was already discussed in (Cohen et al., 2019)):

$$\sqrt{(\boldsymbol{x} - \boldsymbol{x}')\boldsymbol{\Sigma}^{-1}(\boldsymbol{x} - \boldsymbol{x}')} < \frac{1}{2}\left(\Phi^{-1}(q) - \Phi^{-1}(1 - q)\right), \tag{22}$$

In our case, $\boldsymbol{\Sigma}$ is a diagonal matrix $\operatorname{diag}(\boldsymbol{\sigma})^2$ with $\boldsymbol{\sigma} \in \mathbb{R}_+^{D_{\text{in}}}$. Thus Eq. 22 is equivalent to

$$\sqrt{\sum_{d=1}^{D_{\text{in}}}(x_d - x'_d)\frac{1}{\sigma_d^2}(x_d - x'_d)} < \frac{1}{2}\left(\Phi^{-1}(q) - \Phi^{-1}(1 - q)\right). \tag{23}$$

Finally, using the fact that $\Phi^{-1}(q) - \Phi^{-1}(1 - q) = 2\Phi^{-1}(q)$ and eliminating the square root shows that we are certifiably robust if

$$\sum_{d=1}^{D_{\text{in}}}\frac{1}{\sigma_d^2} \cdot |x_d - x'_d|^2 < \left(\Phi^{-1}(q)\right)^2. \tag{24}$$

$\square$

### B.1.1 UNIFORM SMOOTHING FOR $l_1$ PERTURBATIONS OF CONTINUOUS DATA

An alternative base certificate for $l_1$ perturbations is again due to Eiras et al. (2021). Using uniform instead of Gaussian noise later allows us to collective certify robustness to $l_1$-norm-bound perturbations. In the following $\mathcal{U}(\boldsymbol{x}, \boldsymbol{\lambda})$ with $\boldsymbol{x} \in \mathbb{R}^D$, $\boldsymbol{\lambda} \in \mathbb{R}_+^D$ refers to a vector-valued random distribution in which the $d$-th element is uniformly distributed in $[x_d - \lambda_d, x_d + \lambda_d]$.

**Proposition 2.** *Given an output $g_n : \mathbb{R}^{D_{\text{in}}} \to \mathbb{Y}$, let $f(\boldsymbol{x}) = \operatorname{argmax}_{y \in \mathbb{Y}} \operatorname{Pr}_{\boldsymbol{z} \sim \mathcal{U}(\boldsymbol{x}, \boldsymbol{\lambda})}[g(\boldsymbol{z}) = y]$ be the corresponding smoothed classifier with $\boldsymbol{\lambda} \in \mathbb{R}_+^{D_{\text{in}}}$. Given an input $\boldsymbol{x} \in \mathbb{R}^{D_{\text{in}}}$ and smoothed prediction $y = f(\boldsymbol{x})$, let $p = \operatorname{Pr}_{\boldsymbol{z} \sim \mathcal{U}(\boldsymbol{x}, \boldsymbol{\lambda})}[g(\boldsymbol{z}) = y]$. Then, $\forall \boldsymbol{x}' \in \mathbb{H}^{(n)} : f_n(\boldsymbol{x}') = y_n$ with $\mathbb{H}^{(n)}$ defined as in Eq. 2, $w_d = 1/\lambda_d$, $\eta = \Phi^{-1}(q)$ and $\kappa = 1$.*

*Proof.* Based on the definition of $\mathbb{H}^{(n)}$, we need to prove that $\forall \boldsymbol{x}' \in \mathbb{H} : f_n(\boldsymbol{x}') = y_n$ with

$$\mathbb{H} = \left\{ \boldsymbol{x}' \in \mathbb{R}^{D_{\text{in}}} \mid \sum_{d=1}^{D_{\text{in}}}\frac{1}{\lambda_d} \cdot |x_d - x'_d| < \Phi^{-1}(q) \right\}, \tag{25}$$

Eiras et al. (2021) have shown that under the same conditions as above, a prediction $y_n$ is certifiably robust to a perturbed input $\boldsymbol{x}'$ if

$$\sum_{d=1}^{D_{\text{in}}}|\frac{1}{\lambda_d} \cdot (x_d - x'_d)| < \frac{1}{2}\left(\Phi^{-1}(q) - \Phi^{-1}(1 - q)\right), \tag{26}$$

where $q' = \max_{y'_n \neq y_n} \operatorname{Pr}_{\boldsymbol{z} \sim \mathcal{U}(\boldsymbol{x}, \boldsymbol{\lambda})}[g_n(\boldsymbol{z}) = y'_n]$ is the probability of the second most likely prediction under the smoothing distribution. As in our previous proof for Gaussian smoothing, we can obtain a more pessimistic certificate by substituting $1 - q$ for $q'$. Since $\Phi^{-1}(q) - \Phi^{-1}(1 - q) = 2\Phi^{-1}(q)$ and all $\lambda_d$ are non-negative, we know that our prediction is certifiably robust if

$$\sum_{d=1}^{D_{\text{in}}}\frac{1}{\lambda_d} \cdot |x_d - x'_d| < \Phi^{-1}(p). \tag{27}$$

$\square$

### B.2 VARIANCE SMOOTHING

We propose variance smoothing as a base certificate for binary data. Variance smoothing certifies predictions based on the mean and variance of the softmax score associated with a predicted label. It is in principle applicable to arbitrary data types. We focus on discrete data, but all results can

be generalized from discrete to continuous data by replacing any sum over probability mass functions with integrals over probability density functions. We first derive a general form of variance smoothing before discussing our certificates for binary data in Section B.2.1 and Section B.2.2.

Variance smoothing assumes that we make predictions by randomly smoothing a base model's softmax scores. That is, given base model $g : \mathbb{X} \to \Delta_{|\mathbb{Y}|}$ mapping from an arbitrary discrete input space $\mathbb{X}$ to scores from the $|\mathbb{Y}|$-dimensional probability simplex $\Delta_{|\mathbb{Y}|}$, we define the smoothed classifier $f(\boldsymbol{x}) = \text{argmax}_{y \in \mathbb{Y}} \mathbb{E}_{\boldsymbol{z} \sim \Psi(\boldsymbol{x})} [g(\boldsymbol{z})_y]$. Here, $\Psi(\boldsymbol{x})$ is an arbitrary distribution over $\mathbb{X}$ parameterized by $\boldsymbol{x}$, e.g a Normal distribution with mean $\boldsymbol{x}$. The smoothed classifier does not return the most likely prediction, but the prediction associated with the highest expected softmax score.

Given an input $\boldsymbol{x} \in \mathbb{X}$, smoothed prediction $y = f(\boldsymbol{x})$ and a perturbed input $\boldsymbol{x}' \in \mathbb{X}$, we want to determine whether $f(\boldsymbol{x}') = y$. By definition of our smoothed classifier, we know that $f(\boldsymbol{x}') = y$ if $y$ is the label with the highest expected softmax score. In particular, we know that $f(\boldsymbol{x}') = y$ if $y$'s softmax score is larger than all other softmax scores combined, i.e.

$$\mathbb{E}_{\boldsymbol{z} \sim \Psi(\boldsymbol{x}')} [g(\boldsymbol{z})_y] > 0.5 \implies f(\boldsymbol{x}') = y. \tag{28}$$

Computing $\mathbb{E}_{\boldsymbol{z} \sim \Psi(\boldsymbol{x}')} [g(\boldsymbol{z})_y]$ exactly is usually not tractable – especially if we later want to evaluate robustness to many $\boldsymbol{x}'$ from a whole perturbation model $\mathbb{B} \subseteq \mathbb{X}$. Therefore, we compute a lower bound on $\mathbb{E}_{\boldsymbol{z} \sim \Psi(\boldsymbol{x}')} [g(\boldsymbol{z})_y]$. If even this lower bound is larger than $0.5$, we know that prediction $y$ is certainly robust. For this, we define a set of functions $\mathbb{H}$ with $g_y \in \mathbb{H}$ and compute the minimum softmax score across all functions from $\mathbb{H}$:

$$\min_{h \in \mathbb{H}} \mathbb{E}_{\boldsymbol{z} \sim \Psi(\boldsymbol{x}')} [h(\boldsymbol{z})] > 0.5 \implies f(\boldsymbol{x}') = y. \tag{29}$$

For our variance smoothing approach, we define $\mathbb{H}$ to be the set of all functions that have a larger or equal expected value and a smaller or equal variance, compared to our base model $g$ applied to unperturbed input $\boldsymbol{x}$. Let $\mu = \mathbb{E}_{\boldsymbol{z} \sim \Psi(\boldsymbol{x})} [g(\boldsymbol{z})_y]$ be the expected softmax score of our base model $g$ for label $y$. Let $\sigma^2 = \mathbb{E}_{\boldsymbol{z} \sim \Psi(\boldsymbol{x})} \left[ (g(\boldsymbol{z})_y - \nu)^2 \right]$ be the expected squared distance of the softmax score from a scalar $\nu \in \mathbb{R}$. (Choosing $\nu = \mu$ yields the variance of the softmax score. An arbitrary $\nu$ is only needed for technical reasons related to Monte Carlo estimation Section C.2). Then, we define

$$\mathbb{H} = \left\{ h : \mathbb{X} \to \mathbb{R} \,\middle|\, \mathbb{E}_{\boldsymbol{z} \sim \Psi(\boldsymbol{x})} [h(\boldsymbol{z})] \geq \mu \wedge \mathbb{E}_{\boldsymbol{z} \sim \Psi(\boldsymbol{x})} \left[ (h(\boldsymbol{z}) - \nu)^2 \right] \leq \sigma^2 \right\} \tag{30}$$

Clearly, by the definition of $\mu$ and $\sigma^2$, we have $g_y \in \mathbb{H}$. Note that we do not restrict functions from $\mathbb{H}$ to the domain $[0, 1]$, but allow arbitrary real-valued outputs.

By evaluating Eq. 28 with $\mathbb{H}$ defined as in Eq. 29, we can determine if our prediciton is robust. To compute the optimal value , we need the following two Lemmata:

**Lemma 1.** *Given a discrete set $\mathbb{X}$ and the set $\Pi$ of all probability mass functions over $\mathbb{X}$, any two probability mass functions $\pi_1$, $\pi_2 \in \Pi$ fulfill*

$$\sum_{z \in \mathbb{X}} \frac{\pi_2(z)}{\pi_1(z)} \pi_2(z) \geq 1. \tag{31}$$

*Proof.* For a fixed probability mass function $\pi_1$, Eq. 31 is lower-bounded by the minimal expected likelihood ratio that can be achieved by another $\tilde{\pi}(z) \in \Pi$:

$$\sum_{z \in \mathbb{X}} \frac{\pi_2(z)}{\pi_1(z)} \pi_2(z) \geq \min_{\tilde{\pi} \in \Pi} \sum_{z \in \mathbb{X}} \frac{\tilde{\pi}(z)}{\pi_1(z)} \tilde{\pi}(z). \tag{32}$$

The r.h.s. term can be expressed as the constrained optimization problem

$$\min_{\tilde{\pi}} \sum_{z \in \mathbb{X}} \frac{\tilde{\pi}(z)}{\pi_1(z)} \tilde{\pi}(z) \quad \text{s.t.} \quad \sum_{z \in \mathbb{X}} \tilde{\pi}(z) = 1 \tag{33}$$

with the corresponding dual problem

$$\max_{\lambda \in \mathbb{R}} \min_{\tilde{\pi}} \sum_{z \in \mathbb{X}} \frac{\tilde{\pi}(z)}{\pi_1(z)} \tilde{\pi}(z) + \lambda \left( -1 + \sum_{z \in \mathbb{X}} \tilde{\pi}(z) \right). \tag{34}$$

The inner problem is convex in each $\tilde{\pi}(z)$. Taking the gradient w.r.t. to $\tilde{\pi}(z)$ for all $z \in \mathbb{X}$ shows that it has its minimum at $\forall z \in \mathbb{X} : \tilde{\pi}(z) = -\frac{\lambda \pi_1(z)}{2}$. Substituting into Eq. 34 results in

$$\max_{\lambda \in \mathbb{R}} \sum_{z \in \mathbb{X}} \frac{\lambda^2 \pi_1(z)^2}{4\pi_1(z)} + \lambda \left( -1 - \sum_{z \in \mathbb{X}} \frac{\lambda \pi_1(z)}{2} \right) \tag{35}$$

$$= \max_{\lambda \in \mathbb{R}} -\lambda^2 \sum_{z \in \mathbb{X}} \frac{\pi_1(z)}{4} - \lambda \tag{36}$$

$$= \max_{\lambda \in \mathbb{R}} -\frac{\lambda^2}{4} - \lambda \tag{37}$$

$$= 1. \tag{38}$$

Eq. 37 follows from the fact that $\pi_1(z)$ is a valid probability mass function. Due to duality, the optimal dual value 1 is a lower bound on the optimal value of our primal problem Eq. 31. $\qquad \square$

**Lemma 2.** *Given a probability distribution $\mathcal{D}$ over a $\mathbb{R}$ and a scalar $\nu \in \mathbb{R}$, let $\mu = \mathbb{E}_{z \sim \mathcal{D}}$ and $\xi = \mathbb{E}_{z \sim \mathcal{D}} \left[ (z - \nu)^2 \right]$. Then $\xi \geq (\mu - \nu)^2$*

*Proof.* Using the definitions of $\mu$ and $\xi$, as well as some simple algebra, we can show:

$$\xi \geq (\mu - \nu)^2 \tag{39}$$

$$\iff \mathbb{E}_{z \sim \mathcal{D}} \left[ (z - \nu)^2 \right] \geq \mu^2 - 2\mu\nu + \nu^2 \tag{40}$$

$$\iff \mathbb{E}_{z \sim \mathcal{D}} \left[ z^2 - 2z\nu + \nu^2 \right] \geq \mu^2 - 2\mu\nu + \nu^2 \tag{41}$$

$$\iff \mathbb{E}_{z \sim \mathcal{D}} \left[ z^2 - 2z\nu + \nu^2 \right] \geq \mu^2 - 2\mu\nu + \nu^2 \tag{42}$$

$$\iff \mathbb{E}_{z \sim \mathcal{D}} \left[ z^2 \right] - 2\mu\nu + \nu^2 \geq \mu^2 - 2\mu\nu + \nu^2 \tag{43}$$

$$\iff \mathbb{E}_{z \sim \mathcal{D}} \left[ z^2 \right] \geq \mu^2 \tag{44}$$

It is well known for the variance that $\mathbb{E}_{z \sim \mathcal{D}} \left[ (z - \mu)^2 \right] = \mathbb{E}_{z \sim \mathcal{D}} \left[ z^2 \right] - \mu^2$. Because the variance is always non-negative, the above inequality holds. $\qquad \square$

Using the previously described approach and lemmata, we can show the soundness of the following robustness certificate:

**Theorem 3.** *Given a model $g : \mathbb{X} \to \Delta_{|\mathbb{Y}|}$ mapping from discrete set $\mathbb{X}$ to scores from the $|\mathbb{Y}|$-dimensional probability simplex, let $f(\boldsymbol{x}) = \arg\max_{y \in \mathbb{Y}} \mathbb{E}_{\boldsymbol{z} \sim \Psi(\boldsymbol{x})} [g(\boldsymbol{z})_y]$ be the corresponding smoothed classifier with smoothing distribution $\Psi(\boldsymbol{x})$ and probability mass function $\pi_{\boldsymbol{x}}(\boldsymbol{z}) = \Pr_{\tilde{\boldsymbol{z}} \sim \Psi(\boldsymbol{x})} [\tilde{\boldsymbol{z}} = \boldsymbol{z}]$. Given an input $\boldsymbol{x} \in \mathbb{X}$ and smoothed prediction $y = f(\boldsymbol{x})$, let $\mu = \mathbb{E}_{\boldsymbol{z} \sim \Psi(\boldsymbol{x})} [g(\boldsymbol{z})_y]$ and $\sigma^2 = \mathbb{E}_{\boldsymbol{z} \sim \Psi(\boldsymbol{x})} \left[ (g(\boldsymbol{z})_y - \nu)^2 \right]$ with $\nu \in \mathbb{R}$. If $\nu \leq \mu$, we know that $f(\boldsymbol{x}') = y$ if*

$$\sum_{\boldsymbol{z} \in \mathbb{X}} \frac{\pi_{\boldsymbol{x}'}(\boldsymbol{z})^2}{\pi_{\boldsymbol{x}}(\boldsymbol{z})} < 1 + \frac{1}{\sigma^2 - (\mu - \nu)^2} \left( \mu - \frac{1}{2} \right). \tag{45}$$

*Proof.* Following our discussion above, we know that $f(\boldsymbol{x}') = y$ if $\mathbb{E}_{\boldsymbol{z} \sim \Psi(\boldsymbol{x}')} [g(\boldsymbol{z})_y] > 0.5$ with $\mathbb{H}$ defined as in Section 5. We can compute a (tight) lower bound on $\min_{h \in \mathbb{H}} \mathbb{E}_{\boldsymbol{z} \sim \Psi(\boldsymbol{x}')}$ by following the functional optimization approach for randomized smoothing proposed by Zhang et al. (2020). That is, we solve a dual problem in which we optimize the value $h(\boldsymbol{z})$ for each $\boldsymbol{z} \in \mathbb{X}$. By the definition of the set $\mathbb{H}$, our optimization problem is

$$\min_{h:\mathbb{X} \to \mathbb{R}} \mathbb{E}_{\boldsymbol{z} \sim \Psi(\boldsymbol{x}')} [h(\boldsymbol{z})]$$

$$\text{s.t.} \quad \mathbb{E}_{\boldsymbol{z} \sim \Psi(\boldsymbol{x})} [h(\boldsymbol{z})] \geq \mu, \quad \mathbb{E}_{\boldsymbol{z} \sim \Psi(\boldsymbol{x})} \left[ (h(\boldsymbol{z}) - \nu)^2 \right] \leq \sigma^2.$$

The corresponding dual problem with dual variables $\alpha, \beta \geq 0$ is

$$\max_{\alpha, \beta \geq 0} \min_{h: \mathbb{X} \to \mathbb{R}} \mathbb{E}_{\boldsymbol{z} \sim \Psi(\boldsymbol{x}')} \left[ h(\boldsymbol{z}) \right]$$

$$+\alpha \left( \mu - \mathbb{E}_{\boldsymbol{z} \sim \Psi(\boldsymbol{x})} \left[ h(\boldsymbol{z}) \right] \right) + \beta \left( \mathbb{E}_{\boldsymbol{z} \sim \Psi(\boldsymbol{x})} \left[ (h(\boldsymbol{z}) - \nu)^2 \right] - \sigma^2 \right). \tag{46}$$

We first move move all terms that don't involve $h$ out of the inner optimization problem:

$$= \max_{\alpha, \beta \geq 0} \alpha \mu - \beta \sigma^2 + \min_{h: \mathbb{X} \to \mathbb{R}} \mathbb{E}_{\boldsymbol{z} \sim \Psi(\boldsymbol{x}')} \left[ h(\boldsymbol{z}) \right] - \alpha \mathbb{E}_{\boldsymbol{z} \sim \Psi(\boldsymbol{x})} \left[ h(\boldsymbol{z}) \right] + \beta \mathbb{E}_{\boldsymbol{z} \sim \Psi(\boldsymbol{x})} \left[ (h(\boldsymbol{z}) - \nu)^2 \right] \tag{47}$$

Writing out the expectation terms and combining them into one sum (or – in the case of continuous $\mathbb{X}$ – one integral), our dual problem becomes

$$= \max_{\alpha, \beta \geq 0} \alpha \mu - \beta \sigma^2 + \min_{h: \mathbb{X} \to \mathbb{R}} \sum_{\boldsymbol{z} \in \mathbb{X}} h(\boldsymbol{z}) \pi_{\boldsymbol{x}'}(\boldsymbol{z}) - \alpha h(\boldsymbol{z}) \pi_{\boldsymbol{x}}(\boldsymbol{z}) + \beta \left( h(\boldsymbol{z}) - \nu \right)^2 \pi_{\boldsymbol{x}}(\boldsymbol{z}) \tag{48}$$

(recall that $\pi_{\boldsymbol{x}'}$ and $\pi_{\boldsymbol{x}'}$ refer to the probability mass functions of the smoothing distributions). The inner optimization problem can be solved by finding the optimal $h(\boldsymbol{z})$ in each point $\boldsymbol{z}$:

$$= \max_{\alpha, \beta \geq 0} \alpha \mu - \beta \sigma^2 + \sum_{\boldsymbol{z} \in \mathbb{X}} \min_{h(\boldsymbol{z}) \in \mathbb{R}} h(\boldsymbol{z}) \pi_{\boldsymbol{x}'}(\boldsymbol{z}) - \alpha h(\boldsymbol{z}) \pi_{\boldsymbol{x}}(\boldsymbol{z}) + \beta \left( h(\boldsymbol{z}) - \nu \right)^2 \pi_{\boldsymbol{x}}(\boldsymbol{z}) \tag{49}$$

Because $\beta \geq 0$, each inner optimization problem is convex in $h(\boldsymbol{z})$. We can thus find the optimal $h^*(\boldsymbol{z})$ by setting the derivative to zero:

$$\frac{d}{dh(\boldsymbol{z})} h(\boldsymbol{z}) \pi_{\boldsymbol{x}'}(\boldsymbol{z}) - \alpha h(\boldsymbol{z}) \pi_{\boldsymbol{x}}(\boldsymbol{z}) + \beta \left( h(\boldsymbol{z}) - \nu \right)^2 \pi_{\boldsymbol{x}}(\boldsymbol{z}) \overset{!}{=} 0 \tag{50}$$

$$\iff \pi_{\boldsymbol{x}'}(\boldsymbol{z}) - \alpha \pi_{\boldsymbol{x}}(\boldsymbol{z}) + 2\beta \left( h(\boldsymbol{z}) - \nu \right) \pi_{\boldsymbol{x}}(\boldsymbol{z}) \overset{!}{=} 0 \tag{51}$$

$$\implies h^*(\boldsymbol{z}) = -\frac{\pi_{\boldsymbol{x}'}(\boldsymbol{z})}{2\beta \pi_{\boldsymbol{x}}(\boldsymbol{z})} + \frac{\alpha}{2\beta} + \nu. \tag{52}$$

Substituting into Eq. 48 and simplifying leaves us with the dual problem

$$\max_{\alpha, \beta \geq 0} \alpha \mu - \beta \sigma^2 - \frac{\alpha^2}{4\beta} + \frac{\alpha}{2\beta} - \alpha \nu + \nu - \frac{1}{4\beta} \sum_{\boldsymbol{z} \in \mathbb{X}} \frac{\pi_{\boldsymbol{x}'}(\boldsymbol{z})^2}{\pi_{\boldsymbol{x}}(\boldsymbol{z})} \tag{53}$$

In the following, let us use $\rho = \sum_{\boldsymbol{z} \in \mathbb{X}} \frac{\pi_{\boldsymbol{x}'}(\boldsymbol{z})^2}{\pi_{\boldsymbol{x}}(\boldsymbol{z})}$ as a shorthand for the expected likelihood ratio. The problem is concave in $\alpha$. We can thus find the optimum $\alpha^*$ by setting the derivative to zero, which gives us $\alpha^* = 2\beta(\mu - \nu) + 1$. Because $\beta \geq 0$ and ou theorem assumes that $\nu \leq \mu$, $\alpha^*$ is a feasible solution to the dual problem. Substituting into Eq. 53 and simplifying results in

$$\max_{\beta \geq 0} \alpha^* \mu - \beta \sigma^2 - \frac{\alpha^{*2}}{4\beta} + \frac{\alpha^*}{2\beta} - \alpha^* \nu + \nu - \frac{1}{4\beta} \rho \tag{54}$$

$$= \max_{\beta \geq 0} \beta \left( (\mu - \nu)^2 - \sigma^2 \right) + \mu + \frac{1}{4\beta} \left( 1 - \rho \right). \tag{55}$$

Lemma 1 shows that the expected likelihood ratio $\rho$ is always greater than or equal to 1. Lemma 2 shows that $(\mu - \nu)^2 - \sigma^2 \leq 0$. Therefore Eq. 55 is concave in $\beta$. The optimal value of $\beta$ can again be found by setting the derivative to zero:

$$\beta^* = \sqrt{\frac{1 - \rho}{4 \left( (\mu - \nu)^2 - \sigma^2 \right)}}. \tag{56}$$

Recall that our theorem assumes $\sigma^2 \geq (\mu - \nu)^2$ and thus $\beta^*$ is real valued. Substituting into Eq. 56 shows that the maximum of our dual problem is

$$\mu + \sqrt{(1 - p) \left( (\mu - \nu)^2 - \sigma^2 \right)}. \tag{57}$$

By duality, this is a lower bound on our primal problem $\min_{h \in \mathbb{H}} \mathbb{E}_{\boldsymbol{z} \sim \Psi(\boldsymbol{x}')}[h(\boldsymbol{z})]$. We know that our prediction is certifiably robust, i.e. $f(\boldsymbol{x}) = y$, if $\min_{h \in \mathbb{H}} \mathbb{E}_{\boldsymbol{z} \sim \Psi(\boldsymbol{x}')}[h(\boldsymbol{z})] > 0.5$. So, in particular, our prediction is robust if

$$\mu + \sqrt{(1-\rho)\left((\mu - \nu)^2 - \sigma^2\right)} > 0.5 \tag{58}$$

$$\Longleftrightarrow \rho < 1 + \frac{1}{\sigma^2 - (\mu - \nu)^2}\left(\mu - \frac{1}{2}\right)^2 \tag{59}$$

$$\Longleftrightarrow \sum_{\boldsymbol{z} \in \mathbb{X}} \frac{\pi_{\boldsymbol{x}'}(\boldsymbol{z})^2}{\pi_{\boldsymbol{x}}(\boldsymbol{z})} < 1 + \frac{1}{\sigma^2 - (\mu - \nu)^2}\left(\mu - \frac{1}{2}\right)^2 \tag{60}$$

The last equivalence is the result of inserting the definition of the expected likelihood ratio $\rho$. $\qquad \square$

With Theorem 3 in place, we can certify robustness for arbitrary smoothing distributions, assuming we can compute the expected likelihood ratio. When we are working with discrete data and the smoothing distributions factorize (but are not necessarily i.i.d.), this can be done efficiently, as the two following base certificates for binary data demonstrate.

### B.2.1 BERNOULLI VARIANCE SMOOTHING FOR PERTURBATIONS OF BINARY DATA

We begin by proving the base certificate presented in Section 5. Recall that we we use a smoothing distribution $\mathcal{F}(\boldsymbol{x}, \boldsymbol{\theta})$ with $\boldsymbol{\theta} \in [0, 1]^{D_{\text{in}}}$ that independently flips the $d$'th bit with probability $\theta_d$, i.e. for $\boldsymbol{x}, \boldsymbol{z} \in \{0, 1\}^{D_{\text{in}}}$ and $\boldsymbol{z} \sim \mathcal{F}(\boldsymbol{x}, \boldsymbol{\theta})$ we have $\Pr[z_d \neq x_d] = \theta_d$.

**Theorem 1.** *Given an output $g_n : \{0, 1\}^{D_{\text{in}}} \to \Delta_{|\mathbb{Y}|}$ mapping to scores from the $|\mathbb{Y}|$-dimensional probability simplex, let $f_n(\boldsymbol{x}) = \operatorname{argmax}_{y \in \mathbb{Y}} \mathbb{E}_{\boldsymbol{z} \sim \mathcal{F}(\boldsymbol{x}, \boldsymbol{\theta})}[g_n(\boldsymbol{z})_y]$ be the corresponding smoothed classifier with $\boldsymbol{\theta} \in [0, 1]^{D_{\text{in}}}$. Given an input $\boldsymbol{x} \in \{0, 1\}^{D_{\text{in}}}$ and smoothed prediction $y_n = f_n(\boldsymbol{x})$, let $\mu = \mathbb{E}_{\boldsymbol{z} \sim \mathcal{F}(\boldsymbol{x}, \boldsymbol{\theta})}[g_n(\boldsymbol{z})_y]$ and $\sigma^2 = \operatorname{Var}_{\boldsymbol{z} \sim \mathcal{F}(\boldsymbol{x}, \boldsymbol{\theta})}[g_n(\boldsymbol{z})_y]$. Then, $\forall \boldsymbol{x}' \in \mathbb{H}^{(n)} : f_n(\boldsymbol{x}') = y_n$ with $\mathbb{H}^{(n)}$ defined as in Eq. 2, $w_d = \ln\left(\frac{(1-\theta_d)^2}{\theta_d} + \frac{(\theta_d)^2}{1-\theta_d}\right)$, $\eta = \ln\left(1 + \frac{1}{\sigma^2}\left(\mu - \frac{1}{2}\right)^2\right)$ and $\kappa = 0$.*

*Proof.* Based on our definition of the base certificate interface from Section 5, we must show that $\forall \boldsymbol{x}' \in \mathbb{H} : f_n(\boldsymbol{x}') = y_n$ with

$$\mathbb{H} = \left\{\boldsymbol{x}' \in \{0, 1\}^{D_{\text{in}}} \;\middle|\; \sum_{d=1}^{D_{\text{in}}} \ln\left(\frac{(1-\theta_d)^2}{\theta_d} + \frac{(\theta_d)^2}{1-\theta_d}\right) \cdot |x_d' - x_d|^0 < \ln\left(1 + \frac{1}{\sigma^2}\left(\mu - \frac{1}{2}\right)^2\right)\right\}, \tag{61}$$

Because all bits are flipped independently, our probability mass function $\pi_{\boldsymbol{x}}(\boldsymbol{z}) = \Pr_{\tilde{\boldsymbol{z}} \sim \Psi(\boldsymbol{x})}[\tilde{\boldsymbol{z}} = \boldsymbol{z}]$ factorizes:

$$\pi_{\boldsymbol{x}}(\boldsymbol{z}) = \prod_{d=1}^{D_{\text{in}}} \pi_{x_d}(z_d) \tag{62}$$

with

$$\pi_{x_d}(z_d) = \begin{cases} \theta_d & \text{if } z_d \neq x_d \\ 1 - \theta_d & \text{else} \end{cases}. \tag{63}$$

Thus, our expected likelihood ratio can be written as

$$\sum_{\boldsymbol{z} \in \{0,1\}^{D_{\text{in}}}} \frac{\pi_{\boldsymbol{x}'}(\boldsymbol{z})^2}{\pi_{\boldsymbol{x}}(\boldsymbol{z})} = \sum_{\boldsymbol{z} \in \{0,1\}^{D_{\text{in}}}} \prod_{d=1}^{D_{\text{in}}} \frac{\pi_{x_d'}(z_d)^2}{\pi_{x_d}(z_d)} = \prod_{d=1}^{D_{\text{in}}} \sum_{z_d \in \{0,1\}} \frac{\pi_{x_d'}(z_d)^2}{\pi_{x_d}(z_d)}. \tag{64}$$

For each dimension $d$, we can distinguish two cases: If both the perturbed and unperturbed input are the same in dimension $d$, i.e. $x_d' = x_d$, then $\frac{\pi_{x_d'}(\boldsymbol{z})}{\pi_{x_d}(\boldsymbol{z})} = 1$ and thus

$$\sum_{z_d \in \{0,1\}} \frac{\pi_{x_d'}(z_d)^2}{\pi_{x_d}(z_d)} = \sum_{z_d \in \{0,1\}} \pi_{x_d'}(z_d) = \theta_d + (1 - \theta_d) = 1. \tag{65}$$

If the perturbed and unperturbed input differ in dimension $d$, then

$$\sum_{z_d \in \{0,1\}} \frac{\pi_{x'_d}(z_d)^2}{\pi_{x_d}(z_d)} = \frac{(1-\theta_d)^2}{\theta_d} + \frac{(\theta_d)^2}{1-\theta_d}. \tag{66}$$

Therefore, the expected likelihood ratio is

$$\prod_{d=1}^{D_{\text{in}}} \sum_{z_d \in \{0,1\}} \frac{\pi_{x'_d}(z_d)^2}{\pi_{x_d}(z_d)} = \prod_{d=1}^{D_{\text{in}}} \left( \frac{(1-\theta_d)^2}{\theta_d} + \frac{(\theta_d)^2}{1-\theta_d} \right)^{|x'_d - x_d|}. \tag{67}$$

Due to Theorem 3 (and using $\nu = \mu$ when computing the variance), we know that our prediction is robust, i.e. $f_n(\boldsymbol{x}') = y_n$, if

$$\sum_{\boldsymbol{z} \in \{0,1\}^{D_{\text{in}}}} \frac{\pi_{\boldsymbol{x}'}(\boldsymbol{z})^2}{\pi_{\boldsymbol{x}}(\boldsymbol{z})} < 1 + \frac{1}{\sigma^2} \left( \mu - \frac{1}{2} \right)^2 \tag{68}$$

$$\iff \prod_{d=1}^{D_{\text{in}}} \left( \frac{(1-\theta_d)^2}{\theta_d} + \frac{(\theta_d)^2}{1-\theta_d} \right)^{|x'_d - x_d|} < 1 + \frac{1}{\sigma^2} \left( \mu - \frac{1}{2} \right)^2 \tag{69}$$

$$\iff \sum_{d=1}^{D_{\text{in}}} \ln \left( \frac{(1-\theta_d)^2}{\theta_d} + \frac{(\theta_d)^2}{1-\theta_d} \right) |x'_d - x_d| < \ln \left( 1 + \frac{1}{\sigma^2} \left( \mu - \frac{1}{2} \right)^2 \right). \tag{70}$$

Because $x_d$ and $x'_d$ are binary, the last inequality is equivalent to

$$\sum_{d=1}^{D_{\text{in}}} \ln \left( \frac{(1-\theta_d)^2}{\theta_d} + \frac{(\theta_d)^2}{1-\theta_d} \right) |x'_d - x_d|^0 < \ln \left( 1 + \frac{1}{\sigma^2} \left( \mu - \frac{1}{2} \right)^2 \right). \tag{71}$$

$\square$

### B.2.2 SPARSITY-AWARE VARIANCE SMOOTHING FOR PERTURBATIONS OF BINARY DATA

Sparsity-aware randomized smoothing (Bojchevski et al., 2020) is an alternative smoothing approach for binary data. It uses different probabilities for randomly deleting ($1 \rightarrow 0$) and adding ($0 \rightarrow 1$) bits to preserve data sparsity. For a random variable $\boldsymbol{z}$ distributed according to the sparsity-aware distribution $\mathcal{S}(\boldsymbol{x}, \boldsymbol{\theta}^+, \boldsymbol{\theta}^-)$ with $\boldsymbol{x} \in \{0,1\}^{D_{\text{in}}}$ and addition and deletion probabilities $\boldsymbol{\theta}^+, \boldsymbol{\theta}^- \in [0,1]^{D_{\text{in}}}$, we have:

$$\Pr[z_d = 0] = \left( 1 - \theta_d^+ \right)^{1-x_d} \cdot \left( \theta_d^- \right)^{x_d},$$
$$\Pr[z_d = 1] = \left( \theta_d^+ \right)^{1-x_d} \cdot \left( 1 - \theta_d^- \right)^{x_d}.$$

The Bernoulli smoothing distribution we discussed in the previous section is a special case of sparsity-aware smoothing with $\boldsymbol{\theta}^+ = \boldsymbol{\theta}^-$. The runtime of the robustness certificate derived by Bojchevski et al. (2020) increases exponentially with the number of unique values in $\boldsymbol{\theta}^+$ and $\boldsymbol{\theta}^-$, which makes it unsuitable for localized smoothing. Variance smoothing, on the other hand, allows us to efficiently compute a certificate in closed form.

**Theorem 2.** *Given an output $g_n : \mathbb{R}^{D_{\text{in}}} \to \Delta_{|\mathbb{Y}|}$ mapping to scores from the $|\mathbb{Y}|$-dimensional probability simplex, let $f_n(\boldsymbol{x}) = \text{argmax}_{y \in \mathbb{Y}} \mathbb{E}_{\boldsymbol{z} \sim \mathcal{S}(\boldsymbol{x}, \boldsymbol{\theta}^+, \boldsymbol{\theta}^-)} [g_n(\boldsymbol{z})_y]$ be the corresponding smoothed classifier with $\boldsymbol{\theta}^+, \boldsymbol{\theta}^- \in [0,1]^{D_{\text{in}}}$. Given an input $\boldsymbol{x} \in \{0,1\}^{D_{\text{in}}}$ and smoothed prediction $y_n = f_n(\boldsymbol{x})$, let $\mu = \mathbb{E}_{\boldsymbol{z} \sim \mathcal{S}(\boldsymbol{x}, \boldsymbol{\theta}^+, \boldsymbol{\theta}^-)} [g_n(\boldsymbol{z})_y]$ and $\sigma^2 = \text{Var}_{\boldsymbol{z} \sim \mathcal{S}(\boldsymbol{x}, \boldsymbol{\theta}^+, \boldsymbol{\theta}^-)} [g_n(\boldsymbol{z})_y]$. Then, $\forall \boldsymbol{x}' \in \mathbb{H} : f_n(\boldsymbol{x}') = y$ for*

$$\mathbb{H} = \left\{ \boldsymbol{x}' \in \{0,1\}^{D_{\text{in}}} \mid \sum_{d=1}^{D_{\text{in}}} \gamma_d^+ \cdot \mathrm{I}\left[x_d = 0 \neq x'_d\right] + \gamma_d^- \cdot \mathrm{I}\left[x_d = 1 \neq x'_d\right] < \eta \right\}, \tag{72}$$

*where $\boldsymbol{\gamma}^+, \boldsymbol{\gamma}^- \in \mathbb{R}^{D_{\text{in}}}$, $\gamma_d^+ = \ln \left( \frac{(\theta_d^-)^2}{1-\theta_d^+} + \frac{(1-\theta_d^-)^2}{\theta_d^+} \right)$, $\gamma_d^- = \ln \left( \frac{(1-\theta_d^+)^2}{\theta_d^-} + \frac{(\theta_d^+)^2}{1-\theta_d^-}. \right)$ and $\eta = \ln \left( 1 + \frac{1}{\sigma^2} \left( \mu - \frac{1}{2} \right)^2 \right)$.*

*Proof.* Just like with the Bernoulli distribution we discussed in the previous section, all bits are flipped independently, meaning our probability mass function $\pi_{\boldsymbol{x}}(\boldsymbol{z}) = \Pr_{\tilde{\boldsymbol{z}} \sim \Psi(\boldsymbol{x})}[\tilde{\boldsymbol{z}} = \boldsymbol{z}]$ factorizes:

$$\pi_{\boldsymbol{x}}(\boldsymbol{z}) = \prod_{d=1}^{D_{\text{in}}} \pi_{x_d}(z_d) \tag{73}$$

with

$$\pi_{x_d}(z_d) = \begin{cases} \theta_d & \text{if } z_d \neq x_d \\ 1 - \theta_d & \text{else} \end{cases}. \tag{74}$$

As before, our expected likelihood ratio can be written as

$$\sum_{\boldsymbol{z} \in \{0,1\}^{D_{\text{in}}}} \frac{\pi_{\boldsymbol{x}'}(\boldsymbol{z})^2}{\pi_{\boldsymbol{x}}(\boldsymbol{z})} = \sum_{\boldsymbol{z} \in \{0,1\}^{D_{\text{in}}}} \prod_{d=1}^{D_{\text{in}}} \frac{\pi_{x'_d}(z_d)^2}{\pi_{x_d}(z_d)} = \prod_{d=1}^{D_{\text{in}}} \sum_{z_d \in \{0,1\}} \frac{\pi_{x'_d}(z_d)^2}{\pi_{x_d}(z_d)}. \tag{75}$$

We can now distinguish three cases. If both the perturbed and unperturbed input are the same in dimension $d$, i.e. $x'_d = x_d$, then $\frac{\pi_{x'_d}(\boldsymbol{z})}{\pi_{x_d}(\boldsymbol{z})} = 1$ and thus

$$\sum_{z_d \in \{0,1\}} \frac{\pi_{x'_d}(z_d)^2}{\pi_{x_d}(z_d)} = \sum_{z_d \in \{0,1\}} \pi_{x'_d}(z_d) = 1. \tag{76}$$

If $x'_d = 1$ and $x_d = 0$, i.e. a bit was added, then

$$\sum_{z_d \in \{0,1\}} \frac{\pi_{x'_d}(\boldsymbol{z})^2}{\pi_{x_d}(\boldsymbol{z})} = \sum_{z_d \in \{0,1\}} \frac{\pi_1(z_d)^2}{\pi_0(z_d)} = \frac{\pi_1(0)^2}{\pi_0(0)} + \frac{\pi_1(1)^2}{\pi_0(1)} = \frac{\left(\theta_d^-\right)^2}{1 - \theta_d^+} + \frac{\left(1 - \theta_d^-\right)^2}{\theta_d^+} \tag{77}$$

If $x'_d = 0$ and $x_d = 1$, i.e. a bit was deleted, then

$$\sum_{z_d \in \{0,1\}} \frac{\pi_{x'_d}(\boldsymbol{z})^2}{\pi_{x_d}(\boldsymbol{z})} = \sum_{z_d \in \{0,1\}} \frac{\pi_0(z_d)^2}{\pi_1(z_d)} = \frac{\pi_0(0)^2}{\pi_1(0)} + \frac{\pi_0(1)^2}{\pi_1(1)} = \frac{\left(1 - \theta_d^+\right)^2}{\theta_d^-} + \frac{\left(\theta_d^+\right)^2}{1 - \theta_d^-}. \tag{78}$$

Therefore, the expected likelihood ratio is

$$\prod_{d=1}^{D_{\text{in}}} \sum_{z_d \in \{0,1\}} \frac{\pi_{x'_d}(z_d)^2}{\pi_{x_d}(z_d)} \tag{79}$$

$$= \prod_{d=1}^{D_{\text{in}}} \left( \frac{\left(\theta_d^-\right)^2}{1 - \theta_d^+} + \frac{\left(1 - \theta_d^-\right)^2}{\theta_d^+} \right)^{\text{I}\left[x_d = 0 \neq x'_d\right]} \left( \frac{\left(1 - \theta_d^+\right)^2}{\theta_d^-} + \frac{\left(\theta_d^+\right)^2}{1 - \theta_d^-} \right)^{\text{I}\left[x_d = 1 \neq x'_d\right]} \tag{80}$$

$$= \prod_{d=1}^{D_{\text{in}}} \exp\left(\gamma_d^+\right)^{\text{I}\left[x_d = 0 \neq x'_d\right]} \cdot \exp\left(\gamma_d^-\right)^{\text{I}\left[x_d = 1 \neq x'_d\right]}. \tag{81}$$

In the last equation, we have simple used the shorthands $\gamma_d^+$ and $\gamma_d^-$, as defined in Theorem 2 Due to Theorem 3 (and using $\nu = \mu$ when computing the variance), we know that our prediction is robust, i.e. $f_n(\boldsymbol{x}') = y_n$, if

$$\sum_{\boldsymbol{z} \in \{0,1\}^{D_{\text{in}}}} \frac{\pi_{\boldsymbol{x}'}(\boldsymbol{z})^2}{\pi_{\boldsymbol{x}}(\boldsymbol{z})} < 1 + \frac{1}{\sigma^2} \left( \mu - \frac{1}{2} \right)^2 \tag{82}$$

$$\iff \prod_{d=1}^{D_{\text{in}}} \exp\left(\gamma_d^+\right)^{\text{I}\left[x_d = 0 \neq x'_d\right]} \cdot \exp\left(\gamma_d^-\right)^{\text{I}\left[x_d = 1 \neq x'_d\right]} < 1 + \frac{1}{\sigma^2} \left( \mu - \frac{1}{2} \right)^2 \tag{83}$$

$$\iff \sum_{d=1}^{D_{\text{in}}} \gamma_d^+ \cdot \text{I}\left[x_d = 0 \neq x'_d\right] \cdot \gamma_d^- \cdot \text{I}\left[x_d = 1 \neq x'_d\right] < \ln\left( 1 + \frac{1}{\sigma^2} \left( \mu - \frac{1}{2} \right)^2 \right). \tag{84}$$

$\square$

It should be noted that this certificate does not comply with our interface for base certificates Section 5, meaning we can not directly use it to certify robustness to norm-bound perturbations using our collective linear program from Section 6. We can however use it to certify collective robustness to the more refined threat model used in (Schuchardt et al., 2021): Let the set of admissible perturbed inputs be $\mathbb{B}_{\boldsymbol{x}} = \left\{ \boldsymbol{x}' \in \{0,1\}^{D_{\text{in}}} \mid \sum_{d=1}^{D_{\text{in}}} [x_d = 0 \neq x'_d] \leq \epsilon^+ \land \sum_{d=1}^{D_{\text{in}}} [x_d = 1 \neq x'_d] \leq \epsilon^- \right\}$ with $\epsilon^+, \epsilon^y \in \mathbb{N}_0$ specifying the number of bits the adversary is allowed to add or delete. We can now follow the procedure outlined in Section 6 to combine the per-prediction base certificates into a collective certificate for our new collective perturbation model. As discussed in, we can bound the number of predictions that are robust to simultaneous attacks by minimizing the number of predictions that are certifiably robust according to their base certificates:

$$\min_{\boldsymbol{x}' \in \mathbb{B}_{\boldsymbol{x}}} \sum_{n \in \mathbb{T}} \mathrm{I}\left[ f_n(\boldsymbol{x}') = y_n \right] \geq \min_{\boldsymbol{x}' \in \mathbb{B}_{\boldsymbol{x}}} \sum_{n \in \mathbb{T}} \mathrm{I}\left[ \boldsymbol{x}' \in \mathbb{H}^{(n)} \right]. \tag{85}$$

Inserting the linear inequalities characterizing our perturbation model and base certificates results in:

$$\min_{\boldsymbol{x}' \in \{0,1\}^{D_{\text{in}}}} \sum_{n \in \mathbb{T}} \mathrm{I}\left[ \sum_{d=1}^{D_{\text{in}}} \gamma_d^+ \cdot \mathrm{I}\left[ x_d = 0 \neq x'_d \right] + \gamma_d^- \cdot \mathrm{I}\left[ x_d = 1 \neq x'_d \right] < \eta^{(n)} \right] \tag{86}$$

$$\text{s.t.} \quad \sum_{d=1}^{D_{\text{in}}} [x_d = 0 \neq x'_d] \leq \epsilon^+, \quad \sum_{d=1}^{D_{\text{in}}} [x_d = 1 \neq x'_d] \leq \epsilon^-. \tag{87}$$

Instead of optimizing over the perturbed input $\boldsymbol{x}'$, we can define two vectors $\boldsymbol{b}^+, \boldsymbol{b}- \in \{0,1\}^{D_{\text{in}}}$ that indicate in which dimension bits were added or deleted. Using these new variables, Eq. 86 can be rewritten as

$$\min_{\boldsymbol{b}^+, \boldsymbol{b}^- \in \{0,1\}^{D_{\text{in}}}} \sum_{n \in \mathbb{T}} \mathrm{I}\left[ \left( \boldsymbol{\gamma}^+ \right)^T \boldsymbol{b}^+ + \left( \boldsymbol{\gamma}^- \right)^T \boldsymbol{b}^- < \eta^{(n)} \right] \tag{88}$$

$$\text{s.t.} \quad \text{sum}\{\boldsymbol{b}^+\} \leq \epsilon^+, \quad \text{sum}\{\boldsymbol{b}^-\} \leq \epsilon^-, \tag{89}$$

$$\sum_{d | x_d = 1} b_d^+ = 0, \quad \sum_{d | x_d = 0} b_d^- = 0. \tag{90}$$

The last two constraints ensure that bits can only be deleted where $x_d = 1$ and bits can only be added where $x_d = 0$. Finally, we can use the procedure for replacing the indicator functions with indicator variables that we discussed in Section 6). Let $\underline{\eta}^{(n)}$ be the minimum values the weighted sums in the objective function can take on:

$$\underline{\eta}^{(n)} = \min_{\boldsymbol{b}^+, \boldsymbol{b}^- \in \{0,1\}^{D_{\text{in}}}} \left( \boldsymbol{\gamma}^+ \right)^T \boldsymbol{b}^+ + \left( \boldsymbol{\gamma}^- \right)^T \boldsymbol{b}^- \tag{91}$$

$$\text{s.t.} \quad \text{sum}\{\boldsymbol{b}^+\} \leq \epsilon^+, \quad \text{sum}\{\boldsymbol{b}^-\} \leq \epsilon^-, \tag{92}$$

$$\sum_{d | x_d = 1} b_d^+ = 0, \quad \sum_{d | x_d = 0} b_d^- = 0. \tag{93}$$

The above problem can be solved by finding all negative entries of $\boldsymbol{\gamma}^+$ and $\boldsymbol{\gamma}^-$ and then summing up the $\min\left( \text{sum}\{1 - \boldsymbol{x}\}, b_d^+ \right)$ (or $\min\left( \text{sum}\{\boldsymbol{x}\}, b_d^- \right)$, respectively) smallest ones. Using these $\underline{\eta}^{(n)}$ and binary indicator variables $\boldsymbol{t} \in \{0,1\}^{D_{\text{out}}}$, we can restate the above problem as the mixed-integer problem

$$\min_{\boldsymbol{b}^+, \boldsymbol{b}^- \in \{0,1\}^{D_{\text{in}}}, \boldsymbol{t} \in \{0,1\}^{D_{\text{out}}}} \sum_{n \in \mathbb{T}} t_n \tag{94}$$

$$\text{s.t.} \quad \left( \boldsymbol{\gamma}^+ \right)^T \boldsymbol{b}^+ + \left( \boldsymbol{\gamma}^- \right)^T \boldsymbol{b}^- \geq t_n \underline{\eta}^{(n)} + (1 - t_n)\eta^{(n)}, \tag{95}$$

$$\text{sum}\{\boldsymbol{b}^+\} \leq \epsilon^+, \quad \text{sum}\{\boldsymbol{b}^-\} \leq \epsilon^-, \tag{96}$$

$$\sum_{d | x_d = 1} b_d^+ = 0, \quad \sum_{d | x_d = 0} b_d^- = 0. \tag{97}$$

The first constraint ensures that $t_n$ can only be set to 0 if the l.h.s. is greater or equal $\eta_n$, i.e. only when the base certificate can no longer guarantee robustness. The efficiency of the certificate can be improved by applying any of the techniques discussed in Section A.

## C  MONTE CARLO RANDOMIZED SMOOTHING

To make predictions and certify robustness, randomized smoothing requires computing certain properties of the distribution of a base model's output, given an input smoothing distribution. For example, the certificate of Cohen et al. (2019) assumes that the smoothed model $f$ predicts the most likely label output by base model $g$, given a smoothing distribution $\mathcal{N}(\mathbf{0}, \sigma \cdot \mathbf{1})$: $f(\boldsymbol{x}) = \operatorname{argmax}_{y \in \mathbb{Y}} \Pr_{\boldsymbol{z} \sim \mathcal{N}(\mathbf{0}, \sigma \cdot \mathbf{1})} [g(\boldsymbol{x} + \boldsymbol{z}) = y]$. To certify the robustness of a smoothed prediction $y = f(\boldsymbol{x})$ for a specific input x, we have to compute the probability $q = \Pr_{\boldsymbol{z} \sim \mathcal{N}(\mathbf{0}, \sigma \cdot \mathbf{1})} [g(\boldsymbol{x} + \boldsymbol{z}) = y]$ to then calculate the maximum certifiable radius $\sigma \Phi^{-1}(q)$ with standard-normal inverse CDF $\Phi^{-1}$. For complicated models like deep neural networks, computing such properties in closed form is usually not tractable. Instead, they have to be estimated using Monte Carlo sampling. The result are predictions and certificates that only hold with a certain probability.

Randomized smoothing with Monte Carlo sampling usually consists of three distinct steps. First, a small number of samples $N_1$ from the smoothing distribution are used to generate a candidate prediction $\hat{y}$, e.g. the most frequently predicted class. Then, a second round of $N_2$ samples is taken and a statistical test is used to determine whether the candidate prediction is likely to be the actual prediction of smoothed classifier $f$, i.e. whether $\hat{y} = f(\boldsymbol{x})$ with a certain probability (1 - $\alpha_1$). If this is not the case, one has to abstain from making a prediction (or generate a new candidate prediction). To certify the robustness of prediction $\hat{y}$, a final round of $N_3$ samples is taken to estimate all quantities needed for the certificate. In the case of (Cohen et al., 2019), we need to estimate the probability $q = \Pr_{\boldsymbol{z} \sim \mathcal{N}(\mathbf{0}, \sigma \cdot \mathbf{1})} [g(\boldsymbol{x} + \boldsymbol{z}) = \hat{y}]$ to compute the certificate $\sigma \Phi^{-1}(q)$, whose strength is monotonically increasing in $q$. To ensure that the certificate holds with high probability (1 - $\alpha_2$), we have to compute a probabilistic lower bound $\underline{q} \leq q$. If one wants to certify robustness for all predictions, as we do in our experiments, one can also use the same samples for the abstention test and the certificates. One particularly simple abstention mechanism is to just compute the Monte Carlo randomized smoothing certificate to determine whether $\forall \boldsymbol{x}' \in \{\boldsymbol{x}\} : f(\boldsymbol{x}') = \hat{y}$ with high probability, i.e. whether the prediction is robust to input $\boldsymbol{x}'$ that is the result of "perturbing" clean input $\boldsymbol{x}$ with zero adversarial budget.

In the following, we discuss how we perform Monte Carlo randomized smoothing for our base certificates, as well as the baselines we use for our experimental evaluation. In Section C.4, we discuss how we account for the multiple comparisons problem, i.e. the fact that we are not just trying to probabilistically certify a single prediction, but multiple predictions at once.

### C.1  MONTE CARLO BASE CERTIFICATES FOR CONTINUOUS DATA

For our base certificates for continuous data, we follow the approach we already discussed in the previous paragraphs (recall that the certificate of Cohen et al. (2019) is a special case of our certificate with Gaussian noise for $l_2$ perturbations). We are given an input space $\mathbb{X}^{D_{\text{in}}}$, label space $\mathbb{Y}$, base model (or – in the case of multi-output classifiers – base model output) $g : \mathbb{X}^{D_{\text{in}}} \to \mathbb{Y}$ and smoothing distribution $\Psi(\boldsymbol{x})$ (either multivariate Gaussian or multivariate uniform). To generate a candidate prediction, we apply the base classifier to $N_1$ samples from the smoothing distribution in order to obtain predictions $(y^{(1)}, \dots, y^{(N_1)})$ and compute the majority prediction $\hat{y} = \operatorname{argmax}_{y \in \mathbb{Y}} \{n \mid y^{(n)} = \hat{y}\}$. Recall that for Gaussian and uniform noise, our certificate guarantees $\forall \boldsymbol{x}' \in \mathbb{H} : f(\boldsymbol{x}) = \hat{y}$ with

$$\mathbb{H} = \left\{ \boldsymbol{x}' \in \mathbb{X}^{D_{\text{in}}} \,\middle|\, \sum_{d=1}^{D_{\text{in}}} w_d \cdot |x'_d - x_d|^p < \eta \right\}$$

with $\eta = \left(\Phi^{-1}(q)\right)^2$ or $\eta = \Phi^{-1}(q)$ (depending on the distribution), $q = \Pr_{\boldsymbol{z} \sim \mathcal{N}(\mathbf{0}, \sigma \cdot \mathbf{1})} [g(\boldsymbol{x} + \boldsymbol{z}) = \hat{y}]$ and standard-normal inverse CDF $\Phi^{-1}$. To obtain a probabilistic certificate that holds with high probability $1 - \alpha$, we need a probabilistic lower bound on $\eta$. Both $\eta$ are monotonically increasing in $q$, i.e. we can bound them by finding a lower bound $\underline{q}$ on $q$. For this, we take $N_2$ more samples from the smoothing distribution and compute a Clopper-Pearson lower confidence bound (Clopper & Pearson, 1934) on $q$. For abstentions, we use the aforementioned simple

mechanism: We test whether $x \in \mathbb{H}$. Given the definition of $\mathbb{H}$, this is equivalent to testing whether

$$0 < \Phi^{-1}(\underline{q})$$
$$\iff \Phi(0) < \underline{q}$$
$$\iff 0.5 < \underline{q}.$$

If $\underline{q} \leq 0.5$, we abstain.

## C.2 MONTE CARLO VARIANCE SMOOTHING

In variance smoothing, we smooth a model's softmax scores. That is, we are given an input space $\mathbb{X}^{D_{\text{in}}}$, label space $\mathbb{Y}$, base model (or – in the case of multi-output classifiers – base model output) $g : \mathbb{X}^{D_{\text{in}}} \to \Delta_{|\mathbb{Y}|}$ with $|\mathbb{Y}|$-dimensional probability simplex $\Delta_{|\mathbb{Y}|}$ and smoothing distribution $\Psi(x)$ (Bernoulli or sparsity-aware noise, in the case of binary data). To generate a candidate prediction, we apply the base classifier to $N_1$ samples from the smoothing distribution in order to obtain vectors $\left(s^{(1)}, \ldots, s^{(N_1)}\right)$ with $s \in \Delta_{|\mathbb{Y}|}$, compute the average softmax scores $\overline{s} = \frac{1}{N_1} \sum_{n=1}^{N} s$ and select the label with the highest score $\hat{y} = \arg\max_y \overline{s}_y$.

Recall that our certificate guarantees robustness if the optimal value of the following optimization problem is greater than $0.5$:

$$\min_{h:\mathbb{X} \to \mathbb{R}} \mathbb{E}_{z \sim \Psi(x')} [h(z)] \tag{98}$$

$$\text{s.t.} \quad \mathbb{E}_{z \sim \Psi(x)} [h(z)] \geq \mu, \quad \mathbb{E}_{z \sim \Psi(x)} \left[(h(z) - \nu)^2\right] \leq \sigma^2, \tag{99}$$

with $\mu = \mathbb{E}_{z \sim \Psi(x)} [g(z)_{\hat{y}}]$, $\sigma^2 = \mathbb{E}_{z \sim \Psi(x)} \left[(g(z)_{\hat{y}} - \nu)^2\right]$ and a fixed scalar $\nu \in \mathbb{R}$. To obtain a probabilistic certificate, we have to compute a probabilistic lower bound on the optimal value of the optimization problem. Because it is a minimization problem, this can be achieved by loosening its constraints, i.e. computing a probabilistic lower bound $\underline{\mu}$ on $\mu$ and a probabilistic upper bound $\overline{\sigma^2}$ on $\sigma^2$.

Like in CDF-smoothing (Kumar et al., 2020), we bound the parameters using CDF-based nonparametric confidence intervals. Let $F(s) = \Pr_{z \sim \Psi(x)} [g(z)_{\hat{y}} \leq s]$ be the CDF of the distribution of $g_{\hat{y}}$ under the smoothing distribution $\Psi(x)$. Define $M$ thresholds $\leq 0\tau_1 \leq \tau_2 \ldots, \tau_{M-1} \leq \tau_M \leq 1$ with $\forall m : \tau_m \in [0, 1]$. We then take $N_2$ samples $x^{(1)}, \ldots, x^{(N_2)}$ from the smoothing distribution to compute the empirical CDF $\tilde{F}(s) = \sum_{n=1}^{N_2} \mathrm{I} \left[g(z^{(n)})_{\hat{y}} \leq s\right]$. We can then use the Dvoretzky-Keifer-Wolfowitz inequality (Dvoretzky et al., 1956) to compute an upper bound $\hat{F}$ and a lower bound $\underline{F}$ on the CDF of $g_{\hat{y}}$:

$$\underline{F}(s) = \max\left(\tilde{F}(s) - \upsilon, 0\right) \leq F(s) \leq \min\left(\tilde{F}(s) + \upsilon, 1\right) = \overline{F}(s), \tag{100}$$

with $\upsilon = \sqrt{\frac{\ln 2/\alpha}{2 \cdot N_2}}$, which holds with high probability $(1 - \alpha)$. Using these bounds on the CDF, we can bound $\mu = \mathbb{E}_{z \sim \Psi(x)} [g(z)_{\hat{y}}]$ as follows (Anderson, 1969):

$$\mu \geq \tau_M - \tau_1 \overline{F}(\tau_1) + \sum_{m=1}^{M-1} (\tau_{m+1} - \tau_m) \overline{F}(\tau_m). \tag{101}$$

The parameter $\sigma^2 = \mathbb{E}_{z \sim \Psi(x)} \left[(g(z)_{\hat{y}} - \nu)^2\right]$ can be bounded in a similar fashion. Define $\xi_0, \ldots, \xi_M \in \mathbb{R}_+$ with:

$$\xi_0 = \max_{\kappa \in [0, \tau_1]} \left((\kappa - \nu)^2\right)$$
$$\xi_M = \max_{\kappa \in [\tau_M, 1]} \left((\kappa - \nu)^2\right) \tag{102}$$
$$\xi_m = \max_{\kappa \in [\tau_m, \tau_{m+1}]} \left((\kappa - \nu)^2\right) \quad \forall m \in \{1, \ldots, M-1\},$$

i.e. compute the maximum squared distance to $\nu$ within each bin $[\tau_m, \tau_{m+1}]$. Then:

$$\sigma^2 \leq \xi_0 F(\tau_1) + \xi_M \left(1 - F(\tau_M)\right) + \sum_{m=1}^{M-1} \xi_m \left(F(\tau_{m+1} - F(\tau_m))\right) \tag{103}$$

$$= \xi_M + \sum_{m=1}^{M-1} \left(\xi_{m-1} - \xi_m\right) F(\tau_m) \tag{104}$$

$$\leq \xi_M + \sum_{m=1}^{M-1} \mathrm{sgn}\left(\xi_{m-1} - \xi_m\right) \overline{F}(\tau_m) + \left(1 - \mathrm{sgn}\left(\xi_{m-1} - \xi_m\right)\right) \underline{F}(\tau_m) \tag{105}$$

with probability $(1 - \alpha)$. In the first inequality, we bound the expected squared distance from $\nu$ by assuming that the probability mass in each bin $[\tau_m, \tau_{m+1}]$ is concentrated at the farthest point from $\nu$. The equality is a result of reordering the telescope sum. In the second inequality, we upper-bound the CDF where it is multiplied with a non-negative value and lower-bound it where it is multiplied with a negative value.

With the probabilistic bounds $\underline{\mu}$ and $\overline{\sigma^2}$ we can now – in principle – evaluate our robustness certificate, i.e. check whether

$$\sum_{\boldsymbol{z} \in \mathbb{X}} \frac{\pi_{\boldsymbol{x}'}(\boldsymbol{z})^2}{\pi_{\boldsymbol{x}}(\boldsymbol{z})} < 1 + \frac{1}{\overline{\sigma^2} - \left(\underline{\mu} - \nu\right)^2} \left(\underline{\mu} - \frac{1}{2}\right)^2. \tag{106}$$

where the $\pi$ are the probability mass functions of smoothing distributions $\Psi(\boldsymbol{x})$ and $\Psi(\boldsymbol{x}')$. But one crucial detail of Theorem 3 underlying the certificate was that it only holds for $\nu \leq \underline{\mu}$, i.e. only when this condition is fulfilled can we compute the certificate in closed form by solving the corresponding dual problem. To use the method with Monte Carlo sampling, one has to ensure that $\nu \leq \underline{\mu}$ by first computing $\underline{\mu}$ and then choosing some smaller $\nu$.

In our experiments, we use an alternative method that allows us to use arbitrary $\nu$: From our proof of Theorem 3 we know that the dual problem of Eq. 98 is

$$\max_{\alpha, \beta \geq 0} \alpha \underline{\mu} - \beta \overline{\sigma^2} - \frac{\alpha^2}{4\beta} + \frac{\alpha}{2\beta} - \alpha \nu + \nu - \frac{1}{4\beta} \sum_{\boldsymbol{z} \in \mathbb{X}} \frac{\pi_{\boldsymbol{x}'}(\boldsymbol{z})^2}{\pi_{\boldsymbol{x}}(\boldsymbol{z})}, \tag{107}$$

Instead of trying to find an optimal $\alpha$ (which causes problems in subsequent derivations if $\nu \not\leq \underline{\mu}$), we can simply choose $\alpha = 1$. By duality, the result is still a lower bound on the primal problem, i.e. the certificate remains valid. The dual problem becomes

$$\max_{\beta \geq 0} \underline{\mu} - \beta \overline{\sigma^2} + \frac{1}{4\beta} - \frac{1}{4\beta} \sum_{\boldsymbol{z} \in \mathbb{X}} \frac{\pi_{\boldsymbol{x}'}(\boldsymbol{z})^2}{\pi_{\boldsymbol{x}}(\boldsymbol{z})}. \tag{108}$$

The problem is concave in $\beta$ (because the expected likelihood ratio is $\geq 1$). Finding the optimal $\beta$, comparing the result to $0.5$ and solving for the expected likelihood ratio, shows that a prediction is robust if

$$\sum_{\boldsymbol{z} \in \mathbb{X}} \frac{\pi_{\boldsymbol{x}'}(\boldsymbol{z})^2}{\pi_{\boldsymbol{x}}(\boldsymbol{z})} < 1 + \frac{1}{\overline{\sigma^2}} \left(\underline{\mu} - \frac{1}{2}\right)^2. \tag{109}$$

For our abstention mechanism, like in the previous section, we compute the certificate $\mathbb{H}$ and then test whether $\boldsymbol{x} \in \mathbb{H}$. In the case of Bernoulli smoothing and sparsity-aware smoothing), this corresponds to testing whether

$$1 < \ln \left(1 + \frac{1}{\overline{\sigma^2}} \left(\underline{\mu} - \frac{1}{2}\right)\right) \tag{110}$$

$$\iff \underline{\mu} > \frac{1}{2}. \tag{111}$$

## C.3 Monte Carlo center smoothing

While we can not use center smoothing as a base certificate, we benchmark our method against it during our experimental evaluation. The generation of candidate predictions, the abstention mechanism and the certificate are explained in (Kumar & Goldstein, 2021). The authors allow multiple

options for generating candidate predictions. We use the "$\beta$ minimum enclosing ball" with $\beta = 2$ that is based on pair-wise distance calculations.

### C.4 MULTIPLE COMPARISONS PROBLEM

The first step of our collective certificate is to compute one base certificate for each of the $D_{\text{out}}$ predictions of the multi-output classifier. With Monte Carlo randomized smoothing, we want all of these probabilistic certificates to simultaneously hold with a high probability $(1 - \alpha)$. But as the number of certificates increases, so does the probability of at least one of them being invalid. To account for this *multiple comparisons problem*, we use Bonferroni (Bonferroni, 1936) correction, i.e. compute each Monte Carlo certificate such that it holds with probability $(1 - \frac{\alpha}{n})$.

For base certificates that only depend on $q_n = \Pr_{\boldsymbol{z} \sim \Psi^{(n)}} [g_n(\boldsymbol{z}) = \hat{y}_n]$, i.e. the probability of the base classifier predicting a particular label $\hat{y}_n$ under the smoothing distribution, one can also use the strictly better Holm correction (Holm, 1979). This includes our Gaussian and uniform smoothing certificates for continuous data. Holm correction is a procedure than can be used to correct for the multiple comparisons problem when performing multiple arbitrary hypothesis tests. Given $N$ hypotheses, their $p$-values are ordered in ascending order $p_1, \ldots, p_N$. Starting at $i = 1$, the $i$'th hypothesis is rejected if $p_i < \frac{\alpha}{N+1-i}$, until one reaches an $i$ such that $p_i \geq \frac{\alpha}{N+1-i}$.

Fischer et al. (2021) proposed to use Holm correction as part of their procedure for certifying that all (non-abstaining) predictions of an image segmentation model are robust to adversarial perturbations. In the following, we first summarize their approach and then discuss how Holm correction can be used for certifying our notion of collective robustness, i.e. certifying the number of robust predictions. As in Section C.1, the goal is to obtain a lower bound $\underline{q}_n$ on $q_n = \Pr_{\boldsymbol{z} \sim \Psi^{(n)}} [g_n(\boldsymbol{z}) = \hat{y}_n]$ for each of the $D_{\text{out}}$ classifier outputs. Assume we take $N_2$ samples $\boldsymbol{z}^{(1)}, \ldots, \boldsymbol{z}^{(N_2)}$ from the smoothing distribution. Let $\nu_n = \sum_{i=1}^{N_2} \mathrm{I} \left[ g_n(\boldsymbol{z}^{(i)}) = \hat{y}_n \right]$ and let $\pi : \{1, \ldots, D_{\text{out}}\} \to \{1, \ldots, D_{\text{out}}\}$ be a bijection that orders the $\nu_n$ in descending order, i.e. $\nu_{\pi(1)} \geq \nu_{\pi(2)} \cdots \geq \nu_{\pi(D_{\text{out}})}$. Instead of using Clopper-Pearson confidence intervals to obtain tight lower bounds on the $q_n$, Fischer et al. (2021) define a threshold $\tau \in [0.5, 1)$ and use Binomial tests to determine for which $n$ the bound $\tau \leq q_n$ holds with high-probability. Let $\mathrm{BinP}\,(\nu_n, N_2, \leq, \tau)$ be the p-value of the one-sided binomial test, which is monotonically decreasing in $\nu_n$. Following the Holm correction scheme, the authors test whether

$$\mathrm{BinP}\,(\nu_{\pi(k)}, N_2, \leq, \tau) < \frac{\alpha}{D_{\text{out}} + 1 - k} \tag{112}$$

for $k = 1, \ldots, D_{\text{out}}$ until reaching a $k^*$ for which the null-hypothesis can no longer be rejected, i.e. the p-value is g.e.q. $\frac{\alpha}{D_{\text{out}} + 1 - k^*}$. They then know that with probability $1 - \alpha$, the bound $\tau \leq q_n$ holds for all $n \in \{\pi(k) \mid k \in \{1, \ldots, k^*\}\}$. For these outputs, they use the lower bound $\tau$ to compute robustness certificates. They abstain with all other outputs.

This approach is sensible when one is concerned with the least robust prediction from a set of predictions. But our collective certificate benefits from having tight robustness guarantees for each of the individual predictions. Holm correction can be used with arbitrary hypothesis tests. For instance, we can use a different threshold $\tau_n$ per output $g_n$, i.e. test whether

$$\mathrm{BinP}\,(\nu_{\pi(k)}, N_2, \leq, \tau_{pi(k)}) < \frac{\alpha}{D_{\text{out}} + 1 - k} \tag{113}$$

for $k = 1, \ldots, D_{\text{out}}$. In particular, we can use

$$\tau_n = \sup_t \text{ s.t. } \mathrm{BinP}\,(\nu_n, N_2, \leq, t) < \frac{\alpha}{D_{\text{out}} + 1 - \pi^{-1}(n)}, \tag{114}$$

i.e. choose the largest threshold such that the null hypothesis can still be rejected. Eq. 114 is the lower Clopper-Pearson confidence bound with significance $\frac{\alpha}{D_{\text{out}} + 1 - \pi^{-1}(n)}$. This means that, instead of performing hypothesis tests, we can obtain probabilistic lower bounds $\underline{q}_n \leq q_n$ by computing Clopper-Pearson confidence bounds with significance parameters $\frac{\alpha}{D_{\text{out}}}, \ldots, \frac{\alpha}{1}$. The $\underline{q}_n$ can then be used to compute the base certificates. Due to the definition of the $\tau_n$, all of the null hypotheses are rejected, i.e. we obtain valid probabilistic lower bounds on all $q_n$. We can thus use the abstention mechanism from Section C.1, i.e. only abstain if $\underline{q}_n \leq 0.5$.

A direct consequence of the results above is that using Clopper-Pearson confidence intervals and Holm correction will yield stronger per-prediction robustness guarantees and lower abstention rates

than the method of Fischer et al. (2021). The Clopper-Pearson-based method only abstains if one cannot guarantee that $q_n > 0.5$ with high probability, while their method abstains if one cannot guarantee that $q_n \geq \tau$ with $\tau \geq 0.5$ (or specific other predictions abstain). For all non-abstaining predictions, the Clopper-Pearson-based certificate will be at least as strong as that obtained using a single threshold $\tau$, as it computes the tightest bound for which the null hypothesis can still be rejected (see Eq. 114). Consequently, a naïve collective robustness certificate (i.e. counting the number of predictions whose robustness are guaranteed by the base certificates) based on Clopper-Pearson bounds will also be stronger. It should however be noted that this is only relevant for our notion of collective robustness, not for the one considered by Fischer et al. (2021). It should also be noted that their method could potentially be used with other methods of family-wise error rate correction, although they state that "these methods do not scale to realistic segmentation problems" and do not discuss any further details.

Our above discussion shows that, for our notion of collective robustness, the naïve collective certificate using Clopper-Pearson confidence bounds and Holm correction is at least as strong as that of Fischer et al. (2021) (or alternatively: the certificate of Fischer et al. (2021) is a naïve collective certificate with Holm correction, but with weaker per-prediction certificates). Conversely, the certificate based on Clopper-Pearson confidence bounds is at least as strong as that of Fischer et al. (2021) when certifying their notion of collective robustness, i.e. determining whether *all* non-abstaining predictions are robust, given adversarial budget $\epsilon$. To certify this notion of robustness, they simply iterate over all predictions and determine whether all non-abstaining predictions are certifiably robust, given $\epsilon$. Naturally, as the Clopper-Pearson-based certificates are stronger, any prediction that is robust according to (Fischer et al., 2021) is also robust acccording to the Clopper-Pearson-based certificates. The only difference is that, for $\tau > 0.5$, their method will have more abstaining predictions. But, due to the direct correspondence of Clopper-Pearson confidence bounds and Binomial tests, we can modify our abstention mechanism to obtain exactly the same set of abstaining predictions: We simply have to use $\underline{q}_n \leq \tau$ instead of $\underline{q_n} \leq 0.5$ as our abstention criterion. Finally, it should be noted that the proposed collective certificate is at least as strong as the naïve collective certificate (see Section 4). Thus, letting the set of targeted predictions $\mathbb{T}$ be the set of all non-abstaining predictions and checking whether the collective certificate guarantees robustness for all of $\mathbb{T}$ will also result in a certificate that is at least as strong as that of Fischer et al. (2021) in their setting.

In our experiments (see Section 8), we use Holm correction for our naïve collective certificate and the weaker Bonferroni correction for our base certificates. This is only meant to slightly skew the results in favor of our baselines. Holm correction can in principle also be used when computing the base certificates, in order to improve our proposed collective cert.

# D COMPARISON TO THE COLLECTIVE CERTIFICATE OF SCHUCHARDT ET AL.

In the following, we first present the collective certificate for binary graph-structured data proposed by Schuchardt et al. (2021) (see Section D.1. We then show that, when using sparsity-aware smoothing distributions (Bojchevski et al., 2020) – the family of smoothing distributions used both in our work and that of Schuchardt et al. (2021) – our certificate subsumes their certificate. That is, our collective robustness certificate based on localized randomized smoothing can provide the same robustness guarantees (see Section D.2).

## D.1 THE COLLECTIVE CERTIFICATE

Their certificate assumes the input space to be $\mathbb{G} = \{0, 1\}^{N \times D} \times \{0, 1\}^{N \times N}$ – the set of undirected attributed graphs with $N$ nodes and $D$ attributes per node. The model is assumed to be a multi-output classifier $f : \mathbb{G} \to \mathbb{Y}^N$ that assigns a label from label set $\mathbb{Y}$ to each of the nodes. Given an input graph $\mathcal{G} = (\boldsymbol{X}, \boldsymbol{A})$ and a corresponding prediction $\boldsymbol{y} = f(G)$, they want to certify collective robustness to a set of perturbed graphs $\mathbb{B} \subseteq \mathbb{G}$. The perturbation model $\mathbb{B}$ is characterized by four scalar parameters $r_{\boldsymbol{X}}^+, r_{\boldsymbol{X}}^-, r_{\boldsymbol{A}}^+, r_{\boldsymbol{A}}^+ \in \mathbb{N}_0$, specifying the number of bits the adversary is allowed to add ($0 \to 1$) and delete ($1 \to 0$) in the attribute and adjacency matrix, respectively. It can also be extended to feature additional constraints (e.g. per-node budgets). We discuss how these can be integrated after showing our main result. A formal definition of the perturbation model can be found in Section B of (Schuchardt et al., 2021).

The goal of their work is to certify collective robustness for a set of targeted nodes $\mathbb{T} \subseteq \{1, \dots, N\}$, i.e. compute a lower bound on

$$\min_{G' \in \mathbb{B}} \sum_{n \in \mathbb{T}} \mathrm{I}\left[f_n(G') = y_n\right]. \tag{115}$$

Their approach to obtaining this lower-bound shares the same high-level idea as ours: Combining per-prediction base certificates and leveraging some notion of locality. But while our method uses localized randomized smoothing, i.e. smoothing different outputs with different non-i.i.d. smoothing distributions, to obtain base certificates that encode locality, their method uses a-priori knowledge about the strict locality of the classifier $f$. A model is strictly local if each of its outputs $f_n$ only operates on a well-defined subset of the input data. To encode this strict locality, Schuchardt et al. (2021) associate each output $f_n$ with an indicator vector $\boldsymbol{\psi}^{(n)}$ and an indicator matrix $\boldsymbol{\Psi}^{(n)}$ that fulfill

$$\sum_{m=1}^{N} \sum_{d=1}^{D} \psi_m^{(n)} \mathrm{I}\left[X_{m,d} \neq X'_{i,j}\right] + \sum_{i=1}^{N} \sum_{j=1}^{N} \Psi_m^{(n)} \mathrm{I}\left[A_{m,d} \neq A'_{i,j}\right] = 0$$
$$\implies f_n(\boldsymbol{X}, \boldsymbol{A}) = f_n(\boldsymbol{X'}, \boldsymbol{A'}). \tag{116}$$

for any perturbed graph $\mathcal{G}' = (\boldsymbol{X'}, \boldsymbol{A'})$. Eq. 116 expresses that the prediction of output $f_n$ remains unchanged if all inputs in its receptive field remain unchanged. Conversely, it expresses that perturbations outside the receptive field can be ignored. Unlike in our work, Schuchardt et al. (2021) describe their base certificates as sets in adversarial budget space. That is, some certification procedure is applied to each output $f_n$ to obtain a set

$$\mathbb{K}^{(n)} \subseteq [r_{\boldsymbol{X}}^+] \times [r_{\boldsymbol{X}}^-] \times [r_{\boldsymbol{A}}^+] \times [r_{\boldsymbol{X}}^-] \tag{117}$$

with $[k] = \{0, \dots, k\}$. If $\begin{bmatrix} c_{\boldsymbol{X}}^+ & c_{\boldsymbol{X}}^- & c_{\boldsymbol{A}}^+ & c_{\boldsymbol{A}}^- \end{bmatrix}^T \in \mathbb{K}^{(n)}$, then prediction $y_n$ is robust to any perturbed input with exactly $c_{\boldsymbol{X}}^+$ attribute additions, $c_{\boldsymbol{X}}^-$ attribute deletions, $c_{\boldsymbol{A}}^+$ edge additions and $c_{\boldsymbol{A}}^-$ edge deletions. A more detailed explanation can be found in Section 3 of (Schuchardt et al., 2021). Note that the base certificates only depend on the number of perturbations, not their location in the input. Only combining them using the receptive field indicators from Eq. 116 makes it possible to obtain a collective certificate that is better than a naïve combination of the base certificates (i.e. counting how many predictions are certifiably robust to the collective threat model). The

resulting collective certificate is

$$\min_{\boldsymbol{b}^+, \boldsymbol{b}^+, \boldsymbol{B}^+, \boldsymbol{B}^-} \sum_{n \in \mathbb{T}} \mathrm{I}\left[\left[\left(\boldsymbol{\psi}^{(n)}\right)^T \boldsymbol{b}_{\boldsymbol{X}}^+ \quad \left(\boldsymbol{\psi}^{(n)}\right)^T \boldsymbol{b}_{\boldsymbol{X}}^- \quad \sum_{i,j} \Psi_{i,j}^{(n)} \boldsymbol{B}_{i,j}^+ \quad \sum_{i,j} \Psi_{i,j}^{(n)} \boldsymbol{B}_{i,j}^-\right]^T \in \mathbb{K}^{(n)}\right]$$
(118)

$$\text{s.t.} \quad \sum_{m=1}^N b_m^+ \leq r_{\boldsymbol{X}}^+, \quad \sum_{m=1}^N b_m^- \leq r_{\boldsymbol{X}}^-, \quad \sum_{i=1}^N \sum_{j=1}^N B_{i,j}^+ \leq r_{\boldsymbol{A}}^+, \quad \sum_{i=1}^N \sum_{j=1}^N B_{i,j}^- \leq r_{\boldsymbol{A}}^-, \quad (119)$$

$$\boldsymbol{b}^+, \boldsymbol{b}^- \in \mathbb{N}_0^N \quad \boldsymbol{B}^+, \boldsymbol{B}^- \in \mathbb{N}_0^{N \times N}. \quad (120)$$

The variables defined in Eq. 120 model how the adversary allocates their adversarial budget, i.e. how many attributes are perturbed per node and which edges are modified. Eq. 119 ensures that this allocation in compliant with the collective threat model. Finally, in Eq. 118 the indicator vector and matrix $\boldsymbol{\psi}^{(n)}$ and $\boldsymbol{\Psi}^{(n)}$ are used to mask out any allocated perturbation budget that falls outside the receptive field of $f_n$ before evaluating its base certificate.

To solve the optimization problem, Schuchardt et al. (2021) replace each of the indicator functions with binary variables and include additional constraints to ensure that they have value 1 i.f.f. the indicator function would have value 1. To do so, they define one linear constraint per point separating the set of certifiable budgets $\mathbb{K}^{(n)}$ from its complement $\overline{\mathbb{K}}^{(n)}$ in adversarial budget space (the "pareto front" discussed in Section 3 of (Schuchardt et al., 2021)).

From the above explanation, the main drawbacks of this collective certificate compared to our localized randomized smoothing approach and corresponding collective certificate should be clear. Firstly, if the classifier $f$ is not strictly local, i.e. the receptive field indicators $\boldsymbol{\psi}$ and $\boldsymbol{\Psi}$ only have non-zero entries, then all base certificates are evaluated using the entire collective adversarial budget. It thus degenerates to the naïve collective certificate. Secondly, even if the model is strictly local, each of the output may assign varying levels of importance to different parts of its receptive field. Their method is incapable of capturing this additional soft locality. Finally, their means of evaluating the base certificates may involve evaluating a large number of linear constraints. Our method, on the other hand, only requires a single constraint per prediction. Our collective certificate can thus be more efficiently computed.

## D.2 PROOF OF SUBSUMPTION

In the following, we show that any robustness certificate obtained by using the collective certificate of Schuchardt et al. (2021) with sparsity-aware randomized smoothing base certificates can also be obtained by using our proposed collective certificate with an appropriately parameterized localized smoothing distribution. The fundamental idea is that, for randomly smoothed models, completely randomizing all input dimensions outside a receptive field is equivalent to masking out any perturbations outside the receptive field.

First, we derive the certificate of Schuchardt et al. (2021) for predictions obtained via sparsity-aware smoothing. Schuchardt et al. (2021) require base certificates that guarantee robustness when $\left[c_{\boldsymbol{X}}^+ \quad c_{\boldsymbol{X}}^- \quad c_{\boldsymbol{A}}^+ \quad c_{\boldsymbol{A}}^-\right]^T \in \mathbb{K}^{(n)}$, where the $c$ indicate the number of added and deleted attribute and adjacency bits. That is, the certificates must only depend on the number of perturbations, not on their location. To achieve this, all entries of the attribute matrix and all entries of the adjacency matrix, respectively, must share the same distribution. For the attribute matrix, they define scalar distribution parameters $p_{\boldsymbol{X}}^+, p_{\boldsymbol{A}}^- \in [0, 1]$. Given attribute matrix $\boldsymbol{X} \in \{0, 1\}^{N \times D}$, they then sample random attribute matrices $\boldsymbol{Z}_{\boldsymbol{X}}$ that are distributed according to sparsity-aware smoothing distribution $\mathcal{S}\left(\boldsymbol{X}, \boldsymbol{1} \cdot p_{\boldsymbol{X}}^+, \boldsymbol{1} \cdot p_{\boldsymbol{X}}^-\right)$ (see Section B.2.2), i.e.

$$\Pr[(Z_{\boldsymbol{X}})_{m,d} = 0] = \left(1 - p_{\boldsymbol{X}}^+\right)^{1 - X_{m,d}} \cdot \left(p_{\boldsymbol{X}}^-\right)^{X_{m,d}},$$

$$\Pr[(Z_{\boldsymbol{X}})_{m,d} = 1] = \left(p_{\boldsymbol{X}}^+\right)^{1 - X_{m,d}} \cdot \left(1 - p_{\boldsymbol{X}}^-\right)^{X_{m,d}}.$$

Given input adjacency matrix $\boldsymbol{A}$, random adjacency matrices $\boldsymbol{Z}_{\boldsymbol{A}}$ are sampled from the distribution $\mathcal{S}\left(\boldsymbol{A}, \boldsymbol{1} \cdot p_{\boldsymbol{A}}^+, \boldsymbol{1} \cdot p_{\boldsymbol{A}}^-\right)$. Applying Theorem 2 (to the flattened and concatenated attribute and adjacency matrices) shows that smoothed prediction $y_n = f_n(\boldsymbol{X}, \boldsymbol{A})$ is robust to the perturbed graph

$(\boldsymbol{X}', \boldsymbol{A}')$ if

$$
\sum_{m=1}^{N} \sum_{d=1}^{D} \gamma_{\boldsymbol{X}}^{+} \cdot \mathrm{I}\left[X_{m,d} = 0 \neq X'_{m,d}\right] + \gamma_{\boldsymbol{X}}^{-} \cdot \mathrm{I}\left[X_{m,d} = 1 \neq X'_{m,d}\right]
$$
$$
+ \sum_{i=1}^{N} \sum_{i=1}^{N} \gamma_{\boldsymbol{A}}^{+} \cdot \mathrm{I}\left[A_{i,j} = 0 \neq A'_{i,j}\right] + \gamma_{\boldsymbol{A}}^{-} \cdot \mathrm{I}\left[A_{i,j} = 1 \neq A'_{i,j}\right] \tag{121}
$$
$$
< \eta^{(n)}
$$

with $\quad \gamma_{\boldsymbol{X}}^{+} \quad = \quad \ln\left(\frac{\left(p_{\boldsymbol{X}}^{-}\right)^2}{1 - p_{\boldsymbol{X}}^{+}} + \frac{\left(1 - p_{\boldsymbol{X}}^{-}\right)^2}{p_{\boldsymbol{X}}^{+}}\right), \quad \gamma_{\boldsymbol{X}}^{-} \quad = \quad \ln\left(\frac{\left(1 - p_{\boldsymbol{X}}^{+}\right)^2}{p_{\boldsymbol{X}}^{-}} + \frac{\left(p_{\boldsymbol{X}}^{+}\right)^2}{1 - p_{\boldsymbol{X}}^{-}} \cdot\right), \quad \gamma_{\boldsymbol{A}}^{+} \quad =$
$\ln\left(\frac{\left(p_{\boldsymbol{A}}^{-}\right)^2}{1 - p_{\boldsymbol{A}}^{+}} + \frac{\left(1 - p_{\boldsymbol{A}}^{-}\right)^2}{p_{\boldsymbol{A}}^{+}}\right), \gamma_{\boldsymbol{A}}^{-} = \ln\left(\frac{\left(1 - p_{\boldsymbol{A}}^{+}\right)^2}{p_{\boldsymbol{A}}^{-}} + \frac{\left(p_{\boldsymbol{A}}^{+}\right)^2}{1 - p_{\boldsymbol{A}}^{-}} \cdot\right)$ and $\eta^{(n)} = \ln\left(1 + \frac{1}{\sigma^{(n)^2}}\left(\mu^{(n)} - \frac{1}{2}\right)^2\right)$,
where $\mu^{(n)}$ is the mean and $\sigma^{(n)}$ is the variance of the base classifier's output distribution, given the input smoothing distribution. Since the indicator functions for each perturbation type in Eq. 121 share the same weights, Eq. 121 can be rewritten as

$$
\gamma_{\boldsymbol{X}}^{+} c_{\boldsymbol{X}}^{+} + \gamma_{\boldsymbol{X}}^{-} c_{\boldsymbol{X}}^{-} + \gamma_{\boldsymbol{A}}^{+} c_{\boldsymbol{A}}^{+} + \gamma_{\boldsymbol{A}}^{-} c_{\boldsymbol{A}}^{-} \leq \eta^{(n)}, \tag{122}
$$

where $c_{\boldsymbol{X}}^{+}, c_{\boldsymbol{X}}^{-}, c_{\boldsymbol{A}}^{+}, c_{\boldsymbol{A}}^{-}$ are the overall number of added and deleted attribute and adjacency bits, respectively. Eq. 122 matches the notion of base certificates defined by Schuchardt et al. (2021), i.e. it corresponds to a set $\mathbb{K}^{(n)}$ in adversarial budget space for which we provably know that prediction $y_n$ is certifiably robust if $\begin{bmatrix} c_{\boldsymbol{X}}^{+} & c_{\boldsymbol{X}}^{-} & c_{\boldsymbol{A}}^{+} & c_{\boldsymbol{A}}^{-} \end{bmatrix}^{T} \in \mathbb{K}^{(n)}$. When this base certificate is used, i.e. we insert the base certificate Eq. 122 into objective function Eq. 118, the collective certificate of Schuchardt et al. (2021) becomes equivalent to

$$
\min_{\boldsymbol{b}^{+}, \boldsymbol{b}^{+}, \boldsymbol{B}^{+}, \boldsymbol{B}^{-}} \sum_{n \in \mathbb{T}} \mathrm{I}\left[\gamma_{\boldsymbol{X}}^{+} \left(\boldsymbol{\psi}^{(n)}\right)^{T} \boldsymbol{b}_{\boldsymbol{X}}^{+} + \gamma_{\boldsymbol{X}}^{-} \left(\boldsymbol{\psi}^{(n)}\right)^{T} \boldsymbol{b}_{\boldsymbol{X}}^{-}\right.
$$
$$
\left. + \gamma_{\boldsymbol{A}}^{+} \sum_{i,j} \Psi_{i,j}^{(n)} \boldsymbol{B}_{i,j}^{+} + \sum_{i,j} \gamma_{\boldsymbol{A}}^{-} \Psi_{i,j}^{(n)} \boldsymbol{B}_{i,j}^{-} \leq \eta^{(n)}\right] \tag{123}
$$

$$
\text{s.t.} \quad \sum_{m=1}^{N} b_m^{+} \leq r_{\boldsymbol{X}}^{+}, \quad \sum_{m=1}^{N} b_m^{-} \leq r_{\boldsymbol{X}}^{-}, \quad \sum_{i=1}^{N} \sum_{j=1}^{N} B^{+}_{i,j} \leq r_{\boldsymbol{A}}^{+}, \quad \sum_{i=1}^{N} \sum_{j=1}^{N} B^{-}_{i,j} \leq r_{\boldsymbol{A}}^{-}, \tag{124}
$$

$$
\boldsymbol{b}^{+}, \boldsymbol{b}^{-} \in \mathbb{N}_0^{N} \quad \boldsymbol{B}^{+}, \boldsymbol{B}^{-} \in \mathbb{N}_0^{N \times N}. \tag{125}
$$

Next, we show that obtaining base certificates through localized randomized smoothing with appropriately chosen parameters and using these base certificates within our proposed collective certificate (see Section 6) will result in the same optimization problem. Instead of using the same smoothing distribution for all outputs, we use different distribution parameters for each one. For the $n$'th output, we sample random attributes matrices from distribution $\mathcal{S}\left(\boldsymbol{X}, \boldsymbol{\Theta}_{\boldsymbol{X}}^{+ (n)}, \boldsymbol{\Theta}_{\boldsymbol{X}}^{- (n)}\right)$ with $\boldsymbol{\Theta}_{\boldsymbol{X}}^{+ (n)}, \boldsymbol{\Theta}_{\boldsymbol{X}}^{- (n)} \in [0, 1]^{N \times D}$. Note that, in order to avoid having to index flattened vectors, we overload the definition of sparsity-aware smoothing to allow for matrix-valued parameters. For example, the value $\boldsymbol{\Theta}_{\boldsymbol{X}}^{+ (n)}{}_{n,d}$ indicates the probability of flipping the value of input attribute $X_{n,d}$ from 0 to 1 and the value $\boldsymbol{\Theta}_{\boldsymbol{X}}^{- (n)}{}_{n,d}$ indicates the probability of flipping the value of input attribute $X_{n,d}$ from 1 to 0. We choose the following values for these parameters:

$$
\boldsymbol{\Theta}_{\boldsymbol{X}}^{+ (n)}{}_{m,d} = \psi_m^{(n)} \cdot p_{\boldsymbol{X}}^{+} + \left(1 - \psi_m^{(n)}\right) \cdot 0.5, \tag{126}
$$

$$
\boldsymbol{\Theta}_{\boldsymbol{X}}^{- (n)}{}_{m,d} = \psi_m^{(n)} \cdot p_{\boldsymbol{X}}^{-} + \left(1 - \psi_m^{(n)}\right) \cdot 0.5, \tag{127}
$$

where $\boldsymbol{\psi}^{(n)}$ is the receptive field indicator vector defined in Eq. 116 and $p_{\boldsymbol{X}}^{+}, \cdot p_{\boldsymbol{X}}^{-} \in [0, 1]$ are the same flip probabilities we used for the certificate of Schuchardt et al. (2021). Due to this parameterization, attribute bits inside the receptive field are randomized using the same distribution as in the certificate of Schuchardt et al. (2021), while attribute bits outside are set to either

0 or 1 with equal probability. Similarly, we sample random adjacency matrices from distribution $\mathcal{S}\left(\boldsymbol{A}, \boldsymbol{\Theta}_{\boldsymbol{A}}^{+\,(n)}, \boldsymbol{\Theta}_{\boldsymbol{A}}^{-\,(n)}\right)$ with $\boldsymbol{\Theta}_{\boldsymbol{A}}^{+\,(n)}, \boldsymbol{\Theta}_{\boldsymbol{A}}^{-\,(n)} \in [0,1]^{N \times D}$ and

$$\Theta_{\boldsymbol{A}\,i,j}^{+\,(n)} = \Psi_{i,j}^{(n)} \cdot p_{\boldsymbol{A}}^{+} + \left(1 - \Psi_{i,j}^{(n)}\right) \cdot 0.5, \tag{128}$$

$$\Theta_{\boldsymbol{A}\,u,j}^{-\,(n)} = \Psi_{i,j}^{(n)} \cdot p_{\boldsymbol{A}}^{-} + \left(1 - \Psi_{i,j}^{(n)}\right) \cdot 0.5, \tag{129}$$

where $\boldsymbol{\Psi}^{(n)}$ is the receptive field indicator matrix defined in Eq. 116. Note that, since we only alter the distribution of bits outside the receptive field, the smoothed prediction $y_n = f_n(\boldsymbol{X}, \boldsymbol{A})$ will be the same as the one obtained via the smoothing distribution used by Schuchardt et al. (2021). Applying Theorem 2 (to the flattened and concatenated attribute and adjacency matrices) shows that smoothed prediction $y_n = f_n(\boldsymbol{X}, \boldsymbol{A})$ is robust to the perturbed graph $(\boldsymbol{X}', \boldsymbol{A}')$ if

$$
\begin{aligned}
&\sum_{m=1}^{N} \sum_{d=1}^{D} \tau_{\boldsymbol{X}\,m,d}^{+} \cdot \mathrm{I}\left[X_{m,d} = 0 \neq X'_{m,d}\right] + \tau_{\boldsymbol{X}\,m,d}^{-} \cdot \mathrm{I}\left[X_{m,d} = 1 \neq X'_{m,d}\right] \\
&+ \sum_{i=1}^{N} \sum_{j=1}^{N} \tau_{\boldsymbol{A}\,i,j}^{+} \cdot \mathrm{I}\left[A_{i,j} = 0 \neq A'_{i,j}\right] + \tau_{\boldsymbol{A}\,i,j}^{-} \cdot \mathrm{I}\left[A_{i,j} = 1 \neq A'_{i,j}\right] \\
&< \eta^{(n)}.
\end{aligned}
\tag{130}
$$

Because we only changed the distribution outside the receptive field, the scalar $\eta^{(n)}$, which depends on the output distribution's mean and variance $\mu$ and $\sigma$ will be the same as the one obtained via the smoothing scheme used by Schuchardt et al. (2021) et al. Due to Theorem 2 and the definition of our smoothing distribution parameters in Eqs. (126) to (129), the scalars $\tau_{\boldsymbol{X}\,m,d}^{+}, \tau_{\boldsymbol{X}\,m,d}^{-}, \tau_{\boldsymbol{A}\,i,j}^{+}, \tau_{\boldsymbol{A}\,i,j}^{-}$ have the following values:

$$\tau_{\boldsymbol{X}\,m,d}^{+} = \psi_m^{(n)} \cdot \gamma_{\boldsymbol{X}}^{+} + \left(1 - \psi_m^{(n)}\right) \cdot 2 \cdot \ln\left(\frac{(1-0.5)^2}{0.5} + \frac{0.5^2}{1-0.5}\right) \tag{131}$$

$$\tau_{\boldsymbol{X}\,m,d}^{-} = \psi_m^{(n)} \cdot \gamma_{\boldsymbol{X}}^{-} + \left(1 - \psi_m^{(n)}\right) \cdot 2 \cdot \ln\left(\frac{(1-0.5)^2}{0.5} + \frac{0.5^2}{1-0.5}\right) \tag{132}$$

$$\tau_{\boldsymbol{A}\,i,j}^{-} = \Psi_{i,j}^{(n)} \cdot \gamma_{\boldsymbol{A}}^{+} + \left(1 - \Psi_{i,j}^{(n)}\right) \cdot 2 \cdot \ln\left(\frac{(1-0.5)^2}{0.5} + \frac{0.5^2}{1-0.5}\right) \tag{133}$$

$$\tau_{\boldsymbol{A}\,i,j}^{-} = \Psi_{i,j}^{(n)} \cdot \gamma_{\boldsymbol{A}}^{-} + \left(1 - \Psi_{i,j}^{(n)}\right) \cdot 2 \cdot \ln\left(\frac{(1-0.5)^2}{0.5} + \frac{0.5^2}{1-0.5}\right), \tag{134}$$

where the $\gamma$ are the same weights as those of the base certificate Eq. 121 of Schuchardt et al. (2021). Inserting the above values of $\tau$ into the base certificate Eq. 130 and using the fact that $\ln\left(\frac{(1-0.5)^2}{0.5} + \frac{0.5^2}{1-0.5}\right) = \ln(1) = 0$ results in

$$
\begin{aligned}
&\sum_{m=1}^{N} \sum_{d=1}^{D} \psi_m^{(n)} \cdot \gamma_{\boldsymbol{X}}^{+} \cdot \mathrm{I}\left[X_{m,d} = 0 \neq X'_{m,d}\right] + \psi_m^{(n)} \cdot \gamma_{\boldsymbol{X}}^{-} \cdot \mathrm{I}\left[X_{m,d} = 1 \neq X'_{m,d}\right] \\
&+ \sum_{i=1}^{N} \sum_{j=1}^{N} \Psi_{i,j}^{(n)} \cdot \gamma_{\boldsymbol{A}}^{-} \cdot \mathrm{I}\left[A_{i,j} = 0 \neq A'_{i,j}\right] + \Psi_{i,j}^{(n)} \cdot \gamma_{\boldsymbol{A}}^{-} \cdot \mathrm{I}\left[A_{i,j} = 1 \neq A'_{i,j}\right] \\
&< \eta^{(n)}.
\end{aligned}
\tag{135}
$$

While our collective certificate derived in Section 6 only considers one perturbation type, we have already discussed how to certify robustness to perturbation models where there are multiple perturbation types in Section B.2.2: We use a different budget variable per input dimension and perturbation type. Furthermore, the attribute bits of each node share the same noise level. Therefore, we can use the method discussed in Section A.3, i.e. use a single budget variable per node instead of using

one per node and attribute. Modelling our collective problem in this way, using Eq. 135 as our base certificates and rewriting the first two sums using inner products results in the optimization problem

$$
\min_{\boldsymbol{b}^+,\boldsymbol{b}^+,\boldsymbol{B}^+,\boldsymbol{B}^-} \sum_{n \in \mathbb{T}} \mathrm{I}\left[\gamma_{\boldsymbol{X}}^+ \left(\boldsymbol{\psi}^{(n)}\right)^T \boldsymbol{b}_{\boldsymbol{X}}^+ + \gamma_{\boldsymbol{X}}^- \left(\boldsymbol{\psi}^{(n)}\right)^T \boldsymbol{b}_{\boldsymbol{X}}^- \right.
$$

$$
\left. +\gamma_{\boldsymbol{A}}^+ \sum_{i,j} \Psi_{i,j}^{(n)} \boldsymbol{B}_{i,j}^+ + \sum_{i,j} \gamma_{\boldsymbol{A}}^- \Psi_{i,j}^{(n)} \boldsymbol{B}_{i,j}^- \leq \eta^{(n)}\right] \quad (136)
$$

$$
\text{s.t.} \quad \sum_{m=1}^N b_m^+ \leq r_{\boldsymbol{X}}^+, \quad \sum_{m=1}^N b_m^- \leq r_{\boldsymbol{X}}^-, \quad \sum_{i=1}^N \sum_{j=1}^N B^+{}_{i,j} \leq r_{\boldsymbol{A}}^+, \quad \sum_{i=1}^N \sum_{j=1}^N B^-{}_{i,j} \leq r_{\boldsymbol{A}}^-, \quad (137)
$$

$$
\boldsymbol{b}^+, \boldsymbol{b}^- \in \mathbb{N}_0^N \quad \boldsymbol{B}^+, \boldsymbol{B}^- \in \mathbb{N}_0^{N \times N}. \quad (138)
$$

This optimization problem is identical to that of Schuchardt et al. (2021) from Eqs. (123) to (125). The only difference is in how these problems would be mapped to a mixed-integer linear program. We would directly model the indicator functions in the objective using a single linear constraint. Schuchardt et al. (2021) would use multiple linear constraints, each corresponding to one point in the adversarial budget space.

To summarize: For randomly smoothed models, masking out perturbations using a-priori knowledge about a model's strict locality is equivalent to completely randomizing (here: flipping bits with probability $50\%$) parts of the input. While Schuchardt et al. (2021) only derived their certificate for binary data, it is conceivable that their approach could be applied to strictly local models for continuous data. Considering our certificates for Gaussian (Proposition 1) and (Proposition 2) uniform smoothing, where the base certificate weights are $\frac{1}{\sigma^2}$ and $\frac{1}{\lambda}$, respectively, it should again be possible to perform the same masking operation as Schuchardt et al. (2021) by using $\sigma \to \infty$ and $\lambda \to \infty$.

Finally, it should be noted that the certificate by Schuchardt et al. (2021) allows for additional constraints, e.g. on the adversarial budget per node or the number of nodes controlled by the adversary. As all of them can be modelled using linear constraints on the budget variables (see Section C of their paper), they can be just as easily integrated into our mixed-integer linear programming certificate.

# E  EXPERIMENTAL SETUP AND ADDITIONAL EXPERIMENTS

In the following, we first explain the metrics we use for measuring the strength of certificates, and how they can be applied to the different types of randomized smoothing certificates used in our experiments. We then discuss the experimental setup, as well hyperparameters and additional experimental results for our semantic segmentation Section E.2 and node classification Section 8.2 experiments.

## E.1  METRICS

We use two metrics for measuring certificate strength: Certified accuracy (i.e. the percentage of correct and certifiably robust predictions) and certified ratio (i.e. the percentage of certifiably robust predictions, regardless of correctness). We use Monte Carlo randomized smoothing (see Section C. Therefore, we may have to abstain from making predictions. Abstentions are counted as non-robust and incorrect. In the case of center smoothing, either all or none predictions abstain (this is inherent to the method. In our experiments, center smoothing never abstained). In the following, let $\mathbb{Z} = \{d \in \{1, \ldots, D_{\text{out}}\} \mid y_n = \hat{y}_n\}$ be the indices of correct predictions, given an input $\boldsymbol{x}$.

**Naïve collective certificate**. The naïve collective certificate certifies each prediction independently. Like with our base certificates, let $\mathbb{H}^{(n)}$ be the set of perturbed inputs $y_n$ is robust to. Let $\mathbb{B}_{\boldsymbol{x}}$ be the collective perturbation model. Then $\mathbb{L} = \{d \in \{1, \ldots, D_{\text{out}}\} \mid \mathbb{B}_x \subseteq \mathbb{H}^{(n)}\}$ is the set of all certifiably robust predictions. The certified ratio can be computed as $\frac{|\mathbb{L}|}{D_{\text{out}}}$ and the certified accuracy as $\frac{|\mathbb{L} \cap \mathbb{Z}|}{D_{\text{out}}}$.

**Center smoothing** Center smoothing does not determine which predictions are robust, but only how many are. Let $k$ be the number of certifiably robust predictions. The certified ratio can be easily

computed as $\frac{k}{D_{\text{out}}}$. For the certified accuracy, we have to make the worst-case assumption that the non-robust predictions are correct predictions, i.e. compute $\frac{\max(0, k - (D_{\text{out}} - |\mathbb{Z}|))}{D_{\text{out}}}$

**Collective certificate**. Let $l(\mathbb{T})$ be the optimal value of our collective certificate and set of targeted nodes $\mathbb{T}$. Then both the certified ratio and certified accuracy can be computed via $\frac{l(\mathbb{T})}{D_{\text{out}}}$ with $\mathbb{T} = \{1, \ldots, D_{\text{out}}\}$ and $\mathbb{T} = \mathbb{Z}$, respectively.

### E.2 SEMANTIC SEGMENTATION

We first list all parameters and details of our training and certification procedures, before discussing additional experimental results in Section E.2.3.

#### E.2.1 EXPERIMENTAL SETUP AND HYPERPARAMETERS

**Model.** As the base model for the semantic segmentation task, we use a U-Net model (Ronneberger et al., 2015) with a ResNet-18 (He et al., 2016) backbone, as implemented by the Pytorch Segmentation Models library (version 0.13) (Yakubovskiy, 2020). We use the library's default parameters. In particular, the inputs to the U-Net segmentation head are the features of the ResNet model after the first convolutional layer and after each ResNet block (i.e. after every fourth of the subsequent layers). The U-Net segmentation head uses (starting with the original resolution) 16, 32, 64, 128 and 256 convolutional filters for processing the features at the different scales. To avoid dimension mismatches in the segmentation head, all input images are zero-padded to a height and width that is the next multiple of 32.

**Data and preprocessing.** We evaluate our certificates on the Pascal-VOC 2012 segmentation validation set. We do not use the test set, because evaluating metrics like the certified accuracy requires access to the ground-truth labels. For training, we use the 10582 Pascal segmentation masks extracted from the SBD dataset (Hariharan et al., 2011) (referred to as "Pascal trainaug" or "Pascal augmented training set" in other papers). SBD uses a different data split than the official Pascal-VOC 2012 segmentation dataset. We avoid data leakage by removing all training images that appear in the validation set. We downscale both the training and the validation images and ground-truth masks to 50% of their original height and width, so that we can use larger batch sizes and thus use our compute time to more thoroughly evaluate a larger range of different smoothing distributions. The segmentation masks are downscaled using nearest-neighbor interpolation, the images are downscaled using the INTER_AREA operation implemented in OpenCV (Bradski, 2000).

**Training and data augmentation.** We initialize our model weights using the weights provided by the Pytorch Segmentation Models library, which were obtained by pre-training on ImageNet. We train our models for 256 epochs, using Dice loss with batch size 64 and Adam($lr = 0.001, \beta_1 = 0.9, \beta_2 = 0.999, \epsilon = 10^{-8}, \text{weight\_decay} = 0$. Every 8 epochs, we compute the mean IOU on the validation set. After training, we use the model that achieved the highest validation mean IOU. We apply the following train-time augmentations: Each image is randomly shifted by up to 10% of its height and width, scaled by a factor from $[0.5, 2.0]$ and rotated between $-10$ and $10$ degrees using the ShiftScaleRotate augmentation implemented by the Albumentations library (version 0.5.2) (Buslaev et al., 2020). The images are than cropped to a fixed size of $128 \times 160$. Where necessary, the images are padded with zeros. Padded parts of the segmentation mask are ignored during training. After these operations, each input is randomly perturbed using a Gaussian distribution with a fixed standard deviation $\sigma \in \{0, 0.01, \ldots, 0.5\}$. All samples are clipped to $[0, 1]$ to retain valid RGB-images.

**Certification.** We evaluate all certificates on the first 50 images from the validation set that – after downscaling – have a resolution of $166 \times 250$. For our experiments, we use Monte Carlo randomized smoothing (see discussion in Section C). We use 12288 samples for making smoothed predictions and 153600 samples for certification. We use the significance parameter $\alpha$ to 0.01, i.e. all certificates hold with probability 0.99. For the center smoothing baseline, we use the default parameters suggested by the authors ($\Delta = 0.05, \beta = 2, \alpha_1 = \alpha_2$). For the naïve collective certificate baseline, we use Holm correction to account for the multiple comparisons problem, which yields strictly better results than Bonferroni correction. For our localized smoothing certificates, we use Bonferroni correction. For our localized smoothing distribution, we partition the input image into a regular grid of size $4 \times 6$ and define minimum standard deviation $\sigma_{\min}$ and maximum standard deviation $\sigma_{\max}$.

Let $\mathbb{J}^{(k,l)}$ be the set of all pixel coordinates in grid cell $(k,l)$. To smooth outputs in grid cell $(i,j)$, we use a smoothing distribution $\mathcal{N}(\mathbf{0}, \mathrm{diag}(\boldsymbol{\sigma})$ with, $\forall k \in \{1, \ldots, 4\}, l \in \{1, \ldots, 6\}, d \in \mathbb{J}^{(k,l)}$,

$$\sigma_d = \sigma_{\min} + (\sigma_{\max} - \sigma_{\min}) \cdot \frac{\max(|i-k|, |l-j|)}{6}, \tag{139}$$

i.e. we linearly interpolate between $\sigma_{\min}$ and $\sigma_{\max}$ based on the $l_\infty$ distance of grid cells $(i,j)$ and $(k,l)$. All results are reported for the relaxed linear programming formulation of our collective certificate (see Section A.4). For each grid cell, we use $\frac{1}{24} = \frac{1}{4\cdot 6}$ of the samples, which corresponds to 512 samples for prediction and 6400 samples for certification. The collective linear program is solved using MOSEK (version 9.2.46) (MOSEK ApS, 2019) through the CVXPY interface (version 1.1.13) (Diamond & Boyd, 2016).

### E.2.2 HARDWARE

All experiments on image segmentation were performed using a Xeon E5-2630 v4 CPU @ 2.20GHz, an NVIDA GTX 1080TI GPU and 128 GB of RAM.

### E.2.3 COMPLETE EXPERIMENTAL EVALUATION

As explained in Section 8.1, the objective of our experiments on semantic segmentation is to verify our claim that localized smoothing allows for a better accuracy-robustness trade-off than randomized smoothing with i.i.d. distributions, i.e. it allows us to retain a higher prediction quality and simultaneously provide stronger collective robustness guarantees.

The first step of our evaluation procedure is to evaluate the naïve and the center smoothing baselines for Gaussian standard deviations $\sigma \in \{0.1, 0.2, 0.3, 0.4, 0.5\}$ and – for each $\sigma$ – find a combination of localized smoothing parameters $\sigma_{\min}, \sigma_{\mathrm{out}}$ that result in equal or higher accuracy and stronger certificates, as measured by certified accuracy and certified ratio. Figs. 5 to 8 show the certified accuracy and certified ratio curve for all 10 pairs of baseline and locally smoothed models (5 per baseline). The AUC of the certified ratio and certified ratio curves for all smoothed models are summarized in Tables 1 and 2. Safe for center smoothing with $\sigma = 0.2$, which has a slightly higher certified ratio but lower accuracy, we can outperform all baselines.

The second step is to fix the found localized smoothing parameters and perform a fine-grained search over $\sigma \in [0, 1]$. This is to ensure that our previous results were not caused by a particularly poor choice of baseline parameters. Table 5 summarizes the results. It shows that there is no $\sigma$ in our search space for which the baselines outperform the locally smoothed model w.r.t. our metrics of certificate strength.

Figure Fig. 4 demonstrates why there is no $\sigma$ that outperforms our baselines, using the certified ratio of the naïve collective certificate baseline as an example. The certificate strength, as measured by the certified ratio's AUC, increases with $\sigma$. But the model's accuracy decreases with $\sigma$. Therefore, strengthening the certificate requires sacrificing some accuracy. But the locally smoothed models were already as accurate or more accurate than the baselines, which makes it impossible for the baselines to simultaneously attain a higher accuracy and stronger certificates. To summarize: Localized smoothing appears to indeed offer a better trade-off between certified accuracy and robustness.

In Section 8.1, we mentioned that we compare base classifiers with different trained weights. For i.i.d. smoothing with $\sigma$, we loaded models trained with $\sigma_{\mathrm{train}} = \sigma$. For localized smoothing, we loaded models trained with $\sigma_{\mathrm{train}} = \sigma_{\min}$. To show that this is not the reason why we outperform the baselines, we compare the same i.i.d. and localized smoothing distributions as before, but use $\sigma_{\mathrm{train}} = \sigma_{\min}$ for all models. The results are summarized in Appendix E.2.3 and Table 4. Overall, there is only a single instance in which the baselines are better than the localized smoothing certificate in any of our metrics: For $\sigma = 0.3$, the naïve baseline attains an accuracy of $79.5\%$ that is $0.4$ p.p. higher than that of the locally smoothed model. The AUC of its certified accuracy and certified ratio curves – $0.492$ and $0.5615$, respectively – are however much smaller than the $0.5869$ and $0.6579$ guaranteed by localized smoothing. So, even with this different choice of weights, localized smoothing appears to guarantee a better trade-off between accuracy and provable robustness.

While it resulted in significantly stronger certificates, the proposed collective certificate added very little additional computational cost, compared to the naïve collective certificate. The naïve collective certificate requires taking Monte Carlo samples and then performing a small number of vector

operations to compute the per-prediction Gaussian smoothing certificates. The proposed collective certificates requires taking Monte Carlo samples and computing the same type of per-prediction certificate, but then solves a linear program on top. The sampling dominated the runtime, with an average $460\,\text{s}$ per image (both for the baseline and the proposed certificate, as we used the same number of samples for both). Averaged over all experiments and images, computing the per-prediction certificates took $15.85\,\text{s}$ for the baseline and $0.15\,\text{s}$ for the proposed certificate (the increased cost for the baseline is due to the more complicated Holm correction – with Bonferroni correction both are equally fast). Averaged over all images and tested adversarial budgets, solving each collective linear program only took $0.68\,\text{s}$, i.e. much less than the Monte Carlo sampling that is necessary for both the baseline and our method.

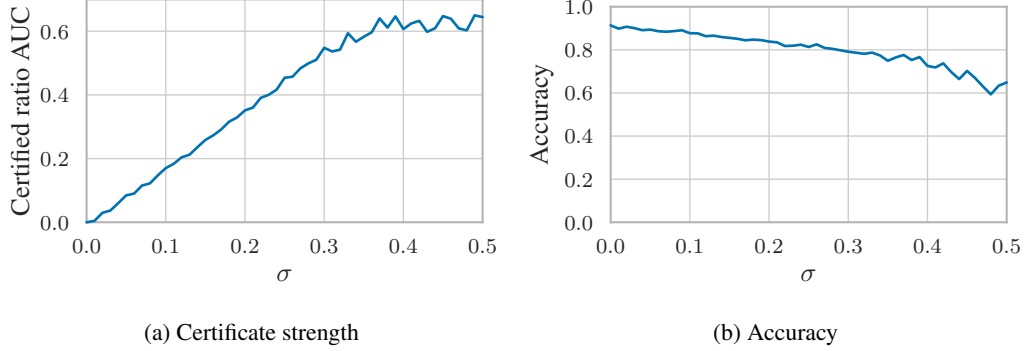

(a) Certificate strength         (b) Accuracy

Figure 4: Trade-off between accuracy and certifiably robustness at the example of the naïve collective certificate and certified ratio AUC. Increasing the standard deviation $\sigma$ strengthens the certificate, but decreases the accuracy.

| Naïve collective certificate | | | Localized smoothing | | | | |
|---|---|---|---|---|---|---|---|
| $\sigma$ | Acc. | AUC of Cert. Acc. | AUC of Cert. Ratio | $\sigma_{\min}$ | $\sigma_{\max}$ | Acc. | AUC of Cert. Acc. | AUC of Cert. Ratio |
| 0.1 | 87.7% | 0.1555 | 0.1702 | 0.055 | 0.28 | 87.7% | **0.1807** | **0.193** |
| 0.2 | 83.8% | 0.3127 | 0.3515 | 0.16 | 0.35 | **83.9%** | **0.3627** | **0.3925** |
| 0.3 | 79.1% | 0.4899 | 0.548 | 0.25 | 0.7 | 79.1% | **0.5869** | **0.6579** |
| 0.4 | 72.5% | 0.5426 | 0.6067 | 0.25 | 1.5 | **76.4%** | **0.7764** | **0.8999** |
| 0.5 | 64.9% | 0.5684 | 0.6444 | 0.3 | 1.5 | **74.8%** | **0.7143** | **0.8091** |

Table 1: Comparison of the naïve collective certificate baseline and localized smoothing for $\sigma \in \{0.1, 0.2, 0.3, 0.4, 0.5\}$ and different $\sigma_{\min}$ and $\sigma_{\max}$. In all cases, the localized smoothing certificate retains an equal or higher accuracy and simultaneously provides stronger robustness guarantees.

| Center smoothing | | | Localized smoothing | | | | |
|---|---|---|---|---|---|---|---|
| $\sigma$ | Acc. | AUC of Cert. Acc. | AUC of Cert. Ratio | $\sigma_{\min}$ | $\sigma_{\max}$ | Acc. | AUC of Cert. Acc. | AUC of Cert. Ratio |
| 0.1 | 88.3% | 0.1673 | 0.194 | 0.08 | 0.15 | **88.4%** | **0.1889** | **0.2004** |
| 0.2 | 84.7% | 0.281 | **0.3464** | 0.12 | 0.4 | **84.8%** | **0.3151** | 0.3427 |
| 0.3 | 80.3% | 0.3192 | 0.4339 | 0.2 | 0.6 | **80.5%** | **0.4935** | **0.5533** |
| 0.4 | 75% | 0.2316 | 0.3720 | 0.25 | 1.5 | **76.4%** | **0.7764** | **0.8999** |
| 0.5 | 68.3% | 0.1498 | 0.2914 | 0.25 | 1.5 | **76.4%** | **0.7764** | **0.8999** |

Table 2: Comparison of the center smoothing baseline and localized smoothing for $\sigma \in \{0.1, 0.2, 0.3, 0.4, 0.5\}$ and different $\sigma_{\min}$ and $\sigma_{\max}$. Safe for $\sigma = 0.2$, where the baseline has a slightly higher certified ratio AUC but a lower accuracy, localized smoothing retains an equal or higher accuracy and simultaneously provides stronger robustness guarantees.

| Naïve collective certificate | | | Localized smoothing | | | | |
|---|---|---|---|---|---|---|---|
| $\sigma$ | Acc. | AUC of Cert. Acc. | AUC of Cert. Ratio | $\sigma_{\min}$ | $\sigma_{\max}$ | Acc. | AUC of Cert. Acc. | AUC of Cert. Ratio |
| 0.1 | 86.9% | 0.1519 | 0.1686 | 0.055 | 0.28 | **87.7%** | **0.1807** | **0.193** |
| 0.2 | 83.8% | 0.3184 | 0.357 | 0.16 | 0.35 | **83.9%** | **0.3627** | **0.3925** |
| 0.3 | **79.5%** | 0.492 | 0.5615 | 0.25 | 0.7 | 79.1% | **0.5869** | **0.6579** |
| 0.4 | 73.5% | 0.6094 | 0.711 | 0.25 | 1.5 | **76.4%** | **0.7764** | **0.8999** |
| 0.5 | 72.5% | 0.7001 | 0.7981 | 0.3 | 1.5 | **74.8%** | **0.7143** | **0.8091** |

Table 3: Comparison of the naïve collective certificate baseline and localized smoothing for $\sigma \in \{0.1, 0.2, 0.3, 0.4, 0.5\}$ and different $\sigma_{\min}$ and $\sigma_{\max}$. Unlike in Table 1, we also load the model trained with $\sigma_{\min}$ for the baselines. Safe for $\sigma = 0.3$, where the baseline has a higher accuracy but much weaker certificates, localized smoothing again offers a higher accuracy and stronger certificates.

| Center smoothing | | | Localized smoothing | | | | |
|---|---|---|---|---|---|---|---|
| $\sigma$ | Acc. | AUC of Cert. Acc. | AUC of Cert. Ratio | $\sigma_{\min}$ | $\sigma_{\max}$ | Acc. | AUC of Cert. Acc. | AUC of Cert. Ratio |
| 0.1 | 88.3% | 0.1624 | 0.19 | 0.08 | 0.15 | **88.4%** | **0.1889** | **0.2004** |
| 0.2 | 83.9% | 0.268 | 0.3392 | 0.12 | 0.4 | **84.8%** | **0.3151** | **0.3427** |
| 0.3 | 80% | 0.3721 | 0.5009 | 0.2 | 0.6 | **80.5%** | **0.4935** | **0.5533** |
| 0.4 | 74.9% | 0.3494 | 0.5397 | 0.25 | 1.5 | **76.4%** | **0.7764** | **0.8999** |
| 0.5 | 62.9% | 0.1656 | 0.4212 | 0.25 | 1.5 | **76.4%** | **0.7764** | **0.8999** |

Table 4: Comparison of the center smoothing baseline and localized smoothing for $\sigma \in \{0.1, 0.2, 0.3, 0.4, 0.5\}$ and different $\sigma_{\min}$ and $\sigma_{\max}$. Unlike in Table 2, we also load the model trained with $\sigma_{\min}$ for the baselines. In all cases, the locally smoothed model has a higher accuracy and stronger certificates.

| Localized smoothing | | | | | Baselines | | |
|---|---|---|---|---|---|---|---|
| $\sigma_{\min}$ | $\sigma_{\max}$ | Acc. | AUC of Cert. Acc. | AUC of Cert. Ratio | Baseline | Best AUC of Cert. Acc | Best AUC of Cert. Ratio |
| 0.055 | 0.28 | 87.7% | **0.1807** | **0.193** | | 0.1555 | 0.1702 |
| 0.16 | 0.35 | 83.9% | **0.3627** | **0.3925** | Naïve | 0.296 | 0.3293 |
| 0.25 | 0.7 | 79.1% | **0.5869** | **0.6579** | Collective | 0.4899 | 0.548 |
| 0.25 | 1.5 | 76.4% | **0.7764** | **0.8999** | Certificate | 0.5753 | 0.6465 |
| 0.3 | 1.5 | 74.8% | **0.7143** | **0.8091** | | 0.5753 | 0.6465 |
| 0.08 | 0.15 | 88.4% | **0.1889** | **0.2004** | | 0.1488 | 0.1704 |
| 0.12 | 0.4 | 84.8% | **0.3151** | **0.3427** | Center | 0.2715 | 0.3286 |
| 0.2 | 0.6 | 80.5% | **0.4935** | **0.5533** | Smoothing | 0.3306 | 0.4255 |
| 0.25 | 1.5 | 76.4% | **0.7764** | **0.8999** | | 0.3538 | 0.5029 |

Table 5: Optimizing baseline standard deviation $\sigma$. For the same $(\sigma_{\min}, \sigma_{\max})$ as in Tables 1 and 2, we perform a grid search for $\sigma \in [0, 0.5]$ that yield certified accuracy or certified ratio curves with higher AUC, and an accuracy that at least equals the locally smoothed models. Such a $\sigma$ could not be found. Localized smoothing appears to offer a better accuracy-robustness trade-off.

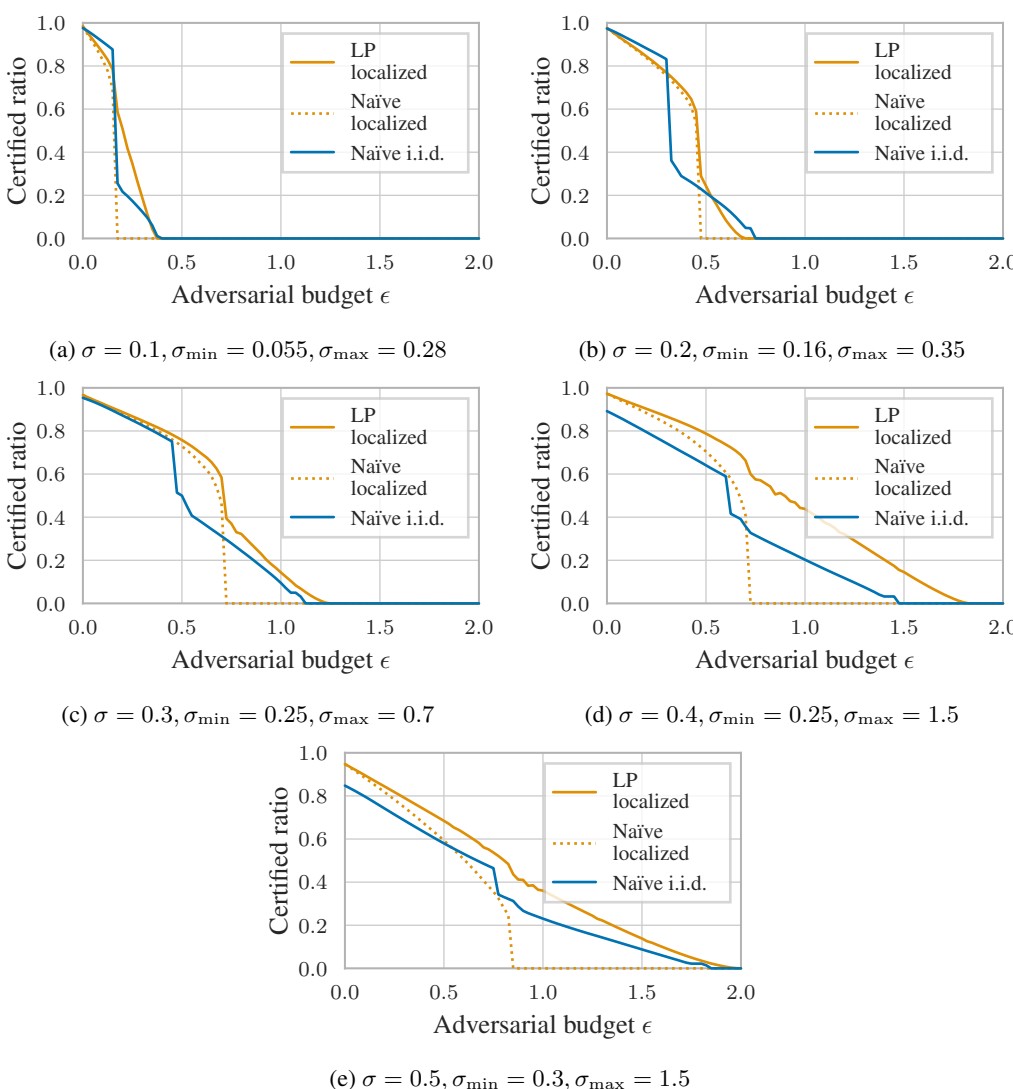

(a) $\sigma = 0.1, \sigma_{\min} = 0.055, \sigma_{\max} = 0.28$

(b) $\sigma = 0.2, \sigma_{\min} = 0.16, \sigma_{\max} = 0.35$

(c) $\sigma = 0.3, \sigma_{\min} = 0.25, \sigma_{\max} = 0.7$

(d) $\sigma = 0.4, \sigma_{\min} = 0.25, \sigma_{\max} = 1.5$

(e) $\sigma = 0.5, \sigma_{\min} = 0.3, \sigma_{\max} = 1.5$

Figure 5: Certified ratios of U-Net models under varying adversarial budgets $\epsilon$. We compare the naïve i.i.d. smoothing baseline (with standard deviation $\sigma$) to localized smoothing (with parameters $\sigma_{\min}, \sigma_{\max}$ such that the locally smoothed model has a higher or equal accuracy). Combining the localized smoothing base certificates using the proposed linear program (solid orange line) instead of evaluating them independently (dotted orange line) yields stronger guarantees. For $\sigma \in \{0.3, 0.4, 0.5\}$, the localized smoothing certificate outperforms the baseline for all $\epsilon$.

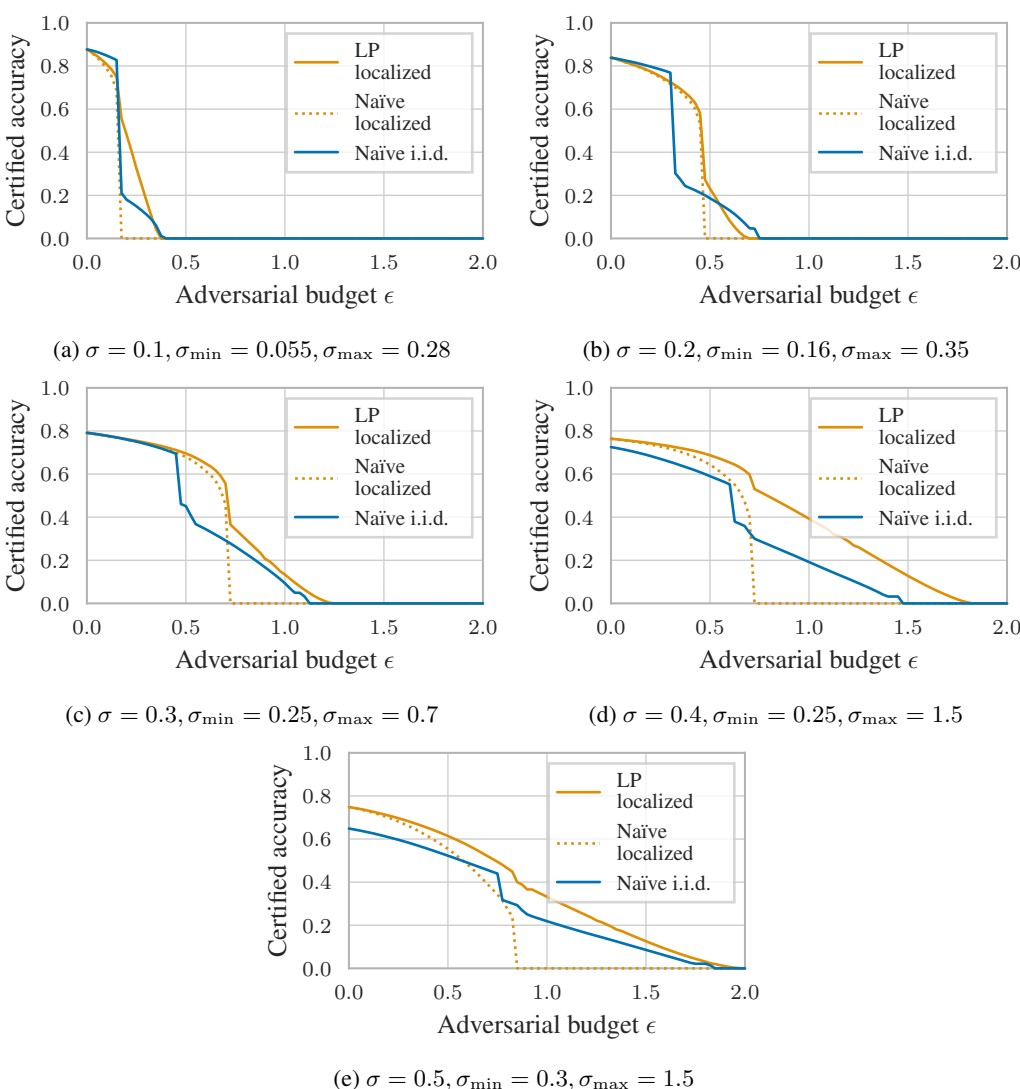

(a) $\sigma = 0.1, \sigma_{\min} = 0.055, \sigma_{\max} = 0.28$

(b) $\sigma = 0.2, \sigma_{\min} = 0.16, \sigma_{\max} = 0.35$

(c) $\sigma = 0.3, \sigma_{\min} = 0.25, \sigma_{\max} = 0.7$

(d) $\sigma = 0.4, \sigma_{\min} = 0.25, \sigma_{\max} = 1.5$

(e) $\sigma = 0.5, \sigma_{\min} = 0.3, \sigma_{\max} = 1.5$

Figure 6: Certified accuracies of U-Net models under varying adversarial budgets $\epsilon$. We compare the naïve collective certificate baseline (with standard deviation $\sigma$) to localized smoothing (with parameters $\sigma_{\min}, \sigma_{\max}$ such that the locally smoothed model has a higher or equal accuracy). Combining the localized smoothing base certificates using the proposed linear program (solid orange line) instead of evaluating them independently (dotted orange line) yields stronger guarantees. For $\sigma \in \{0.3, 0.4, 0.5\}$, the localized smoothing certificate outperforms the baseline for all $\epsilon$.

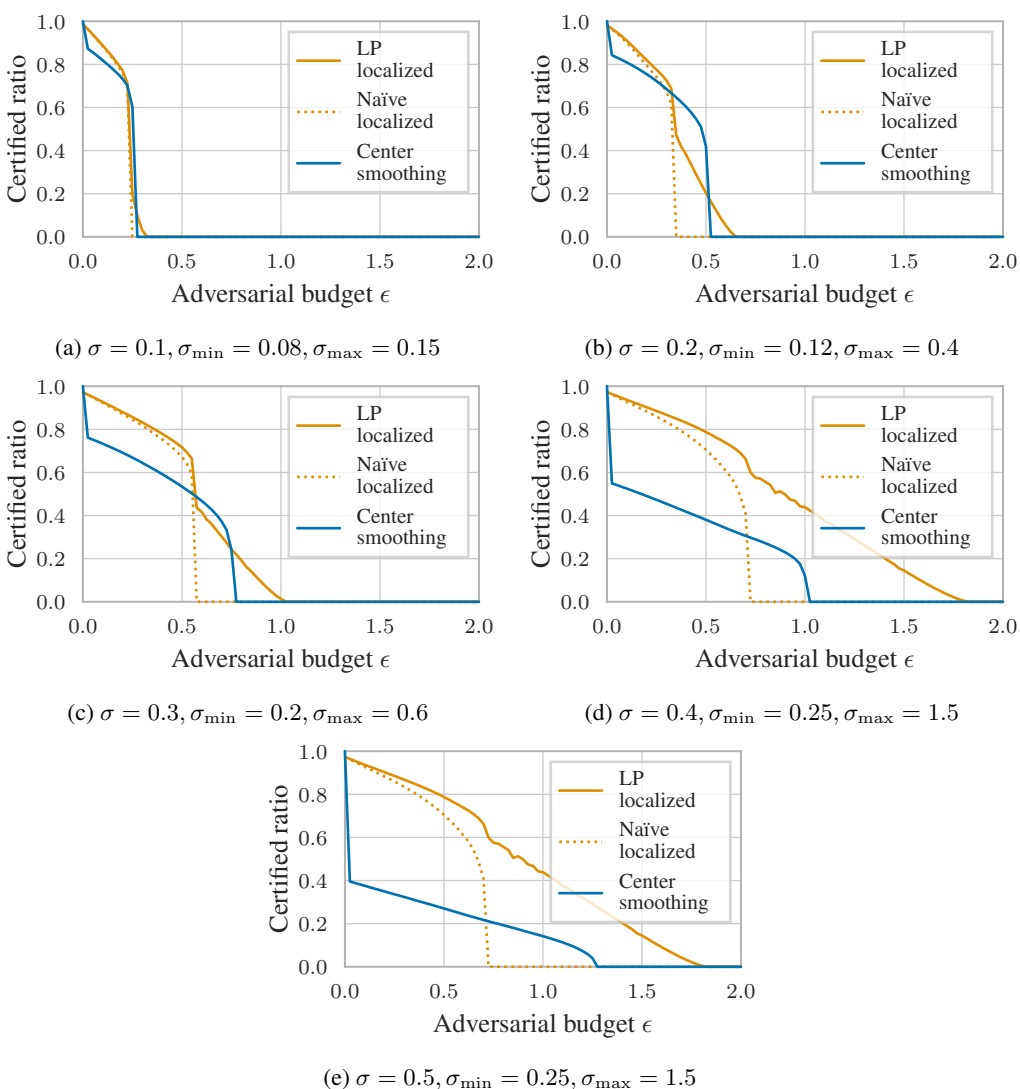

Figure 7: Certified ratios of U-Net models under varying adversarial budgets $\epsilon$. We compare the center smoothing baseline (with standard deviation $\sigma$) to localized smoothing (with parameters $\sigma_{\min}, \sigma_{\max}$ such that the locally smoothed model has a higher or equal accuracy). For $\epsilon = 0$, center smoothing has higher certified ratios, i.e. it abstains at a lower rate. For $\sigma = 0.2$, the center smoothing certified accuracy curve has a higher AUC than both the naïve combination of localized smoothing certificates (dotted line) and the proposed collective certificate (solid line). But for other $\sigma$ localized smoothing outperforms center smoothing. The gap widens with increasing $\sigma$.

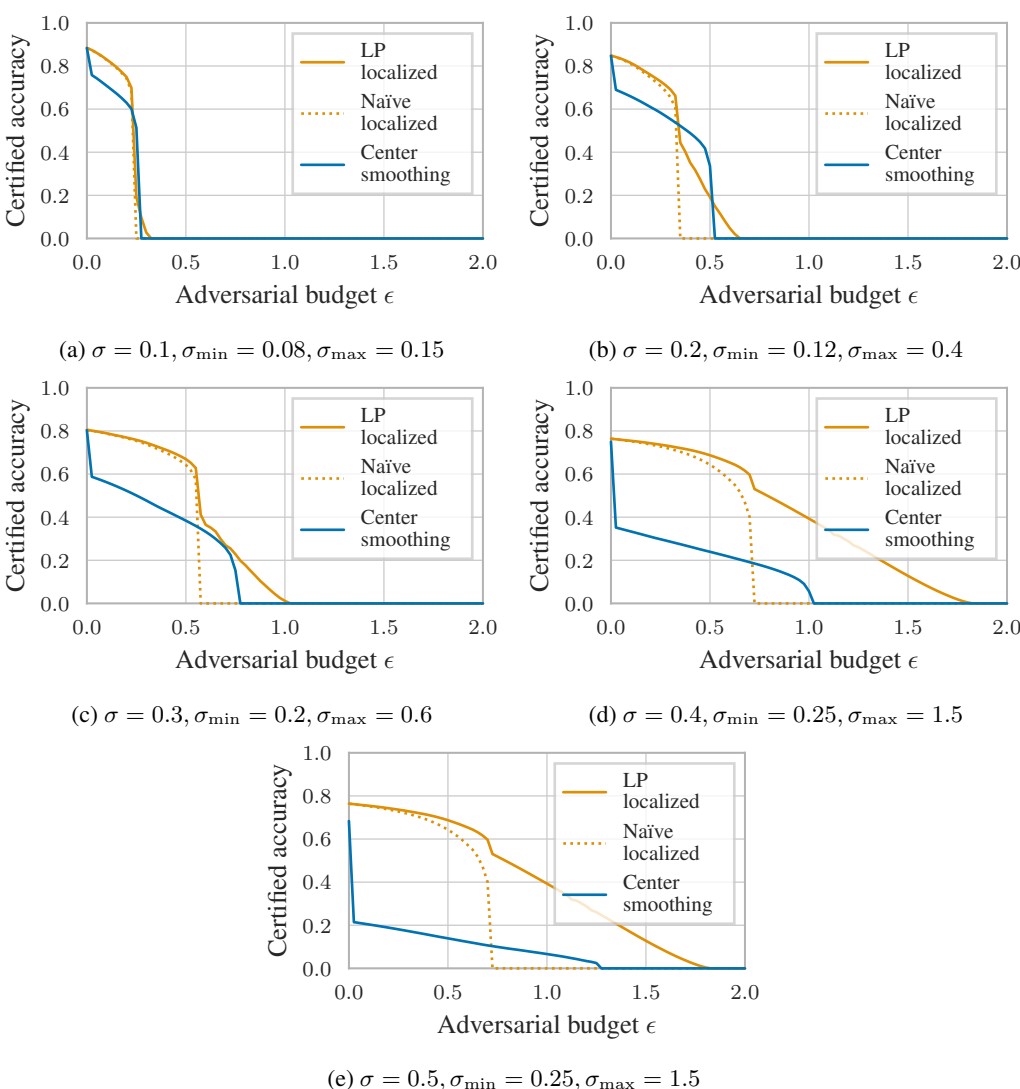

(a) $\sigma = 0.1, \sigma_{min} = 0.08, \sigma_{max} = 0.15$

(b) $\sigma = 0.2, \sigma_{min} = 0.12, \sigma_{max} = 0.4$

(c) $\sigma = 0.3, \sigma_{min} = 0.2, \sigma_{max} = 0.6$

(d) $\sigma = 0.4, \sigma_{min} = 0.25, \sigma_{max} = 1.5$

(e) $\sigma = 0.5, \sigma_{min} = 0.25, \sigma_{max} = 1.5$

Figure 8: Certified accuracies of U-Net models under varying adversarial budgets $\epsilon$. We compare the center smoothing baseline (with standard deviation $\sigma$) to localized smoothing (with parameters $\sigma_{min}, \sigma_{max}$ such that the locally smoothed model has a higher or equal accuracy). Combining the localized smoothing base certificates using the proposed linear program (solid orange line) instead of evaluating them independently (dotted orange line) yields stronger guarantees. The gap between center smoothing and localized smoothing widens with increasing $\sigma$.

### E.3  Node classification

We first discuss the experimental setup and the necessary parameters for our training and certification procedure before we show additional experimental results in Section E.3.3.

#### E.3.1  Experimental Setup and Hyperparameters

**Metric** In the node classification setting we are generally only concerned with certified accuracy. To calculate this accuracy, we only consider nodes that are correctly classified, where the method does not abstain, and is certifiably robust for the given perturbation.

**Model** We test two different models: 2-layer APPNP (Klicpera et al., 2019) and 6-layer GCN (Kipf & Welling, 2017). For both models we use a hidden size of $64$ and dropout with a probability of $0.5$. Furthermore, for the propagation step of APPNP we use $10$ for the number of interactions and $0.15$ as value for $\alpha$.

**Data and preprocessing.** We evaluate our approach on the Cora-ML node classification dataset. We perform standard preprocessing, i.e., remove self-loops, make the graph undirected and select the largest connected component. For the localized smoothing approach we perform Metis clustering (Karypis & Kumar, 1998) to partition the graph into $5$ clusters. We create an affinity ranking by counting the number of edges which are connecting cluster $i$ and $j$. This ranking is used to select the noise parameter for smoothing the attributes of cluster $j$ while classifying a node of cluster $i$.

**Training and data augmentation** All models are trained with a learning rate of $0.001$ and weight decay of $0.001$. The models we use for sparse smoothing are trained with the noise distribution that is also reported for certification. The localized smoothing models are trained on the their minimal noise level, i.e., not with localized noise but with only $\theta^+_{\min}$ and $\theta^-_{\min}$.

**Certification** In the node classification setting the noise parameter space is large. Instead of only one noise parameter as used in the image segmentation scenario, we can vary flip probabilities for addition and deletion. Apart from that we need lower and upper values for each of those. Therefore, we just selected the minimum parameters and focused on optimising the baseline around these noise levels as described in Section 8.2. The specific noise parameters and baseline search space regions can be seen in the captions of the respective figures. The collective linear program is solved using MOSEK (version 9.2.46) (MOSEK ApS, 2019) through the CVXPY interface (version 1.1.13) (Diamond & Boyd, 2016).

#### E.3.2  Hardware

All experiments on image segmentation were performed using an AMD EPYC 7543 CPU @ 2.80GHz, an NVIDA A100 GPU and $32\,$GB of RAM.

#### E.3.3  Additional Experimental Evaluation

In Fig. 9, we can see a comparison of our localized smoothing approach to sparse smoothing for a GCN model with 6-layers. Here, the smoothing distribution is the same as the one used in Fig. 3 of Section 8.2. In Fig. 10, we can see a comparison for an APPNP model but with smaller minimal noise levels.

We can see that in all experiments for deletion, the localized smoothing approach outperforms the baselines. In the deletion setting we increase the certified accuracy curve's AUC from an average $12.28$ to $18.93$ (i.e. by $54\%$). In the addition scenario the AUC, averaged over all experiments, decreases from $6.85$ to $6.08$ (i.e. by $11.25\%$). This is mainly due to the results of the experiment that can be seen in Fig. 10. In the baseline evaluation process both noise parameters for sparse smoothing are significantly larger than our min-values. This shows that our values may be too small in this scenario.

Fig. 11 compares the variance certificate to the sparse smoothing certificate of Bojchevski et al. (2020). That is, we use only one cluster and the same noise levels and models for our approach and the baseline, i.e., $\theta^+ = 0.01$ and $\theta^- = 0.6$. We observe that for both models the variance certificate yields better results for deletion. However, in the addition setting, it is outperformed by the baseline. If we look at Theorem 2, we can see that we multiply the adversarial budget for addition with $\gamma^+$

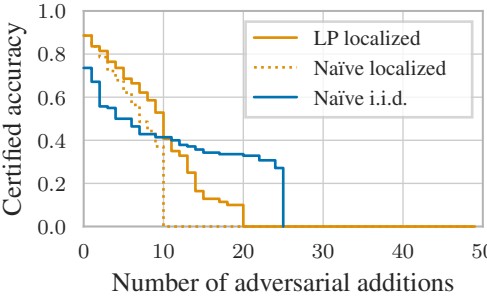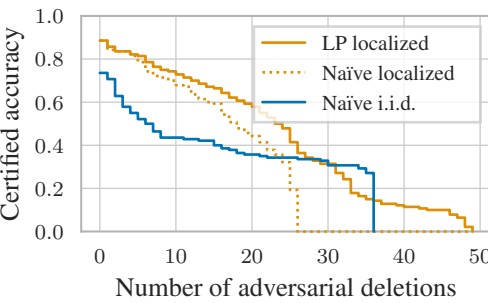

Figure 9: Certified accuracy for varying number of attribute additions (left) and deletions (right) for the GCN model. We compare localized smoothing ($\theta_{\min}^+ = 0.075$, $\theta_{\min}^- = 0.6$, $\theta_{\max}^+ = 0.15$, $\theta_{\max}^- = 0.95$) with sparse smoothing with ($\theta^+ = 0.085$, $\theta^- = 0.609$) for addition and deletion. This configuration yields the largest certified accuracy curve AUC of 10.41 for addition and 14.78 for deletion compared to all combinations $\theta^+ \in \{0.007, 0.0085, 0.01\}$ and $\theta^- \in [0.1, \dots, 0.827]$. In the deletion scenario we outperform the baseline with an AUC of 21.76 (15.76 non-collective). For addition the sparse smoothing performs slightly better than our certificate with AUC of 9.19 (6.39 non-collective). However, the our approach performs significantly better in small perturbation regions and also in clean accuracy (0 perturbation).

and the one for deletion with $\gamma^-$. As $\gamma^+$ has $\theta^+$ in the denominator the small noise level for addition results in larger values for $\gamma^+$. For these parameters we have $\gamma^+ = 2.795$ and $\gamma^- = 0.491$ which shows why the certificate is less robust to adversarial additions. This is consistent with the other experiments where our results are on par or worse than the baseline for additions but we outperform it for deletions.

The Monte Carlo sampling dominated the runtime. Averaged over all experiments, it took $1034\,\mathrm{s}$ (both for the baseline and the proposed certificate, as we used the same number of samples for both). Averaged over all experiments, the per-prediction certificates took $0.11\,\mathrm{s}$ for the baseline and $0.66\,\mathrm{s}$ for the proposed certificate (note that the baseline uses the sparsity-aware smoothing certificate of Bojchevski et al. (2020), while the proposed collective certificate uses our novel variance smoothing certificate (see Theorem 2), which requires estimating both the mean and variance of softmax scores). Averaged over all tested adversarial budgets, solving each collective linear program only took $10.9\,\mathrm{s}$, i.e. less than the Monte Carlo sampling that is necessary for both the baseline and our method. The reason that the linear programs for graphs required more time than those for image, even though they involve fewer constraints and variables, is that a different, not as well-vectorized implementation was used.

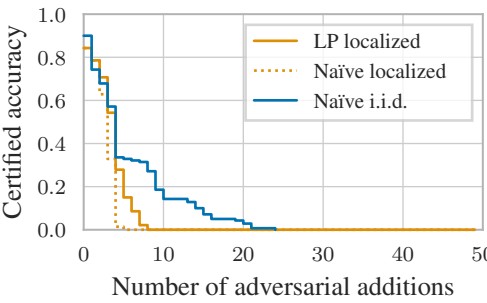 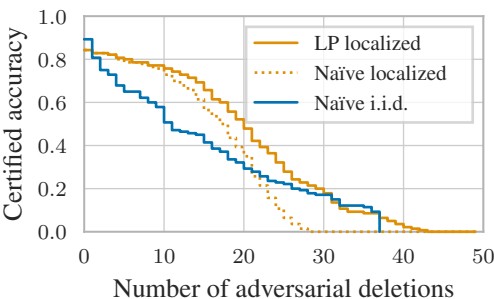

Figure 10: Certified accuracy for varying number of attribute additions (left) and deletions (right) for the APPNP model. We compare localized smoothing ($\theta^+_{\min} = 0.0075$, $\theta^-_{\min} = 0.65$, $\theta^+_{\max} = 0.08$, $\theta^-_{\max} = 0.95$) with sparse smoothing with ($\theta^+ = 0.0085$, $\theta^- = 0.827$) for addition and ($\theta^+ = 0.007$, $\theta^- = 0.755$) for deletion. These configurations yield the largest certified accuracy curve AUC of 5.62 for addition and 14.29 for deletion compared to all combinations $\theta^+ \in \{0.007, 0.0085, 0.01\}$ and $\theta^- \in [0.1, \ldots, 0.827]$. In the deletion scenario we outperform the baseline with an AUC of 18.76 (14.86 non-collective). However, the sparse smoothing performs better for addition than our certificate which only yields a AUC of 3.39 (2.54 non-collective). We observe that the optimal parameters for addition are both significantly larger than our minimal ones.

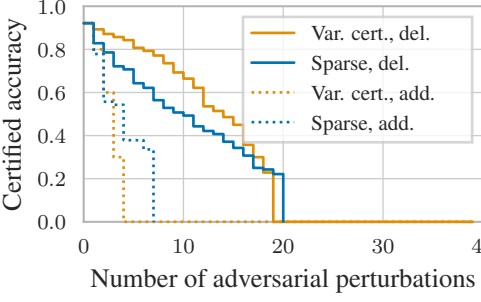 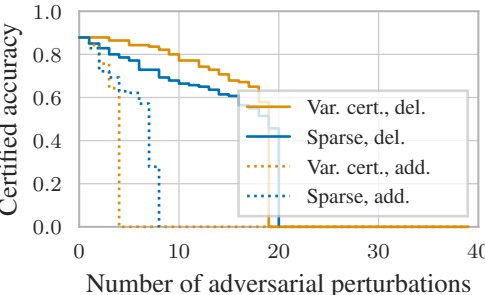

Figure 11: Comparison of the variance certificate with sparse smoothing. In this approach we only used one cluster, i.e., use the same noise levels. Left we can see the results for an APPNP model and on the right for a 6-layer GCN. The models are trained with the same noise parameters $\theta^+ = 0.01$ and $\theta^- = 0.6$. We observe that in for both models the variance certificate yields better results for deletion. However, in the addition setting, it is outperformed by the baseline.

