# OpenReview forum: "Localized Randomized Smoothing for Collective Robustness Certification"
_ICLR.cc/2022/Conference — ICLR 2022 Submitted_

### Official Review · Reviewer_z9Ri · 2021-10-19

**Correctness:** 3
**Technical Novelty And Significance:** 2
**Empirical Novelty And Significance:** 2
**Recommendation:** 5
**Confidence:** 4

**Main Review:**

**Strengths**

I very much appreciate the extensive experiments in tuning in the baseline with several sigma to guarantee (1) best robust accuracy of baseline when compared to localized randomized smoothing (2) when comparing certified ratios, the accuracies are comparable. The paper's proposed methodology is very well written and the formulation is beautifully simple and intuitive.


**General Questions**

1. Why is Equation 1 not an equality? Is not the set H for every pixel output is exactly tight? Or is the assumption here that H can be an over-approximation to the true certified region?

2. In the case where H is given by the localized smoothing using Propositions 1 or 2, will this result in an exact equality for (1)?

3. Can the authors comment of the compleixty for solving (5) with the proposed splits in the inputs? How does the method compare against the Naive approach. I believe it should be much more expensive as it requires the computation of the lower bound to the probability of success and then followed by solving the high dimensional linear program.

4. There are no comparisons against Fischer et al on certifying semantic segmentation.

5. It is not clear to how does the Naive classifier in Fgiure 2 work. Is it simply to perform RS certification for every pixel independently and counting the number of certifiable functions with a shared sigma over all inputs? If so, it is not clear to me why does the dashed line perform worse than this Naive approach?

6. It is also not clear to me why does solving the linear program reformulation work better than direct RS with anisotropic certificates (dashed line vs solid orange). Since the linear program lower bounds the original objective should not it be providing a pessimistic certified accuracy compared to the dash orange? That is to say, it tends to produce more adversaries that flip predictions under the lower bound but does not necessarily flip prediction of the original binary objective?

7. This is related to (6). What is the key motivation/intuition behind believing that solving (3) is better than directly certifying the anisotropic certificates. This link is very much un clear to me.

8. One of the key weaknesses that I see so far is the small certification experiments. For example, on the segmentation task, the certified accuracy is reported only on 50 images of the validation dataset. Perhaps a different dataset experiments (Cityscapes) following Fischer et al would make the submission stronger.

9. It is not clear to me if showing state-of-the-art certification results is not the objective of the paper, what is the key objective/insight of this reformulation? Showing that localized RS has a better trade-off between accuracy and robustness is not sufficiently novel for several reasons. (1) The per pixel sigma certification was derived in prior art which the paper faithfully state. (2) The proposed method of nonshared sigma over all method (without linear relaxation) is expected to be better intuitively as it overparameterizes the smoothing distribution holding the global shared sigma RS as a special case. So, I do not follow: what are the key *new* insights that I am missing here (please correct me if I am wrong here). (3) The certification time is not reported which is expected to be much larger.

10. I generally enjoyed reading the variance smoothing part (section B.2 in the appendix). The authors' are to be commended on it. Can the authors comment on how much of improved radius can the incorporation of the variance bring compared to classical RS certification. Does it actually tighten the radius for other certificates beyond gaussian? If so, this should be highlighted more in the paper as I believe this is significant. If not, then why doing variance smoothing and not directly with the expectation fixed and lower bounding over functions with a fixed expected prediction with Bernoulli distribution? So in short, why variance smoothing?



**Minor comments**
1. Page 3, line 9: "be the probability of g y" >> "of predicting y under g".
2. Caption of Figure 2. Sigma_min is given two values, I believe the second should be Sigma_max.
3. Page 8, line 11: sigma_min assigned twice.
3. Page 8, line 16: "Figure Fig. 2".
4. Proposition 4 in page 17. The radius shown by Eiras et al is without inverse CDF. It is only the difference in predictions between the top and runner up class.
5. Figure 1. I generally advise to have 4x6 grid in the figure aligning with the text. This is particularly the case as when Figure 1 was referenced in the experiments, it was after describing the 4x6 splits of the input.

**Summary Of The Paper:**

**Summary**

The paper proposes a localized smoothing approach for certifying structured output models. The threat model of interest in this setting is the bounded input perturbation that results in the highest number of prediction flips per pixel. The paper proposes to utilize the idea that when certifying pixel predictions at location (i,j), one can smooth the input pixels (k,l) far away from (i,j) by larger noise magnitude (standard deviation) resulting in a higher certified radii as they perhaps may not play a significant role in predicting the label of (i,j). The paper formulates the problem of finding worst cases adversaries that result in misprediction to the task of finding adversaries outside the certified region with anisotropic smoothing over the input. The paper then lower bounds the binary objective with a box constraints and solve the problem using linear  programs. Experiments are conducted on image segmentation tasks along with node classification.

**Summary Of The Review:**

See above.

---

> ### Author Response · Authors · 2021-11-18
> **Response Reviewer z9Ri**
>
> We would like to thank the Reviewer for their thorough review and the good questions which helped us in improving our submission.
> For the reviewers convenience, we have highlighted the most important modifications to our manuscript in blue.
> We have also made changes to address all of the reviewer's minor comments.
>
> To appropriately respond to the large number of very detailed questions, we have to split our response into multiple posts.
> Several questions (5, 6, 7, 9.2, ...) revolve around how our method is related to existing anisotropic smoothing methods and why it offers stronger robustness guarantees. We discuss this matter in our first post, before responding to individual questions.
>
> ### References
> [1] Cohen et al., Certified Adversarial Robustness via Randomized Smoothing, ICML 2019
> [2] Jan van den Brand, A deterministic linear program solver in current matrix multiplication time, SODA 2020
> [3] Fischer et al., Scalable Certified Segmentation via Randomized Smoothing, ICML 2021
> [4] Kumar et al., Certifying Confidence via Randomized Smoothing, NeurIps 2020

---

> > ### Author Response · Authors · 2021-11-18
> > **Response Reviewer z9Ri - Questions 9 & 10**
> >
> > **It is not clear to me if showing state-of-the-art certification results is not the objective of the paper, what is the key objective/insight of this reformulation? Showing that localized RS has a better trade-off between accuracy and robustness is not sufficiently novel for several reasons.**
> > 1. **The per pixel sigma certification was derived in prior art which the paper faithfully state.**
> > 2. **The proposed method of nonshared sigma over all method (without linear relaxation) is expected to be better intuitively as it overparameterizes the smoothing distribution holding the global shared sigma RS as a special case.**
> > 3. **The certification time is not reported which is expected to be much larger.**
> > **[So, I do not follow: what are the key new insights that I am missing here (please correct me if I am wrong here)]**
> >
> > That we were not interested in showing SOTA certification results is a misunderstanding. We have updated our perhaps misleading statement. What we meant to express is that any model and dataset is only a means of evaluating certificates. We therefore opted to use well-known and well-established architectures, instead of using the latest SOTA models (although our method, which is a black-box certificate, could in principle be applied to them).
> > We have updates this part of Section 8
> >
> > With this misunderstanding out of the way, let us respond to the sub-questions:
> >
> > (1) This is correct. The main contribution here is that we identify that multiple different “per-sigma” certificates, including novel base certificates derived by us, share a common interface (i.e. correspond to linear constraints in adversarial budget space). This allows us to use them as interchangeable components for our novel collective robustness certificate.
> >
> > (2) This is directly related to our response to Question 5. Using different $\sigma_d$ for different input dimensions will result in a certificate that guarantees more robustness for dimensions smoothed with larger $\sigma_d$.
> > For our collective perturbation model (i.e. $(|x - x’||_p \leq \epsilon$), this is however not beneficial.
> > As long as we consider predictions independently, the adversary can just allocate all of their adversarial budget to the input dimension smoothed with the smallest $\sigma_d$.
> > Using anisotropic smoothing to certify collective robustness is only beneficial because our novel method considers all per-prediction certificates collectively instead of evaluating them independently.
> >
> > (3) See Question 3 above (the computational overhead caused by the LP small)
> >
> > Now, to answer the question of “what are the key new insights here”:
> > The key insights are:
> > * Localized randomized smoothing, i.e. the idea of using different anisotropic distributions for smoothing different outputs, allows us to retain higher accuracy and provide stronger collective robustness guarantees than naively combining i.i.d. or anisotropic smoothing certificates.
> > * Identifying that both, existing single-prediction certificates for anisotropic smoothing and novel certificates derived in our submission, share a common interface, allowing us to use them interchangeably within our collective certification framework
> > * (Mixed-integer) linear programming can be used to compute our collective certificate.
> >
> > **I generally enjoyed reading the variance smoothing part (section B.2 in the appendix). [...] So in short, why variance smoothing?**
> >
> > We are delighted that the Reviewer enjoyed this section. Our primary motivation for using variance smoothing is its efficiency when applied to discrete data. Existing methods for randomized smoothing of discrete data could in principle be generalized to non-i.i.d. distributions, but would then have an exponential runtime (e.g. exponential in the number of distinct bit flip probabilities). Variance smoothing allows us to compute certificates in constant time.
> >
> > When the classifier’s softmax scores have low variance under the smoothing distribution, variance smoothing can indeed result in stronger  certificates (see for example Fig. 11 on the last page of the appendix, where it provides stronger guarantees against deletions of bits).
> > We would however like to point out that CDF-smoothing (Kumar et al., (2020) [4]) captures not only the mean and variance, but the entire distribution of softmax scores. It can thus provide even stronger guarantees. Generalizing CDF-smoothing to non-i.i.d. distributions on discrete data does however result in the same exponential blow-up as other certificates.
> >
> > Based on the reviewer’s  comment, we have revised our discussion in Section 5 to make this focus on efficiency clearer to future readers.

---

> > > ### Comment · Reviewer_z9Ri · 2021-11-23
> > > **Response**
> > >
> > > I would like to first thank the authors for a very thorough rebuttal. I have read their response to my concerns. I have also read the other reviews and the corresponding rebuttals.
> > >
> > > I will respond back with bullet points matching the numbering as my original post.
> > >
> > > **(1)** The response is very clear. The justification to why this in general not strict equality is crystal clear to me.
> > >
> > > **(2)** The answer to this question was not addressed properly. Since the certification result of Cohen et al with Gaussian noise is tight, considering any region outside the certification ball of Cohen et al there exists an adversary that flips prediction. Based on that, at least under Gaussian smoothing, should not this be exact equality?
> > >
> > > **(3)** I am aware of the general complexity of the problem, number of variables, and constraints. However, I was interested in knowing the exact practical run time to compare against certifying only with Naive randomized smoothing (per classifier iid certification) and Naive collective (per classifier anisotropic) to see how much overhead is there due to the optimization compared to baselines. The authors did report the total time to solve LP. However, can you further comment on what setup (experimental setup) is that 0.67 seconds reported on. This should be clearly shown in the paper. Also, can you report the total certification time for your method, Naive, and collective Naive?
> > >
> > > **(4)** I understand. This means that Fischer et al provides a stronger sense of certification requiring that all classifiers be certifiable. If that is the case, motivation is needed as to why it is more useful to look at the sum of the number of flips over classifiers? A good discussion on this along with a clear difference between the two methods should be stated (as done in the rebuttal) to justify why that other method is not comparable and why it makes more sense to certify the sum of classifiers as opposed to them all at once.
> > >
> > >
> > > **(5,6,7)** I thank the authors for their very well thought rebuttal explaining the difference between Naive, Naive collective, and their method. This was not clear from the text (perhaps not even mentioned) at all. Based on that, it does seem to be that Naive and Naive collective do not fairly compare against the proposed method. This is since for each output function the same $\epsilon$ is provided to flip predictions while in the proposed method there is one global $\epsilon$ budget to flip all classes. This puts Naive and Naive Collective at an edge for the adversary, as is it is easier for adversaries to attack Naive and Naive collective as the budget is n times bigger (n is the number of outputs, i.e. pixels). I believe a better fair comparison is to compare your method with any Naive variant with a budget of $n\epsilon$. This will definitely highlight the real benefit of doing the collective certification (as opposed to Naive).
> > >
> > > **(8)** While I see the authors' arguments in regards to the experiments, I still believe it is very necessary to understanding that this is not a lucky consequence on a single dataset. I believe given the similarity of the setup and requiring to only changing the dataset, I do not see any strong reason to why this can not be done to strengthing the paper.

---

> > > > ### Author Response · Authors · 2021-11-24
> > > > **Follow-up, Reviewer z9Ri**
> > > >
> > > > We appreciate that the reviewer has spent additional time and effort to respond to our rebuttal.
> > > > Before discussing the different comments, we would first like to point out that some of the suggested changes (e.g. specifying hardware) were already implemented in our revision from November 18 (see changelog above). We have however made some additional modifications, as suggested by the reviewer.
> > > >
> > > > Due to the character limit, we again split our response into multiple comments.
> > > >
> > > >
> > > > ### References
> > > > [1] Cohen et al., Certified Adversarial Robustness via Randomized Smoothing, ICML 2019
> > > > [2] Lee et al., Tight Certificates of Adversarial Robustness for Randomly Smoothed Classifiers, NeurIPS 2019
> > > > [3] Fischer et al., Scalable Certified Segmentation via Randomized Smoothing, ICML 2021

---

> > > > > ### Author Response · Authors · 2021-11-24
> > > > > **Follow-up, Reviewer z9Ri, Questions 5-8**
> > > > >
> > > > > **(5, 6, 7). [...] it does seem to be that Naive and Naive collective do not fairly compare against the proposed method. This is since for each output function the same is provided to flip predictions while in the proposed method there is one global budget to flip all classes. This puts Naive and Naive Collective at an edge for the adversary, as is it is easier for adversaries to attack Naive and Naive collective as the budget is n times bigger (n is the number of outputs, i.e. pixels).**
> > > > >
> > > > > The reviewer is entirely correct in their observation that the naive certificates lead to very poor robustness guarantees, as they implicitly assume an adversary whose budget is $n$ times bigger (or to be more precise - an adversary that can craft $n$ different attacks using the same budget).
> > > > > But this is not a mistake in our evaluation procedure. Instead, it is a deficiency of naive collective robustness certificates and the exact reason why dedicated collective robustness certificates (like ours) are needed!
> > > > >
> > > > > Perhaps it is helpful to put the experiments in context of the remaining paper.
> > > > > Our goal is to develop an effective method for certifying collective robustness, i.e. lower-bound $\\min\_{x’ : ||x - x’||\_p \\leq \\epsilon} \\sum\_n \\mathrm{I} \\left[f_n(x') = y\_n\\right]$
> > > > > for a given adversarial budget $\epsilon$.
> > > > > We present different methods for computing this lower bound (center smoothing, the naive certificate, the proposed collective certificate …).
> > > > > The purpose of the experiments is to determine which of these certificates is most effective.
> > > > > That is, given a specific collective adversarial budget $\epsilon$, which of the methods can certify robustness for the largest number of predictions?
> > > > >
> > > > > Making the proposed comparison would not be fair and lead to incorrect results.
> > > > > It would imply that the naive collective certificate evaluated with budget $\epsilon / N$ was a valid certificate for the collective adversarial budget $\epsilon$ (afterall, comparing our valid certificate to an invalid one would neither be fair nor informative).
> > > > > A simple counterexample shows that the naive certificate evaluated with $\epsilon / N$ is in fact not a valid collective certificate for collective adversarial budget $\epsilon$. Assume that all outputs are the same function, i.e. $\forall x, n, m: f_n(x) = f_m(x)$.
> > > > > Further assume that this function is robust to perturbations of magnitude $\\leq 2 \cdot \epsilon / N$, but that there is an adversarial example $x’$ with $||x - x’||_p = 3 \cdot \epsilon / N$.
> > > > > Using the naive collective certificate with budget $\\epsilon / N$ would suggest that all predictions were robust (afterall, $\\epsilon / N < 2 \\cdot \\epsilon / N$). But we know that they can all be attacked using the same adversarial example $x’$ with magnitude  $3 \cdot \epsilon / N \leq \epsilon$, i.e. none of the predictions are actually robust and the computed certificate would not be valid for collective adversarial budget $\\epsilon$.
> > > > >
> > > > > **8.) While I see the authors' arguments in regards to the experiments, I still believe it is very necessary to understanding that this is not a lucky consequence on a single dataset. I believe given the similarity of the setup and requiring to only changing the dataset, I do not see any strong reason to why this can not be done to strengthing the paper.**
> > > > >
> > > > > We agree that, while it may not be necessary, there is no reason why we cannot perform experiments on additional datasets.
> > > > > The deadline for updating our manuscript has already passed.
> > > > > We do however plan to perform additional experiments on Cityscapes and alternative node classification datasets (e.g. CiteSeer, PubMed) before the camera-ready deadline.
> > > > >
> > > > > To conclusively show that our certificate offers a better accuracy-robustness trade-off than i.i.d. smoothing, we have to evaluate multiple combinations of anisotropic smoothing parameters and compare them to a wide range of i.i.d. smoothing parameters (like we did in our initial submission).
> > > > > This makes our experiments somewhat time-consuming to perform.
> > > > > Nevertheless, we hope to complete the experiments on the node classification datasets before the end of the final discussion phase, in which case we will try to summarize the results on OpenReview.

---

> > > > > > ### Comment · Reviewer_z9Ri · 2021-11-25
> > > > > > **Follow up**
> > > > > >
> > > > > > I thank the authors for their feedback and thorough prompt response.
> > > > > >
> > > > > > (2) This is very clear now. I believe what is meant with *not exact* is that the result is certification and *not* verification, i.e. a computed certified accuracy is a lower bound to the true verified accuracy as this is under the worst case smoothed classifier satisfying certain properties. I thank the authors' for this clarification and why their inequality even under Gaussian smoothing is not necessarily an equality. This addresses this concern very well.
> > > > > >
> > > > > > (3) This concern has also been addressed. I do believe however, the runtimes should be reported in the main paper as opposed to leaving them in the appendix.
> > > > > >
> > > > > > (4) I do like the argument on cancer segmentation and ad targeting on graphs to justifying why the adversary model proposed in this paper might be more relevant than the stronger notion of "all predictions should stay constant" of Fischer et al. I strongly suggest authors to be upfront about this. This was not clear in the main paper nor the differences between your model and Fischer et al's work was. Stating exactly the adversary in a small subsection before presenting your work along with examples to why such an adversary makes sense is extremely important for future readers not to get confused.
> > > > > >
> > > > > > (5-7) I do not think it is unfair. As per the given counter example, it says that every function is not certified if attacked with $3\epsilon/N$. In this setup, the total budget will be $3\epsilon$ which is already better certified than the collective certificate with budget $\epsilon$. The expectation is that (if functions are different) each function will have a different certificate budget and the constraint is that the sum adds up to at most $\epsilon$. Showing with your method that looking at all classifiers jointly when designing this effective $\epsilon$ perturbations is better or even similar to a method that designs the adversaries for classifiers independently is more interesting and informative. In fact, it says that the proposed method can squeeze more certification out by looking at the joint behavior of all classifiers. Currently, as it stands, the Naive certificate is by construction worse than the collective certificate and it is no surprise that the proposed method is better making this result uninformative.
> > > > > >
> > > > > > (8) I think it is necessary to do such experiments for two reasons. (1) The paper is proposing a new different collective adversary. (2) The comparisons are only against naive baselines. If this were to be a standard problem, working in a classical notion of additive adversarial perturbations, showing comparisons against SOTA on few datasets can be enough. However, if the adversary setting is new and thus the baselines are all Naive, more experiments are important.
> > > > > >
> > > > > > I am willing to increase by score based on the discussions so far and later coming discussions

---

> > > > > > > ### Author Response · Authors · 2021-11-26
> > > > > > > **Follow-up 2, Reviewer z9Ri**
> > > > > > >
> > > > > > > We want to thank the reviewer for the effort invested in our paper and the fruitful discussion, thanks to which we could already strongly improve our submission.
> > > > > > >
> > > > > > > It appears like everything but the second to last comment (5-7) has been resolved.
> > > > > > > We agree with the suggested improvements in (3), (4) and (8) and will incorporate them into our camera-ready version, on top of the changes we have already made last week.
> > > > > > >
> > > > > > > Now, let us address the final open point (5-7):
> > > > > > > First, we would like to present an alternative view on why the suggested experiment of comparing the proposed certificate with budget $N \\cdot \\epsilon$ to the naive baseline with budget $\\epsilon$ (or equivalently: the proposed method with budget $\\epsilon$ to the naive baseline with budget $\\epsilon / N$) would not be well-founded.
> > > > > > > After that (in our next comment), we would like to discuss what appears to be the actual underlying issue that has led us into this whole discussion.
> > > > > > >
> > > > > > > Now, to go back to the suggested experiment: Our above counter-example shows that there are classifiers and per-prediction certificates for whích
> > > > > > > $$\\min\_{x’ : ||x’ - x|| \leq \\epsilon} \\sum\_{n=1}^N \\mathrm{I} \\left[f_n(x’) = y_n \\right]
> > > > > > > \\lneq
> > > > > > > \\sum\_{n=1}^N \\mathrm{I}
> > > > > > >  \\min\_{x’ : ||x’ - x|| \leq \\epsilon / N} \\left[x’ \\in \\mathrm{H}^{(n)} \\right].
> > > > > > > $$
> > > > > > > That is: Even the best possible collective certificate, which can determine exactly how many predictions are simultaneously robust, would potentially be “outperformed” by the proposed procedure. This cannot possibly be a reasonable standard to apply to any valid robustness certificate.
> > > > > > > We do not even need to construct a specific counter example. For very large $\\epsilon$, the l.h.s. term will likely be either very small or zero (the adversary can essentially perturb the input as they please). But for $N \\to \\infty$ (or just very large $N$), all of the base certificates will be evaluated with an adversarial budget $\\epsilon / N$ that is close to $0$, i.e. the r.h.s. function will have value $N$ and thus falsely certify robustness for all predictions.
> > > > > > >
> > > > > > > We do agree with the reviewer that outperforming even this upper bound on the naive certificate would further emphasize the effectiveness of our proposed method. But, since this upper bound is not a valid certificate, we may as well compare our method to any other arbitrary upper bound on the naive certificate (or even better: measure the difference in certificate strength between our certificate and the valid naive certificate, which we do in our experiments).

---

> > > > > > > > ### Author Response · Authors · 2021-11-26
> > > > > > > > **Follow-up 2, Reviewer z9Ri, High-level problem**
> > > > > > > >
> > > > > > > > The actual problem, which has led us into this discussion, appears to be summarized in the reviewer’s last sentence in (5-7):
> > > > > > > >
> > > > > > > > **Currently, as it stands, the Naive certificate is by construction worse than the collective certificate and it is no surprise that the proposed method is better making this result uninformative.**
> > > > > > > >
> > > > > > > > This is actually not the case! Yes, the general formulation of our collective robustness certificate will always be greater *or equal* to the naive certificate (we discussed this last week, in our original response to comment (6)).
> > > > > > > > But with most base certificates, we will have an equality, i.e. the collective approach that considers a single perturbed input and the naive approach that considers $N$ different inputs will be equally bad.
> > > > > > > >
> > > > > > > > Most robustness certificates (including randomized smoothing with i.i.d. noise) certify robustness to norm-bound perturbations, i.e.
> > > > > > > > $\\mathbb{H}^{(n)} = \\{x’ \\mid ||x - x’|| \\leq c_n \\}$ for some constant $c_n$.
> > > > > > > > If we use such certificates as per-prediction certificates / base certificates, we have:
> > > > > > > > $$
> > > > > > > >  \\min\_{x’ : ||x’ - x|| \leq \\epsilon}
> > > > > > > > \\sum\_{n=1}^N
> > > > > > > > \\mathrm{I} \\left[x’ \\in \\mathrm{H}^{(n)} \\right]
> > > > > > > > =
> > > > > > > >  \\min\_{x’ : ||x’ - x|| \leq \\epsilon}
> > > > > > > > \\sum\_{n=1}^N
> > > > > > > > \\mathrm{I} \\left[||x - x’|| \\leq c_n  \\right]
> > > > > > > > =
> > > > > > > > \\sum\_{n=1}^N
> > > > > > > > \\mathrm{I} \\left[\\epsilon \\leq c_n  \\right]
> > > > > > > > =
> > > > > > > > \\sum\_{n=1}^N
> > > > > > > >  \\min\_{x’ : ||x’ - x|| \leq \\epsilon}
> > > > > > > > \\mathrm{I} \\left[||x - x’|| \\leq c_n  \\right]
> > > > > > > > =
> > > > > > > > \\sum\_{n=1}^N
> > > > > > > >  \\min\_{x’ : ||x’ - x|| \leq \\epsilon}
> > > > > > > > \\mathrm{I} \\left[x’ \\in \\mathrm{H}^{(n)} \\right].
> > > > > > > > $$
> > > > > > > > In the first equality, we just substitute the definition of the base certificates. The second equality is due to the fact that the optimal solution for the collective problem is to just pick any $x’$ with $||x-x’|| \\leq \\epsilon$. The third equality is because this is also the optimal solution to the naive problem, where each indicator function is optimized independently. In the last equality, we again use the definition of the base certificates.
> > > > > > > > The same argument also holds when each output is smoothed with the same anisotropic smoothing distribution: The optimal solution for both the collective and naive problem is to just assign all adversarial to the input dimension with the lowest noise level (we also discussed this in our original response to comment (6) last week).
> > > > > > > >
> > > > > > > > Our collective certificate is only better than the naive certificate because we use our novel localized randomized smoothing approach, in which each output is smoothed using a different anisotropic smoothing distribution.
> > > > > > > >
> > > > > > > > This leads us to a very crucial detail:
> > > > > > > > The purpose of our experiments is to determine whether the collective certificate based on *localized smoothing* is better than the (naive) collective certificate based on *i.i.d. smoothing* (which, as our Equation above shows, is equivalent to the “proper” collective certificate based on i.i.d. smoothing).
> > > > > > > > This is not some trivial fact or necessarily true “by construction”.
> > > > > > > > Yes, the proper collective certificate based on localized smoothing will be better than the naive collective certificate based on localized smoothing.
> > > > > > > > But that is not what we want to show in our experiments (we only draw both graphs to demonstrate that the collective LP is indeed beneficial).
> > > > > > > > We also explicitly state this in the very first paragraph of our experimental evaluation:
> > > > > > > > “Verifying our main claim that *localized randomized* smoothing offers a better trade-off between accuracy and certifiable robustness than smoothing with *i.i.d. distributions*.”
> > > > > > > >
> > > > > > > > Finally, we would like to point out that the naive collective certificate is not some arbitrarily constructed function that is designed to perform poorly.
> > > > > > > > Instead, it is the only currently available option (aside from center smoothing) for certifying collective robustness for models that are not strictly local. And that is because -- so far -- robustness certification literature is almost exclusively focused on certificates for single-output models that can only be combined in this naive manner.
> > > > > > > >
> > > > > > > > To summarize:
> > > > > > > > 1. The “proper” collective certificate (Eq.1 in our paper) is in many cases equal to the naive collective certificate, in particular when using randomized smoothing with i.i.d. noise.
> > > > > > > > 2. Only because we use our novel localized smoothing approach is the proper collective certificate actually better than the naive certificate.
> > > > > > > > 3. Our experiments are not meant to show that the proper collective localized smoothing certificate is better than the naive localized smoothing certificate.
> > > > > > > > 4. Our experiments are meant to compare localized smoothing certificates to i.i.d. smoothing certificates.
> > > > > > > > 5. The naive collective certificate is not some arbitrarily constructed function. It is the best currently available option for certifying collective robustness.

---

> > > > > ### Author Response · Authors · 2021-11-24
> > > > > **Follow-up, Reviewer z9Ri, Questions 4**
> > > > >
> > > > > **4.) If that is the case, motivation is needed as to why it is more useful to look at the sum of the number of flips over classifiers? A good discussion on this along with a clear difference between the two methods should be stated (as done in the rebuttal) to justify why that other method is not comparable and why it makes more sense to certify the sum of classifiers as opposed to them all at once.**
> > > > >
> > > > > First, we would like to point out that Section C.4 of our revised manuscript from November 18 already features a discussion on the difference between (Fischer et al. (2021) [3]) and the naive collective certificate (and by extension the proposed collective certificate, which is an upper bound on the naive certificate).
> > > > > In essence: (Fischer et al. (2021), [3]) is the naive collective certificate, but with weaker pre-prediction certificates than the “tight” Gaussian smoothing certificates used in our experiments.
> > > > >
> > > > > We do not believe that one notion of collective robustness is necessarily better than the other. They capture different properties of the classifier, both of which can be relevant in different use cases.
> > > > > For example, when using a GNN to decide which ads to serve to users of a social network, it may be important that as many users as possible receive the correct ad, even under small changes of the network. But whether the served ad for one out of billions of users changes is not too relevant.
> > > > > But when using a segmentation model for cancer screening, it may be critical that no part of a tumor is missed.
> > > > >
> > > > > However, following the reviewer’s comment, we have extended Section C.4 to show the following:
> > > > > If one were interested in determining whether all (non-abstaining) predictions are robust, then applying our collective Gaussian smoothing certificate to all (non-abstaining) predictions and testing whether the number of certified predictions equals the number of (non-abstaining) predictions will be at least as strong as the certificate of Fischer et al. (2021) [3].
> > > > > Again, this is due to the fact that Fischer et al. (2021) [3] do nothing but independently certify  each prediction using a method that is at most as strong as the “tight” Gaussian smoothing certificate.
> > > > > We also reference this comparison in the updated version of our related work section.

---

> > > > > ### Author Response · Authors · 2021-11-24
> > > > > **Follow-up, Reviewer z9Ri, Question 3**
> > > > >
> > > > > **3.) [...] However, can you further comment on what setup (experimental setup) is that 0.67 seconds reported on. This should be clearly shown in the paper. Also, can you report the total certification time for your method, Naive, and collective Naive?**
> > > > >
> > > > > We agree with the reviewer that measuring runtime without specifying the used hardware is not very informative. Therefore, our revision from November 18 already specified the used hardware in Sections E2.2 and E3.2 (Xeon E5-2630 v4 CPU@2.20GHz, an NVIDA GTX 1080TI GPU and 128GB of RAM for the segmentation experiments and an AMD EPYC 7543 CPU @2.80GHz, an NVIDA A100 GPU and 32GB of RAM for the graph experiments).
> > > > >
> > > > > The naive certificates involve two steps:
> > > > > 1. Monte carlo sampling to obtain many predictions of the base classifier
> > > > > 2. Computing the per-prediction certificates.
> > > > >
> > > > > The time needed for computing these per-prediction certificates is negligible, as it only requires a single or few vector operations. Nevertheless, we have updated our submission to also include these numbers.
> > > > > The proposed collective certificate requires the same steps as above, followed by solving a linear program.
> > > > >
> > > > > For our image segmentation experiments, the average time needed to obtain all Monte Carlo samples for an image was $460s$ (we used 153600 samples for all certificates; using 10^5 - 10^6 samples is common in randomized smoothing literature).
> > > > > Computing the per-prediction certificates for the naive i.i.d. baseline took an average $15.85s$ per image.
> > > > > Computing the per-prediction certificates for the naive localized smoothing certificate took an average $0.15s$.
> > > > > The reason that the i.i.d. certificates took longer to compute is that we used the more effective, but also more complicated, Holm correction instead of Bonferroni correction when computing the $\underline{q}$ (see discussion in Section C.4). Had we also used Bonferroni correction for the i.i.d. baseline, both would have taken the same amount of time (afterall, they are both just instances of the same Gaussian smoothing certificate).
> > > > > The time needed for solving an instance of the collective LP (averaged over all images and adversarial budgets) was $0.67s$.
> > > > >
> > > > > For the node classification experiments, the time needed to obtain all Monte Carlo samples was $1034s$ (we used 500000 samples for all certificates; using 10^5 - 10^6 samples is common in randomized smoothing literature).
> > > > > Computing the per-prediction certificates for the naive i.i.d. baseline took $0.11s$.
> > > > > Computing the per-prediction certificates for the naive localized smoothing baseline took $0.66s$ (we use variance smoothing, i.e. both the mean and variance have to be bounded based on the Monte Carlo samples, see Section C.4).
> > > > > Solving the collective LP took an average $10.9s$.
> > > > > The reason that this took longer than solving the LP for image segmentation (even though the number of variables and constraints was smaller than for our image segmentation experiments) was that we used a not-as-well vectorized implementation.
> > > > >
> > > > > In both cases, we see that the cost of computing both the naive certificates and the proposed collective certificate is mainly due to the Monte Carlo sampling step and that the computational overhead from solving the collective LP is small in comparison.
> > > > > It should however be noted that while Monte Carlo sampling is costly, randomized smoothing is still preferable to other, even more expensive certification techniques, e.g. those based on Abstract Interpretation or Semi-Definite Programming. Afterall, its efficiency was one of the main draws of (Cohen et al. (2019) [1]).
> > > > >
> > > > > Following the reviewer’s comment, we have also specified the (average) cost needed for computing the per-prediction certificates in our submission and have updated Section 8 to include more explicit references to Section E, where all measured runtimes and the used hardware are stated.

---

> > > > > ### Author Response · Authors · 2021-11-24
> > > > > **Follow-up, Reviewer z9Ri, Questions 1-2**
> > > > >
> > > > > **1.) The response is very clear. The justification to why this in general not strict equality is crystal clear to me.**
> > > > >
> > > > > We are pleased to hear that.
> > > > >
> > > > > **2.) [...] Since the certification result of Cohen et al with Gaussian noise is tight, considering any region outside the certification ball of Cohen et al there exists an adversary that flips prediction. Based on that, at least under Gaussian smoothing, should not this be exact equality?**
> > > > >
> > > > > The notion of “tightness” used by Cohen et al. (2019) [1] (and other randomized smoothing papers, e.g. (Lee at al. (2019) [2])) is actually somewhat misleading.
> > > > > It does *not* mean that for any region outside the certification ball there exists an adversary that flips the prediction $y$, i.e. a “tight” randomized smoothing certificate is not an “exact” certificate.
> > > > > Instead, a certificate is considered tight if it is the strongest (e.g. largest l2-ball) certificate that holds for *all* smoothed classifiers that predict $y$ with a specific probability.
> > > > >
> > > > > What exactly “tightness” means in the context of randomized smoothing is best demonstrated by examining how the certificate of Cohen et al. (2019) [1] is derived (the same argument can be found in Appendix A and the proof sketch in Section 3.1 of their paper. For the reviewer’s convenience, we use the same notation as in their paper).
> > > > > Given a base classifier $f : \\mathbb{R}^D \\rightarrow \\mathbb{Y}$ and an input $x \in \\mathbb{R}^D$,
> > > > > let
> > > > > $y = \mathrm{argmax}_{y’ \in \mathbb{Y}} \\Pr\_{\\epsilon \\sim \\mathcal{N}(0, \sigma)}  \left[ f(x) = y’ \right]$
> > > > > be the smoothed prediction and and
> > > > > $p\_A = \\Pr\_{\\epsilon \\sim \\mathcal{N}(0, \sigma)}  \left[ f(x + \\epsilon) = y \right]$ be the probability of predicting $y$.
> > > > >
> > > > > To determine whether this prediction is robust to $l_2$ perturbations of magnitude $\delta$, one would ideally want to solve the following optimization problem:
> > > > > $$\\min_{x’ : ||x’ - x||_2 \\leq \\delta}
> > > > > \\Pr\_{\\epsilon \\sim \\mathcal{N}(0, \sigma)}  \left[ f(x' + \\epsilon) = y \right]
> > > > > $$
> > > > > i.e. determine to what extent the adversary can reduce the probability of predicting class $y$.
> > > > > If one were able to compute this exactly, one would obtain an exact certificate for the smoothed classifier.
> > > > >
> > > > > However, even computing the probabilities is not tractable for sufficiently complicated classifiers, as it requires integrating them over the entire input space.
> > > > > Instead, the above optimization problem is relaxed by considering the *worst-case classifier* $f^*$ from a family of classifiers $\\mathbb{F}$
> > > > > $$\\min_{x’ : ||x’ - x||_2 \\leq \\delta}
> > > > > \\min\_{f^* \\in \\mathbb{F}}
> > > > > \\Pr\_{\\epsilon \\sim \\mathcal{N}(0, \sigma)}  \left[ f^*(x' + \\epsilon) = y \right]
> > > > > $$
> > > > >
> > > > > In the case of Cohen et al. (2019) [1], this family of classifiers is the set of all classifiers that, given clean input $x$, predict class $y$ with the same probability as the actual classifier $f$, i.e.
> > > > > $$\mathbb{F} = \\left\\{
> > > > > h : \\mathbb{R}^D \rightarrow \\mathbb{Y}
> > > > > \mid
> > > > >  \\Pr_{\\epsilon \\sim \\mathcal{N}(0, \sigma)} \left[ h(x + \\epsilon) = y \right] = p_A
> > > > > \\right\\}$$
> > > > >
> > > > > For this family of classifiers, the *relaxed* optimization problem that considers the worst-case classifier $f^*$ from $\\mathbb{F}$ can be solved without making any further relaxations. That is what Cohen et al. (2019) [1] mean when speaking of tightness.
> > > > > But, unless the worst-case classifier $f^*$ is identical to the actual base classifier $f$, the relaxed optimization problem will only be a lower bound on the original optimization problem, i.e.~it will not result in an exact certificate.
> > > > > The worst-case classifier $f^*$  and the actual base classifier are only identical when $f$ is a linear classifier (see Fig. 3 of their paper).
> > > > >
> > > > > We hope that the above discussion has elucidated the difference between “tightness” and “exactness” and why our collective certificate is not necessarily an exact certificate, even when using the method of Cohen et al. (2019) [1] as a base certificate (unless our base classifier is a linear classifier).

---

> > ### Author Response · Authors · 2021-11-18
> > **Response Reviewer z9Ri - Questions 5 - 8**
> >
> > **5.) It is not clear to how does the Naive classifier in Figure 2 work. [...] why does the dashed line perform worse than this Naive approach?**
> >
> > The naive approach means performing RS certification for every pixel independently and then counting the number of certificates that guarantee robustness to the collective threat model.
> > The dashed line is this naive approach applied to our localized smoothing base certificates. The used smoothing distributions randomly perturb pixels with standard deviations between $\sigma_{min} = 0.15$ and $\sigma_\max = 1.5$, while the i.i.d. smoothing baseline uses $\sigma = 0.4$ for all pixels.
> > That means that, for the prediction of a specific output, some region of the input is perturbed with less noise, i.e. considered less robust. The adversary can then allocate their whole perturbation budget to this region. Thus, it is expected to perform worse than the i.i.d. smoothing certificate.
> >
> > We have updated the legends of all our figures to better differentiate between the naive collective certificate applied to i.i.d. smoothing, the naive collective certificate applied to localized smoothing and the LP-based collective certificate applied to localized smoothing.
> > We have further updated the second paragraph of our experimental evaluation (Section 8) to clarify the difference.
> >
> > **6.) It is also not clear to me why does solving the linear program reformulation work better than direct RS with anisotropic certificates (dashed line vs solid orange).**
> >
> > A detailed explanation can be found in the first post.
> >
> > To summarize:
> > The proposed collective certificate is the optimization problem
> >  $min_{x' : ||x - x'||_p \leq \epsilon} \sum_n \mathrm{I} \left[x' \in \mathbb{H}^{(n)} \right]$.
> >
> > The naive collective certificate is $ \sum_n min_{x' : ||x - x'||_p \leq \epsilon} \mathrm{I} \left[x' \in \mathbb{H}^{(n)} \right]$.
> >
> > The proposed collective certificate is at least as strong as the naive collective certificate due to the general fact that $min_x \sum_n g^{(n)}(x) \geq  \sum_n min_x g^{(n)}(x)$ for any set of functions $g^{(n)}$.
> > The reason that the above inequality can be a strict inequality, i.e. the collective certificate can provide stronger guarantees, is that we use localized randomized smoothing, i.e. smooth different outputs using *different* anisotropic smoothing distributions. Attacking each output requires manipulating a different part of the input. This causes a budget allocation problem for the adversary, who can not attack all outputs once.
> >
> > **7.) This is related to (6). What is the key motivation/intuition behind believing that solving (3) is better than directly certifying the anisotropic certificates.**
> >
> > Eq.(3) is just Eq.(1) after substituting the definition of our threat model and base certificates, i.e. it is our collective certificate.
> > The same reasoning as in our response to Question 6 applies.
> >
> > **8.) One of the key weaknesses that I see so far is the small certification experiments**.
> >
> > We agree that it would be interesting to evaluate our proposed methods on other datasets. However, we thought it was compelling to show two entirely different applications and datasets for multi-output models and thus also included the node classification setting. In these settings we have focused on using our computational resources to compare our method to existing certificates in a manner that is as thorough and fair as possible.

---

> > ### Author Response · Authors · 2021-11-18
> > **Response Reviewer z9Ri - Questions 1 - 4**
> >
> > **1.) Why is Equation 1 not an equality?**
> >
> > The assumption is that $\mathbb{H}^{(n)}$ can be an under-approximation, i.e. the $\mathbb{H}^{(n)}$ must be valid certificates, but might not actually include all inputs to which the prediction is robust.
> >
> > **2.) In the case where H is given by the localized smoothing using Propositions 1 or 2, will this result in an exact equality for [Equation] 1?**
> >
> > It will only be an exact equality if each of the outputs is a linear function (this is a result of Cohen et al. (2019) [1]). Otherwise, randomized smoothing under-approximates the true certified region.
> >
> > **3.) Can the authors comment on the complexity for solving (5) with the proposed splits in the inputs?**
> >
> > The cost of solving an LP depends on the used solver and the size of the constraint matrix, i.e. the number of constraints and variables. Note that, in our experiments, we only use the linearly relaxed (see Section A.4) and not the mixed-integer problem. Linear programs can be solved efficiently (e.g. in matrix multiplication time, see van den Brand (2019) [2])
> >
> > The general formulation of our LP involves $D_\mathrm{in} + D_\mathrm{out}$ variables and $D_\mathrm{out}+ 1$ constraints. Partitioning the input into $N_\mathrm{in}$ and $N_\mathrm{out}$ subsets and quantizing the base certificate parameters using $N_\mathrm{bin}$
> > quantization bins (see Section A) results in a problem with $N_\mathrm{in} + N_\mathrm{out} * N_\mathrm{bin}$ variables and  $N_\mathrm{out} * N_\mathrm{bin} + 1$ constraints.
> > That is, by partitioning / splitting, we can control the problem size independent of the data dimensionality.
> >
> > In our experiments, we found this computational overhead caused by the LP to be very small compared to the cost of obtaining the randomized smoothing certificates via Monte Carlo sampling. On image segmentation, the sampling needed an average 460 seconds per image, while each LP only took an average 0.67 seconds to solve.
> >
> > Following the reviewer’s question, we have moved the discussion of the problem size from Section A to the main text Section 6. Our new submission also states the compute time and the used hardware.
> >
> > **4.) There are no comparisons against Fischer et al on certifying semantic segmentation.**
> >
> > We omitted this comparison as the goal of the certification is quite different from ours. Instead of determining *how many* outputs are robust to simultaneous attacks, they are interested in determining whether *all* outputs are robust.
> >
> > However, we can compare the methods theoretically.
> > When used to determine the number of robust predictions, the certificate of Fischer et al. (2021) [3] will be at most as strong as the “naive” i.i.d. Gaussian smoothing baseline we use in our experiments, i.e. certifying each prediction independently using the Gaussian smoothing certificate (Cohen at al., (2019) [1]) with Holm correction.
> >
> > To give the reviewer a high-level overview:
> > Fischer et al. define a threshold corresponding to a specific certified radius $r \geq 0$.
> > They then iterate over all predictions $y_n$. If the certificate of Cohen et al. (2019) [1] cannot guarantee robustness for perturbations of magnitude $\leq r$, they abstain from making a prediction. Otherwise, they certify robustness up to radius $r$.
> > They further propose to use Holm correction to account for the multiple comparisons problem.
> > For comparison: Our naive i.i.d. baseline also iterates over all predictions $y_n$, but uses the method of Cohen et al. (2019) [1] to compute the *largest* certifiable radius $r_n$ for each of them.
> > It only abstains if robustness cannot be guaranteed for perturbations of magnitude 0.
> > As $r_n \geq r$ for all non-abstaining predictions and our naive baseline abstains less frequently, there will never be any prediction that is certified by Fischer et al. but not by our baseline.
> > Holm correction can also be used for the naive Gaussian smoothing baseline (and our Gaussian smoothing base certificate), we discuss this in Appendix C.4.
> >
> > We have updated paragraph 2 of experiments Section 8  to specifically state this fact and provide a detailed explanation in Section C.4.

---

> > ### Author Response · Authors · 2021-11-18
> > **Response Reviewer z9Ri - Main changes to the submission**
> >
> > Based on the reviewer’s instructive questions, we have made sure to further improve our submission to ensure that the above arguments are more effectively conveyed to future readers. In particular, we have made sure to:
> > 1.  Stress that localized randomized smoothing means using *different* anisotropic smoothing distributions for different outputs in Sections 1, 4 and 9.
> > 2. In Section 4, highlight that, because we use different smoothing distributions for different outputs, the certified levels of robustness to different parts of the input differ among outputs.
> > 3. Mathematically define the naive collective certificate in Section 4
> > 4. Improve our explanation of why the proposed robustness certificate should be stronger than the naive collective certificate in Section 4

---

> > ### Author Response · Authors · 2021-11-18
> > **Response Reviewer z9Ri - Justification for collective certificate**
> >
> > The following discussion should answer questions (5, 6, 7, 9.2), i.e. explain why our method offers stronger robustness guarantees than independently certifying each prediction using an anisotropic smoothing certificate.
> > While we have already made most of these arguments in our paper (particularly in Sections 1 and 4), we have made several improvements to our submission in order to ensure that they even more effectively conveyed to future readers (see next post).
> >
> > As a running example, we will use Fig 1, i.e. performing image segmentation using our localized randomized smoothing method.
> >
> > The core idea of localized randomized smoothing is that we obtain smoothed predictions by randomly smoothing each output $g_n$ using a *different* anisotropic smoothing distribution $\Psi^{(n)}$ that matches our assumptions about the model’s locality.
> > **For example:** When labeling pixels in the top-right corner of Fig. 1, we apply more Gaussian noise to input pixels in the far-away bottom-left corner and less noise to input pixels in the top-right corner.
> >
> > Randomized smoothing allows us to compute base certificates $\mathbb{H}^{(1)}, \dots, \mathbb{H}^{(n)}$.
> > Importantly, because we use anisotropic smoothing distributions, each base certificate $\\mathbb{H}^{(n)}$ will certify varying levels of robustness for different parts of the input. Because we use different smoothing distributions for different outputs, these certified levels of robustness will vary among outputs.
> > **For example:** Outputs in the top-right corner are more robust to perturbations of the bottom-left corner than to perturbations of the top-right corner. Outputs in the bottom-left corner are more robust to perturbations of the top-right corner than to perturbations of the bottom-left corner.
> >
> > The purpose of our method is to provide a collective certificate, i.e. compute a lower bound on
> > $min_{x’ : ||x - x’||_p \leq \epsilon} \sum_n \mathrm{I}\left[f_n(x') = y_n]\right]$.
> > In fact, there are two possible lower bounds that can be computed using the base certificates:
> >
> > $$\\min\_{x’ : ||x - x’||\_p \\leq \\epsilon} \\sum\_n \\mathrm{I} \\left[f_n(x') = y\_n\\right]
> > \\geq
> > \\min\_{x’ : ||x - x’||\_p \\leq \\epsilon} \\sum\_n \\mathrm{I} \\left[x' \\in \\mathbb{H}^{(n)}\\right]
> > \geq
> > \\sum\_n \\mathrm{I}
> > \\min\_{x’ : ||x - x’||\_p \\leq \\epsilon}
> > \\left[x' \\in \\mathbb{H}^{(n)}\\right].
> > $$
> >
> >
> > As the reviewer correctly pointed out in Question 1, the first inequality holds (and is not necessarily tight), because the $H^{(n)}$ are valid, but not necessarily exact robustness certificates. The **middle term** is our **collective robustness certificate**.
> > The second inequality is just due to the general fact that $\\min\_x \\sum_n g^{(n)}(x) \geq \\min\_x \\sum_n g^{(n)}(x)$ for any set of functions $g^{(n)}$ (i.e. the minimum of a sum of functions is greater than the sum of their minima).
> > The **r.h.s. term** amounts to iterating over all base certificates and checking whether they can certify robustness to the collective threat model with collective adversarial budget $\\epsilon$. It **is the naive collective robustness certificate!** Note that we have a separate minimization problem and a separate $x’$ for each output $f_n$. Thus, this certificate models an adversary that can target each output independently.
> > **For example:** They could first attack each output in the top-right corner of the image by allocating all of their adversarial budget $\epsilon$ to the top-right corner. They could then target each output in the bottom-left corner of the image by allocating all of their adversarial budget to the bottom-left corner.
> >
> > The key question is: Why should the proposed collective certificate be stronger than the naive collective certificate, i.e. why should the second inequality ever be a strict inequality?
> > The important detail is that the collective certificate is a monolithic optimization problem with a single variable $x’$ that corresponds to the adversarially perturbed input.
> > Thus, this optimization problem models that the adversary has to attack all  outputs by using their limited adversarial budget $\epsilon$ to construct a single perturbed input $x’$.
> > Because each output is smoothed using a *different* anisotropic smoothing distributions, attacking each output requires manipulating a different part of the input. This causes a budget allocation problem for the adversary.
> > **For example:** The adversary is still able to target outputs in the top-right corner by allocating all of their adversarial budget to the top-right corner. But, as is encoded by the base certificates, this will have little or no effect on outputs in the bottom-left corner.
> > Because they have already exhausted all of their adversarial budget, they will not be able to attack the outputs in the bottom-left corner at the same time.

---

> ### Author Response · Authors · 2021-11-29
> **Additional experimental results**
>
> As promised in our discussion last week, we are happy to show an additional preliminary experimental result on the graph dataset Citeseer [1]. Since the reviewers Fv4q, y7cf and z9Ri asked for more datasets, we have started to conduct experiments on other graph datasets and also plan to apply our method to other semantic segmentation datasets and models before the camera ready deadline.
> To present new results before the end of the discussion phase, we performed our experiments with a coarser search grid for the smoothing parameters. More experiments and a detailed parameter search will be added to the camera ready version.
>
> For this preliminary experiment, we used a fixed probability of adding bits,
> $\\theta\_\\mathrm{min}^+ = \\theta\_\\mathrm{max}^+= \\theta^+ = 0.01$, i.e. both the baseline and our localized smoothing method flip 1-valued bits with the same probability. We tested six different noise parameters for attribute deletion: $0.4, 0.5, 0.6, 0.7, 0.8, 0.9$. The best-performing baseline based on the area under the certified accuracy curve was the one with $\\theta = 0.8$. We compared this optimized baseline against our smoothing approach with $\\theta\_\\mathrm{min}^-$ = 0.6 and $\\theta\_max^- = 0.95$ with 5 clusters (same as in the Cora ML experiments of our original submission). The base classifier is an APPNP model.
>
> The figures can be found at https://figshare.com/s/663060823a14d65abcda .
> In the scenario with attribute deletion (“citeseer_06_deletion.pdf”) our LP-based localized smoothing approach achieves certified accuracies with an AUC of 11.225 (7.525 naive) compared to the baseline’s 4.558. It outperforms the baseline for all tested adversarial budgets. The scenario of attribute addition (“citeseer_06_addition.pdf”) has to be viewed carefully, as we used a fixed noise level instead of optimizing the distribution parameter to yield a higher certified accuracy. As in our experiments on Cora ML, the LP-based certificate achieves a marginally better certified accuracy AUC of 2.108 (1.633 naive) than the baseline with 2.066, but is outperformed for larger adversarial budgets.
>
> [1] Sen et al., Collective classification in network data, AI Magazine, 29(3):93, 2008

---

### Official Review · Reviewer_y7cf · 2021-10-20

**Correctness:** 3
**Technical Novelty And Significance:** 2
**Empirical Novelty And Significance:** 2
**Recommendation:** 3
**Confidence:** 5

**Main Review:**

Strengths: The studied problem is important and the paper is easy to follow (before evaluation) in general. The proposed method is somewhat novel.

Weaknesses: The key difference between the proposed method and the existing works is unclear. The paper lacks comparison with the existing method. Evaluation is insufficient and some metrics are not reasonable.



The main idea is motivated by several existing works and the key difference between the proposed method and these existing works are unclear for me.
For instance, what’s the key difference between the proposed theoretical results (e.g., Prop 1&2) and those in Eiras et al.? Is it possible for Fischer et al. to be adapted to the considered setting?


No comparison with center smoothing or/and Fischer et al., if possible.

Figures are confused for me. For instance, what is the difference between the “dotted line” and “Naive” in Figure 2? Similar to all the other figures.

Why using certified ratio? I do not think it meaningful to certify pixels/nodes that are wrongly classified.

The image segmentation is only evaluated on Pascal voc2012. I suggest the authors to evaluate on more datasets, e.g., Cityscape, which is used in Fischer et al.
The authors use AUC to report the

It is not standard to use AUC to show image segmentation results.  A commonly used metric is mIoU (mean intersection over union).

UNet is an outdated model for semantic segmentation model. I suggest the authors to evaluate on more recent models such as deeplab v3, danet, hrnet, etc.

What’s the computational overhead to find the optimal $\sigma_{\min}$ and $\sigma_{\max}$?

While computation is an issue for localized RS, the way to partition an image into grid cells and certifying grid cells independently already looses semantic relationship between pixels in different cells and thus violates the purpose of semantic segmentation.



**Summary Of The Paper:**

The authors consider tasks mapping a single input to multiple outputs and study robustness certificate against input perturbations. To achieve the goal, the authors propose a collective certificate where each output is dependent on the entire input but assigns different levels of importance to different input regions, and then derived the collective certificate based on localized randomized smoothing. The proposed collective certificate is evaluated on both image segmentation and node classification tasks.


**Summary Of The Review:**

Collective robustness certificate is an important research problem. The proposed method is novel to some extent, but its key difference with the existing work is not unclear for me, and so as the performance comparison. My another major concern is on evaluation. For instance, the image segmentations results are not reported using reasonable metrics and the models and datasets are rather outdated on my end.

---

> ### Author Response · Authors · 2021-11-18
> **Reponse Reviewer y7cf**
>
> We would like to thank the reviewer for the detailed comments on our submission.
>
> Based on this review, we have made multiple changes to our submission (see also changelog in the post above).
> For the reviewer's convenience, we have highlighted the most important changes to our manuscript in blue.
>
> In the next comment, we answer all questions concerning related work.
> We then provide a separate comment in which we respond to all questions concerning the experimental evaluation.
> Before that, we would like to very briefly summarize our main contributions to make the subsequent discussions easier to follow.
>
> Our main contributions are:
> * Localized randomized smoothing, i.e.  the idea of using *different* anisotropic distributions for smoothing different outputs / sets of outputs and the insight that this smoothing paradigm allows us to retain higher accuracy and provide stronger collective robustness guarantees than randomized smoothing with i.i.d. noise.
> * Identifying that both, existing single-prediction certificates for anisotropic smoothing and novel certificates derived in our submission, share a common interface (i.e. are described by linear inequalities in the adversarial budget space), allowing us to use them interchangeably within our collective certification framework.
> * Variance-smoothing, a novel method for deriving randomized smoothing certificates matching the identified interface. We use it to derive new, efficient certificates for discrete data.
> * A (mixed-integer) linear program for combining the per-prediction certificates obtained through localized randomized smoothing
>
> ### References
> [1] Eiras et al., ANCER: Anisotropic Certification via Sample-wise Volume Maximization, 2021, arXiv:2107.04570
> [2] Fischer et al., Scalable Certified Segmentation via Randomized Smoothing, ICML 2021
> [3] Cohen et al., Certified Adversarial Robustness via Randomized Smoothing, ICML 2019
> [4] Schuchardt et al., Collective Robustness Certificates: Exploiting Interdependence in Graph Neural Networks, ICLR 2021
> [5] Bojchevski et al., Efficient Robustness Certificates for Discrete Data: Sparsity-Aware Randomized Smoothing for Graphs, Images and More, ICML 2020

---

> > ### Author Response · Authors · 2021-11-18
> > **Response Reviewer y7cf - Experimental evaluation**
> >
> >
> > **Figures are confused for me. For instance, what is the difference between the “dotted line” and “Naive” in Figure 2? Similar to all the other figures.**
> >
> > The “naive” line corresponds to performing randomized smoothing with i.i.d. noise and then counting the number of predictions that are certifiably robust to the collective threat model.
> > The dotted line corresponds to applying the same naive collective certification approach to the per-prediction base certificates we obtain through localized smoothing.
> >
> > We have updated the labels in our figure to “naive i.i.d.”, “naive localized” and “LP localized” and added additional explanations to paragraph 2 of Section 8 to clarify this.
> > We have further added a more detailed explanation of the naive collective certificate to section 4.
> >
> > **Why using certified ratio? I do not think it meaningful to certify pixels/nodes that are wrongly classified.**
> >
> > We agree that certified accuracy is a more meaningful metric for supervised learning tasks, which is why we performed our experiments using both certified accuracy and certified ratio (which was used in prior work on randomized smoothing on graphs, e.g. Bojchevski et al. (2020) [5]).
> >
> > Experiments featuring both certified accuracy and certified ratio could already be found in the appendix of our original submission.
> > Based on the reviewer’s comments, we have moved the certified accuracy figure to the main section and the certified ratio figure to the appendix.
> >
> > **The authors use AUC to report the image segmentation results. It is not standard to use AUC to show image segmentation results.**
> >
> > This appears to be a misunderstanding. At the beginning of our experimental evaluation, we explain that “We further report the AUC of these metrics [(certified accuracy and ratio)] w.r.t $\epsilon$”, i.e. we compute the AUC of the metrics we use to assess certificate strength.
> > We do not use our method to certify some AUC function.
> >
> > Our updated submission clarifies this when introducing the metrics in paragraph 2 of Section 8. We have further changed all occurances to “AUC of the certified accuracy curve” or similar formulations.
> >
> > **What’s the computational overhead to find the optimal $\sigma_\mathrm{min}$ and $\sigma_\mathrm{max}$ and ?**
> >
> > As in other randomized smoothing certificates, the smoothing distribution parameters are hyperparameters to be chosen by the user.
> >
> > **While computation is an issue for localized RS, the way to partition an image into grid cells and certifying grid cells independently already looses semantic relationship between pixels in different cells and thus violates the purpose of semantic segmentation.**
> >
> > It is not entirely clear to us what exactly is meant with “loosing semantic relationship between pixels”. It would be great if the reviewer could elaborate on this in more detail.
> > If this is supposed mean that we segment / label each part of the image independently:
> > This is not the case. We use different smoothing distributions for labeling different parts of the image (this is the basic idea behind localized smoothing), but still let each output operate on the entire input image.
> >
> > **the paper is easy to follow (before evaluation)**
> >
> > Firstly, we are happy to hear that the reviewer found the motivation for our work and the derivation of our method easy to follow.
> > We believe that changes we have made to the experiment section following the reviewer’s feedback and that of other reviewers (e.g. more explicit references  to appendix, disentangling discussion of example figures and the experiments at large, more detailed explanation of metrics, better labeling of figures, better explanation of baselines etc.) should make it much easier to follow for future readers.
> >
> > Finally, we would like to respond to the reviewer’s suggestion of **using more recent segmentation architectures than U-Net and performing experiments on Cityscapes instead of Pascal VOC.**
> >
> > While we agree that it could be interesting to apply our method to additional classifiers and datasets, any of the used datasets and classifiers only serve as a means of comparing our method to existing certificates.
> > We therefore opted to use one of the most well-known and well-established baselines for image segmentation, even if it has been superseded by more recent architectures.
> > We would further like to point out that this is common practice in robustness certification. For example, randomized smoothing certificates for image classification are typically evaluated using AlexNet-like or vanilla ResNet models (e.g. in Cohen et al. 2019 [3]), although they are outmatched by more recent architectures..
> > We have updated the first paragraph of our experimental evaluation (Section 8) to better convey this to future readers.
> >
> > Should the reviewer have any further questions or would like to provide additional comments, we would be happy to respond.

---

> > ### Author Response · Authors · 2021-11-18
> > **Response Reviewer y7cf - Related work**
> >
> > In this comment, we explain how our method is related to prior work, beginning with the questions concerning the work of Eiras et al. (2021) [1] and Fischer et al. (2021) [2].
> >
> > **For instance, what’s the key difference between the proposed theoretical results (e.g., Prop 1&2) and those in Eiras et al.?**
> >
> > As stated in our submission, Prop 1&2 (Gaussian and uniform smoothing) are not novel results, we “merely express [them] in a manner that matches our interface”. The actual contribution is identifying that they share this interface and can thus be used as interchangeable components within our collective robustness certification method.
> > Note that the certificates of Eiras et al. (2021) [1] were originally derived to certify individual predictions, not for certifying collective robustness.
> > In addition to using this existing work, we also derive novel certificates for discrete data that match this interface.
> >
> >
> > To highlight this, we have updated Section 5 of our submission to make this clarification before the individual base certificates are discussed. We have further renamed the newly derived certificates for discrete data  to “Theorems” 1 and 2, to differentiate them from the existing work that we merely use in a novel way.
> >
> > **Is it possible for Fischer et al. to be adapted to the considered setting?**
> >
> > If “setting” refers to “Certifying the number of robust predictions”, then yes. However, the resulting certificate will be at most as strong as the “naive” i.i.d. Gaussian smoothing baseline we use in our experiments, i.e. certifying each prediction independently using the Gaussian smoothing certificate (Cohen at al., 2019) [3] with Holm correction.
> >
> > If “setting“ refers to “using Fischer et al. as a base certificate”, then also yes. Again, using (Cohen at al., (2019)) [3] with Holm correction will yield better results.
> >
> > We have expanded our discussion of the multiple comparisons problem (Appendix C.4) to discuss Holm correction and the adaptation of Fischer et al. “to our setting” in more detail.
> > We have further updated the second paragraph of our experimental evaluation (Section 8) to clarify this.
> >
> > To give the reviewer  a high-level overview of why the method of Fischer et al. (2021) [2] is weaker:
> > Fischer et al. define a threshold corresponding to a specific certified radius $r \geq 0$.
> > They then iterate over all predictions $y_n$. If the certificate of Cohen et al. (2019) [3] cannot guarantee robustness for perturbations of magnitude $\leq r$, they abstain from making a prediction. Otherwise, they certify robustness up to radius $r$.
> > They further propose to use Holm correction to account for the multiple comparisons problem.
> > For comparison: Our naive i.i.d. baseline also iterates over all predictions $y_n$, but uses the method of Cohen et al. (2019) [3] to compute the *largest* certifiable radius $r_n$ for each of them.
> > It only abstains if robustness cannot be guaranteed for perturbations of magnitude 0.
> > As $r_n \geq r$ for all non-abstaining predictions and our naive baseline abstains less frequently, there will never be any prediction that is certified by Fischer et al. (2021) [1] but not by our baseline.
> >
> > **No comparison with center smoothing or/and Fischer et al., if possible.**
> >
> > We experimentally compare to center smoothing (the results can be found in Fig. 7&8 and Tables 2&3 in the appendix) and our method compares favorably to it.
> > Based on this comment, we have made sure to explicitly state that these tables and figures can be found in the appendix -- not the main section -- and to summarize the contents of the appendix.
> > As explained above, the naive i.i.d. smoothing baseline is actually a stronger certificate than the method of Fischer et al. (2021) [1] (for our notion of collective robustness).
> >
> > **Finally, we would like to discuss how our method is related to that of   of Schuchardt et al. (2021) [4]**
> >
> > Both share the same high-level idea of combining per-prediction certificates into a collective certificate, but their method is substantially different.
> > It is designed for binary data and strictly local models, i.e. each output operates on a small receptive field. Their certificate essentially amounts to masking out perturbations outside these receptive field. Our certificate instead leverages the locality encoded by our localized smoothing  base certificates (i.e. smoothing different outputs using different anisotropic distributions) and as such does not require strict locality.
> >
> > Based on the feedback of Reviewer Fv4q, we have added a discussion on how our approach and how the resulting linear program are related to that of Schuchardt et al. (2021) [4] to  Sections 4 and 6. We have further added a new section (Section D) discussing this in more detail.

---

> ### Comment · Reviewer_y7cf · 2021-11-29
> **Author's response did not address my main comments**
>
> The authors clarify some of my comments. However, I still think the novelty does not reach the bar and the evaluation is insufficient (datasets, metrics, running time, etc). I will maintain my score.

---

### Official Review · Reviewer_ruKP · 2021-11-02

**Correctness:** 4
**Technical Novelty And Significance:** 3
**Empirical Novelty And Significance:** 3
**Recommendation:** 8
**Confidence:** 4

**Main Review:**

Strengths
- The collective certification strategy in the paper can be used with most of the existing literature on randomized smoothing, which allows it to be quite versatile and adapt to new developments in the field. The MILP relaxation and the distribution sharing ideas in the paper also highlight some critical limitations/ future research directions for randomized smoothing.
- The variance smoothing idea in the paper is quite exciting and a natural next step for getting better certificates.
- The empirical results in Table 5 that compare the best achievable baseline performance with the performance of the proposed methods provide compelling evidence for the efficacy of the suggested approach.

Weaknesses
- No training counterpart is suggested for the proposed local smoothing strategy. The current results use a model trained with $\sigma_\min$ as the base model. It seems a bit counterintuitive to have essentially different prediction strategies during training and testing. Some of the local smoothing ideas should be reflected in training to make the objectives better aligned.

**Summary Of The Paper:**

The paper makes three contributions in particular:
- A local version of randomized smoothing for multi-output classifiers. The authors suggest using a customized smoothing distribution for certifying each output of the multi-output classifier. The custom distributions allow them to produce tighter guarantees for each output.
- A new analysis method of variance smoothing for discrete data uses the average softmax value instead of the majority vote as the prediction rule. The authors use the first and second-order statistics (mean and variance) to provide robustness guarantees in this method.
- A collective certification strategy for multi-output classifiers using a common interface ($\ell_p$ norm ellipsoids) for base certificates for every output. The authors describe a common way of stating the base certified regions for every output. Then the multi-output certification problem can be expressed as a mixed-integer linear program to find a point inside the perturbation model that lies outside the base certified regions for the maximum number of outputs.

**Summary Of The Review:**

In summary, the paper explores a very relevant problem in certification against adversarial examples. The main ideas of the paper are pretty novel and would help future research. The paper does a great job of presenting the ideas and discussing some of the potential limitations. However, it would be better if the authors briefly discussed the mismatch between the training objectives and the test time prediction model. Taking all of this into account, I would recommend the paper be accepted.

---

> ### Author Response · Authors · 2021-11-18
> **Response Reviewer ruKP**
>
> We would like to thank the Reviewer for the thorough review of the manuscript and the positive assessment.
>
> We agree with the Reviewer that the training and testing schemes should ideally be aligned to achieve optimal results. However, for the present study, we wanted to ensure that our method was not favored in any way. As localized smoothing allows for more distribution parameters, we wanted to demonstrate the positive results are not due to a better-trained model (which could potentially be obtained by tuning the additional hyperparameters). To this end, we used i.i.d. smoothing distributions for training all models, which skews the results in favor of our baselines which encounter the same distribution at training and test time.

---

### Official Review · Reviewer_Fv4q · 2021-11-02

**Correctness:** 3
**Technical Novelty And Significance:** 2
**Empirical Novelty And Significance:** 2
**Recommendation:** 3
**Confidence:** 3

**Main Review:**

This paper has many merits:
- The motivation behind this work is clearly stated.
- The use of anisotropic smoothing is intuitive.
- The tasks on which the experiments are conducted are in line with the motivation.

However, there several concerns in this work that need to be addressed:
- While the major part of this work is dedicated to the theoretical analysis, it is not clear to me which parts belong to the contributions of this work and which parts are restatements from other work. For example:
1. In terms of the theoretical results in this work in sections 4, 5, and 6: what is exactly new (considered as a contribution)? For example, Proposition 1 and 4 are just a special case of the results of Eiras et.al. for when the covariance matrix is diagonal. In fact, with the formulation setup presented in section 2, a generalization of proposition 1 can be found in Appendix A of [1].

2. In terms of the analysis in sections 4 and 6, what is exactly new? For example, the bound derived in Equation (1) and the analysis in (3-6) are very similar to the results in Schuchardt et. al.

- In terms of experiments:
1. it is mentioned that "showcasing state-of-the-art accuracy on datasets is not part of our objective". Why? If the proposed collective anisotropic certificate is preferable, then it is necessary to show that it improves the best baselines.

2. What is the importance of certified ratio metric? Why not reporting wither certified accuracy or certified AUC in the case of segmentation. While two models could have similar certified ratios, their accuracy could significantly differ.

3. Since the analysis of this work follows the work of Schuchardt et. al., a direct comparison on the strictly local setup should be presented.

4. The experiments conducted include a single benchmark per task. The experiments should include multiple datasets to check whether the assumptions related to the use of anisotropic smoothing hold or not (e.g. homophily).

- The writing of the paper could be significantly improved. There are several parts were the text refer to tables/figures in the appendix without mentioning the appendix. Also, please consider moving more experiments from the appendix to the main work. Moreover, in the caption of Figure 1, $\sigma_{\text{min}}$  is mentioned twice, is that a typo? It is also repeated in page 8 in the second paragraph.



[1]: Certified Defense to Image Transformations via Randomized Smoothing, NeurIPS 2020.

**Summary Of The Paper:**

This paper leverages the recent anisotropic certificates for randomized smoothing for certifying multi-output classifiers. In particular, it leverages anisotropic Gaussian  and Bernoulli smoothing for better collective robustness. Experimental evaluation was conducted on semantic segmentation and node classification to demonstrate the effectiveness of the proposed method.

**Summary Of The Review:**

While the main motivation of this work is clear, there are several parts of this work that need to be revised. In particular, it should be clearly stated which parts of the theoretical results in this work are new. Also, the experimental part of this work can be significantly improved by including stronger baselines and more datasets.

---

> ### Author Response · Authors · 2021-11-18
> **Response Reviewer Fv4q**
>
> We would like to thank the Reviewer for the thorough review of the manuscript and their insightful comments.
> Before responding to their points of criticism in detail, we would like to address their primary question about which parts of our work are novel contributions.
> We will then provide a detailed overview of which results and which parts of the analysis are novel in the next post.
> In the final post, we will discuss their questions concerning the experimental evaluation.
>
> We would further like to point out that we have made multiple improvements to our submission based on the provided feedback, and have highlighted the most important changes to our manuscript in blue.
>
> As stated in the bullet points at the end of the introduction, our work makes the following novel contributions:
> * Localized randomized smoothing. Yes, randomized smoothing with anisotropic noise for single-output classifiers has already been derived. But the idea of using *different* anisotropic smoothing distributions for different outputs (or sets of outputs) of a multi-output model and the insight that this allows us to both retain high accuracy and provide stronger collective robustness guarantees is entirely novel.
> * Our interface for “base certificates”, i.e. single-prediction certificates that we later combine into a collective certificate. We show that certificates for anisotropic smoothing are described by linear constraints in the adversarial budget space (this includes novel certificates derived in our work).
> This makes it possible for them to be used as interchangeable components within our collective certification framework and makes our method applicable to various data types and perturbation models, unlike the method of Schuchardt et al. [1], which is only designed for binary (graph-structured) data.
> * Variance-smoothing, a novel, general-purpose method for obtaining certificates matching our interface. We use this method to derive new and efficient certificates for anisotropic / non-i.i.d. smoothing of discrete data.
> * A collective certificate based on the proposed localized randomized smoothing approach and the identified/proposed interface.
> As stated in our related work section “[Like our certificate], it [(the certificate of Schuchardt et al. [1])]  combines many per-prediction certificates via a (mixed-integer) linear program”.
> While our MILP bears some similarity to theirs, ours is conceptually different and more efficient to solve. We discuss more details in our responses to some of the later questions.
>
> To ensure that the above points are also clear to future readers, we have used the reviewer’s feedback to make the following changes to our submission:
> * Sections 1, 4 and 9: Changes to highlight that localized randomized smoothing means using *different* distributions for different outputs (or sets of outputs), i.e. it is not a trivial application of anisotropic smoothing to multi-output models
> * Section 4: Added a paragraph (paragraph 5) discussing the similarities and differences between our method and that of Schuchardt et al. [1]
> * Section 5: Clarified which base certificates are novel and which are results of prior work before discussing the individual certificates.
> * Renamed “Propositions” 2 and 3 to “Theorems” 1 and 2, to highlight that these certificates are novel contributions.
> * Section 6: Added a paragraph (paragraph 4), comparing our derived MILP to that of Schuchardt et al.
> * Added a new section (Section D), providing a detailed discussion of the certificate of Schuchardt et al. [1]
> * Attributed the certificate for anisotropic Gaussian smoothing to Fischer et al. (2020) [2], who derived it before Eiras et al. (2021) [3]
>
>
> ### References:
> [1] Schuchardt et al., Collective Robustness Certificates: Exploiting Interdependence in Graph Neural Networks, ICLR 2021
> [2] Fischer et al., Certified Defense to Image Transformations via Randomized Smoothing, NeurIPS 2020.
> [3] Eiras et al., ANCER: Anisotropic Certification via Sample-wise Volume Maximization, 2021, arXiv:2107.04570
> [4] Bojchevski et al., Efficient Robustness Certificates for Discrete Data: Sparsity-Aware Randomized Smoothing for Graphs, Images and More, ICML 2020

---

> > ### Author Response · Authors · 2021-11-18
> > **Response Reviewer Fv4q - Experimental evaluation**
> >
> > Finally, we want to address the reviewer’s comments on our experimental evaluation:
> >
> > **It is mentioned that "showcasing state-of-the-art accuracy on datasets is not part of our objective". Why? If the proposed collective anisotropic certificate is preferable, then it is necessary to show that it improves the best baselines.**
> >
> > This statement was perhaps ill-chosen. We are indeed interested in showing that our proposed method is preferable to currently available alternatives and we do so in our experiments.
> > What we meant to say is that selecting and training the perfect base model to achieve SOTA classification accuracy is not our main objective. Instead, as in other work on robustness certification, any dataset and model merely serves as a means of evaluating the efficacy of different certification approaches. For example, the paper the reviewer cited (Fischer et al., 2020 [2]) performs experiments on vanilla ResNet models instead of EfficientNet, but this does not invalidate their results.
> >
> > **What is the importance of certified ratio metric? Why not reporting either certified accuracy or certified AUC in the case of segmentation. While two models could have similar certified ratios, their accuracy could significantly differ.**
> >
> > We agree that certified accuracy is a more meaningful metric for supervised learning tasks, which is why we performed our experiments using both certified accuracy and certified ratio (which was used in prior work on randomized smoothing on graphs, e.g. Bojchevski et al. 2020 [4]). Experiments using both metrics could already be found in the appendix of our original submission.
> > We followed the reviewer’s suggestion and moved the certified accuracy figure to the main section and the certified ratio figure to the appendix.
> >
> > **Since the analysis of this work follows the work of Schuchardt et. al., a direct comparison on the strictly local setup should be presented.**
> >
> > The goal of our work was to develop a collective robustness certificate for models that are not strictly local. Whether or not we are also more effective in the strictly local scenario is not really relevant to this goal.
> > However, we have added an additional section (Section D.2) showing that our method subsumes the certificate of Schuchardt et al. (2020) [1] in the strictly local setup when using the sparsity-aware smoothing distribution used in their paper.
> > Thus, there is no need to perform an experimental comparison.
> > To be more specific: For strictly local models, completely randomizing all input dimensions outside an output’s receptive field (e.g. by flipping bits with a probability of 50%) is equivalent to the masking operation in the certificate of Schuchardt et al. [1]
> >
> > **The experiments conducted include a single benchmark per task. The experiments should include multiple datasets to check whether the assumptions related to the use of anisotropic smoothing hold or not (e.g. homophily).**
> >
> > While we are not opposed to applying our method to other datasets, we do not believe that performing additional experiments to verify that specific models are softly local is necessary.
> > As explained in the introduction of our paper, many GNNs are homophilic by design, it is well-established that CNNs tend to have small effective receptive fields etc.
> > Furthermore, these are properties inherent to the classifier and as such not dependent on the dataset.
> >
> > **EDIT, November 29**: Following our discussion with Reviewer z9RI we have performed a preliminary experiment on an additional graph dataset (see comment above). As expected, the results are very similar to those on Cora ML. We plan to conduct experiments on other graph and image segmentation datasets before the camera ready deadline.
> >
> > **There are several parts where the text refers to tables/figures in the appendix without mentioning the appendix. Also, please consider moving more experiments from the appendix to the main work. Moreover, in the caption of Figure 1, $\sigma_\mathrm{min}$ is mentioned twice, is that a typo? It is also repeated in page 8 in the second paragraph.**
> >
> > * We have made sure to include more explicit references to the Appendix before referencing figures / tables in the experimental section.
> > * While we would in principle like to move more figures and tables to the main section, most  experiments exhibit a similar pattern (Localized randomized can provide stronger robustness guarantees and higher accuracy. The certificate obtained through linear programming is stronger than naively combining the anisotropic smoothing certificates). As such, making this change would not really add much to the discussion.
> > * We have fixed the typo the reviewer pointed out.
> >
> > We hope that we have addressed all comments to the reviewer’s satisfaction and would like to thank them for their feedback, which has helped us to improve our manuscript and show more clearly what our novel contributions are. We would be happy to respond to any additional questions or comments they may have.

---

> > ### Author Response · Authors · 2021-11-18
> > **Reponse Reviewer Fv4q - Novel contributions / Analysis (detailed)**
> >
> > This post lists in detail which specific part of which of the mentioned sections are new and why they should be considered novel.
> >
> > **In terms of the theoretical results in this work in sections 4, 5, and 6: what is exactly new (considered as a contribution)?**
> >
> > * Section 4:
> >   * Localized randomized smoothing (paragraph 1) is new.
> >   * Eq. 1, i.e. the high-level idea of combining per-prediction certificates into a collective certificate is not new, as explained in the related work section. Even though Eq. 1 is never explicitly stated by Schuchardt et al., we do not claim the equation itself to be a core contribution.
> >   * Identifying that localized randomized smoothing - because it results in base certificates that encode the smoothed model's locality - makes Eq.1 a better collective certificate than certifying each output independently is new. (Schuchardt et al. [1] instead base their analysis on a-priori knowledge about a model's strict locality, i.e. predictions are only affected by perturbations to specific parts of the input).
> > * Section 5:
> >   * Identifying the common interface for base certificates obtained through anisotropic smoothing (Eq. 2) is novel. Schuchardt et. al. [1] use a much more inefficient representation that is limited to discrete adversarial budgets, not designed for anisotropic certificates and involves a potentially very large number of linear constraints (one constraint per point in the “pareto front” of the certificate in adversarial budget space. For more details, please refer to Section 3 of their paper).
> >   * Proposition 1 and 4 are not novel. As stated “We merely express them [(the existing results)] in a manner that matches our interface”.
> >   * The fact that Proposition 1 and 4 can be used as  interchangeable components within our framework for collective robustness certification is new. We included these clarifications in the updated version of our submission.
> >   * Proposition 2 and 3 are new certificates derived using our novel variance smoothing method.
> > * Section 6:
> >   * The derived collective MILP is new.
> >   * Identifying a way of controlling the LP problem size independent of the data dimensionality is new.
> >
> > **In terms of the analysis in sections 4 and 6, what is exactly new?**
> > * Section 4:
> >   * As stated above, identifying that -- because we use localized randomized smoothing -- using Eq.1 is better than certifying each output independently is new.
> >
> > * Section 6:
> >   * In this section, we first insert the definition of our perturbation model and base certificates into Eq.1 (see Eq. 3).
> > Because their definitions are entirely different from those used in Schuchardt et al. [1], the resulting optimization problem is different.
> >   * We do not need to perform an additional step of masking out adversarial perturbations outside the outputs’ receptive fields (Eq. 11 in their paper), as our approach is not relying on strict locality, but instead leverages the locality encoded by the base certificates.
> >   *  Eq.5 (replacing indicator variables as boolean variables) is a common modelling technique.
> >   * We do agree that Eq. 4 (i.e. optimizing over adversarial budgets instead of perturbed inputs) is similar, although we do not claim it to be a significant contribution.
> >   * Because our original optimization problem is different from that of Schuchardt et al. [1], the resulting MILP is also substantially different. It does not involve any masking operations and is much more efficiently solvable, as it only requires a single linear constraint per prediction (unlike the “pareto front”-based approach we discussed before).
> >
> > We hope that the above explanations have clarified what our contributions are, how our work differs from that of Schuchardt et al. [1] and that we use both existing and novel anisotropic smoothing certificates as interchangeable components within our certification framework.

---

> ### Author Response · Authors · 2021-11-29
> **Additional experimental results**
>
> We are happy to show an additional preliminary experimental result on the graph dataset Citeseer [1]. Since the reviewers Fv4q, y7cf and z9Ri asked for more datasets, we have started to conduct experiments on other graph datasets and also plan to apply our method to other semantic segmentation datasets and models before the camera ready deadline.
> To present new results before the end of the discussion phase, we performed our experiments with a coarser search grid for the smoothing parameters. More experiments and a detailed parameter search will be added to the camera ready version.
>
> For this preliminary experiment, we used a fixed probability of adding bits,
> $\\theta\_\\mathrm{min}^+ = \\theta\_\\mathrm{max}^+= \\theta^+ = 0.01$, i.e. both the baseline and our localized smoothing method flip 1-valued bits with the same probability. We tested six different noise parameters for attribute deletion: $0.4, 0.5, 0.6, 0.7, 0.8, 0.9$. The best-performing baseline based on the area under the certified accuracy curve was the one with $\\theta = 0.8$. We compared this optimized baseline against our smoothing approach with $\\theta\_\\mathrm{min}^-$ = 0.6 and $\\theta\_max^- = 0.95$ with 5 clusters (same as in the Cora ML experiments of our original submission). The base classifier is an APPNP model.
>
> The figures can be found at https://figshare.com/s/663060823a14d65abcda .
> In the scenario with attribute deletion (“citeseer_06_deletion.pdf”) our LP-based localized smoothing approach achieves certified accuracies with an AUC of 11.225 (7.525 naive) compared to the baseline’s 4.558. It outperforms the baseline for all tested adversarial budgets. The scenario of attribute addition (“citeseer_06_addition.pdf”) has to be viewed carefully, as we used a fixed noise level instead of optimizing the distribution parameter to yield a higher certified accuracy. As in our experiments on Cora ML, the LP-based certificate achieves a marginally better certified accuracy AUC of 2.108 (1.633 naive) than the baseline with 2.066, but is outperformed for larger adversarial budgets.
>
>
> [1] Sen et al., Collective classification in network data, AI Magazine, 29(3):93, 2008

---

### Author Response · Authors · 2021-11-18
**Rebuttal summary**

While we provided detailed responses to the individual reviews and made subsequent improvements to our submission, we found that they shared many questions and criticisms that were due to some confusion concerning how our method relates to prior work and collective robustness certification in general.
We would therefore like to provide a short summary that addresses most of the reviewers’ questions and comments.
After that, we provide a changelog summarizing all modifications we made to our manuscript.

The goal of collective robustness certification is to provide simultaneous robustness guarantees for predictions made by a multi-output model operating on a single input.
Attacking such models is potentially much more challenging for an adversary, as they have to decide on a single perturbed input to manipulate multiple predictions, instead of being able to craft a separate adversarial example for each of the outputs.
This constraint has to be explicitly taken into account to obtain tight robustness guarantees.
Most existing methods do not focus on this collective threat model and can only be used to certify each output independently. This is a detrimental relaxation, as it ignores the collective constraint on the adversary, and will thus result in weak robustness guarantees.

Our work represents the first method that models a proper collective threat model and can be applied to arbitrary multi-output classifiers (Deep CNNs, Transformers, RNNs, GNNs …)
It consists of three steps:
1. Localized randomized smoothing, i.e. smoothing the different outputs of a model using *different* non-i.i.d. smoothing distributions.
2. Certifying the robustness of each of the outputs’ predictions individually, using either existing (for continuous data) or novel (for discrete data) methods.
Because we use different non-i.i.d. distributions for different outputs,
each output has its own *distinct* certificate quantifying how robust it is to perturbations of different parts of the input.
3. Leveraging these locality guarantees provided by the per-prediction base certificates to merge them into a stronger collective certificate.

From this explanation, which we also provide in the introduction of our paper, it is clear that **our method is substantially different from any prior work and presents a significant advance in the only recently emerging subfield of collective robustness certification:**

The only dedicated collective robustness certificate (Schuchardt et al., 2021) [1] prior to our submission is only similar in its high-level idea of combining single-prediction “base certificates” into a stronger collective certificate and the fact that it uses linear programming to do so. It is designed for models whose outputs have a known and well-defined receptive field (not deep CNNs etc.). Instead of leveraging the locality encoded by base certificates - which is only made possible by our localized randomized smoothing approach -  it amounts to masking out perturbations falling outside the receptive field. Thus, their linear program is an entirely different optimization problem.

Fischer et al., (2021) [2] consider a different notion of collective robustness and try to determine whether *all* predictions of a model are robust. Their method is not suitable for determining *how many*  predictions are robust. When used for our setting, their method amounts to certifying each prediction independently (the “naive collective certificate” discussed in our work).

*Non-collective* randomized smoothing certificates using non-i.i.d. smoothing distributions have been derived in prior work (Fischer et al., (2020) [3]; Eiras et al., (2021) [4]).
We do not claim them to be our own contributions, unlike the novel certificates for non-i.i.d. smoothing of discrete data derived by us.
However,
* Using *different* non-i.i.d. distributions for different outputs,
* noticing that non-i.i.d. smoothing certificates encode the locality of a model,
* leveraging the encoded locality for collective robustness certification,
* identifying that various certificates are sufficiently similar (“implement the same interface”) to be used as interchangeable components in our collective certificate

are all novel contributions.

### References:
[1] Schuchardt et al., Collective Robustness Certificates: Exploiting Interdependence in Graph Neural Networks, ICLR 2021
[2] Fischer et al., Certified Defense to Image Transformations via Randomized Smoothing, NeurIPS 2020.
[3] Fischer et al., Scalable Certified Segmentation via Randomized Smoothing, ICML 2021
[4] Eiras et al., ANCER: Anisotropic Certification via Sample-wise Volume Maximization, 2021, arXiv:2107.04570

---

> ### Author Response · Authors · 2021-11-18
> **Changelog**
>
> ### Main changes (highlighted in blue in manuscript):
>
> * Stress that localized smoothing means using *different* non-i.i.d. smoothing distributions for different outputs (Sections 1, 4 and 9).
> * Formally define naive collective certificate (Section 4)
> * Simplify explanation of why proposed certificate is better than naive collective certificate (Section 4)
> * Provide more detailed high-level comparison of our certificate to that of Schuchardt et al. (2020) [1] (Section 4)
> * Discuss difference of our derived MILP to that of Schuchardt et al. (2021) [1] (Section 6)
> * Add new Section D, discussing Schuchardt et al. (2021) [1] in detail and showing that our certificate subsumes theirs when using sparsity-aware randomized smoothing base certificates (the same smoothing distribution as in their paper)
> * State which base certificates are based on prior work and which are novel in paragraph 2 instead of specifying it for each method separately (Section 5)
> * Discuss why the naive baseline is an upper bound on the segmentation certificate of Fischer et al. (2021) [3] (if it were used for certifying the number of robust predictions) (Section C.4)
> * Disentangle discussion of example figures, parameter search procedure and additional experiments in Section 8
> * Specify runtime (and used hardware and solver) for experiments (Section 8 and E)
>
> ### Other changes
>
> * Section 1:
>   * State that capturing budget allocation problem in collective certificate is what makes it stronger
>   * Reduce size of Fig. 1 by 20%
> * Section 2:
>   * Remove redundant comparison of i.i.d. Gaussian smoothing with our approach
>   * Point to detailed comparison with Schuchardt et al. (2021) [1] certificate in appendix
>   * Point to detailed comparison with Fischer et al. (2021) [3] certificate in appendix
>   * Highlight that existing anisotropic smoothing certificates are only one component of our overall approach
> * Section 4:
>   * Further clarify why we use anisotropic smoothing certificates (capture locality)
>   * Write $\\mathbb{B}\_x$ instead of directly inserting the definition of collective perturbation model
> * Section 5:
>   * Attribute anisotropic Gaussian smoothing to Fischer et al. (2020) [2]
>   * Simplify explanation of exponential blow-up when using existing certificate for   discrete localized smoothing
> * Section 6:
>   * “Improving efficiency” paragraph: State number of problem variables / constraints, instead of only specifying them in Section A.
> * Section 7:
>   * Slightly shorten to save 2 lines
> * Section 8:
>   * Clarify that we are interested in SOTA robustness certification, but that any used classifier serves only as a test-bed for certificates, does not have to be latest architecture
>   * Explain that performed experiments are also valid comparisons to Schuchardt et al. (2021) [1] and Fischer et al. (2021) [3]
>   * Clarify that we are interested in AUC of certified robustness/ratio curves, not in certifying AUC of some curve
>   * Clearer figure labels (Naive i.i.d. vs Naive localized vs LP localized)
>   * Separate paragraph for discussion of appendix Section E experiments.
>   * Simplify explanation of parameter search procedure for node classification task
>   * Move discussion of comparing models trained on same smoothing distribution to Section E.
>   * Say that used regular grid is similar to Fig. 1, but not identical
> * Section E
>   * Use “AUC of certified accuracy curve” instead of “certified accuracy AUC”
>
> ### Typos:
> * Section 2
>   * Specify distribution in definition of q
>   * Add missing words to “probability of g [predicting label] y”
> * Section 6:
>   * Add missing word to “interface [for] base certificates”
> * Section 8
>   * Fix duplicate $\\sigma_\mathrm{min}$ in figure captions and text
>   * Replace "Figure Fig. 2" with "Fig. 2"

---

### Decision · Program_Chairs · 2022-01-20

**Decision:**

Reject

**Comment:**

The authors develop a framework for improving robustness certificates obtained by randomly smoothed classifiers in settings with multiple outputs (segmentation or node classification), by combining local robustness certificates obtained for individual classifiers. They validate their results empirically and demonstrate gains from their approach.

The reviewers were mostly in agreement that the authors make a novel and interesting contribution. However, there were a lot of technical concerns raised by reviewers that, while addressed during the discussion phase, would require a substantial revision of the paper to address adequately. Overall, I feel the paper is borderline but recommend rejection and encourage the authors to incorporate feedback from the reviewers and submit to a future venue.